# On the Interaction of Batch Noise, Adaptivity, and Compression, under $(L_0, L_1)$-Smoothness: An SDE Approach

Enea Monzio Compagnoni [1]   Rustem Islamov [1]   Frank Norbert Proske [2]
Aurelien Lucchi [1 †]   Antonio Orvieto [3 4 5 †]   Eduard Gorbunov [6 †]

## Abstract

Distributed stochastic optimization intertwines (i) stochastic gradient noise, (ii) communication compression, and (iii) adaptive/normalized updates. While each factor has been studied in isolation, their joint effect under realistic assumptions remains poorly understood. In this work, we develop a unified theoretical framework for Distributed Compressed SGD (DCSGD) and its sign variant Distributed SignSGD (DSignSGD) under the recently introduced $(L_0, L_1)$-smoothness condition. From a conceptual perspective, we show that the first- and second-order modified equations from the literature do not accurately model the discrete-time stepsize/stability restrictions, especially under $(L_0, L_1)$-smoothness. From a technical perspective, we propose new *first*-order SDEs by carefully incorporating curvature-dependent terms into their drift: This helps capture the fine-grained relationship between learning rate restrictions, gradient noise, compression, and the geometry of the loss landscape. Importantly, we do so under general gradient noise assumptions, including heavy-tailed and affine-variance regimes, which extend beyond the classical bounded-variance setting. Our results suggest that normalizing the updates of DCSGD emerges as a natural condition for stability, with the degree of normalization precisely determined by the gradient noise structure, the landscape's regularity, and the compression rate. In contrast, DSignSGD converges even under heavy-tailed noise with standard learning rate schedules. Together, these findings offer both new theoretical insights and perspectives, and practical guidance.

## 1. Introduction

Distributed stochastic gradient methods are the workhorse of modern large-scale learning. In practice, their behavior is shaped by three effects that interact nontrivially:

1. **Batch noise.** Stochastic gradient methods rely on minibatches to reduce computational cost, but this introduces uncertainty in the gradient estimates. In practice, this noise may not only be non-vanishing but can exhibit complex, heavy-tailed behavior (Simsekli et al., 2019). Such noise has a profound impact on convergence rates, stability, and generalization, especially in nonconvex landscapes.

2. **Communication compression.** In distributed systems, communicating full-precision gradients is often prohibitively expensive. To alleviate this, gradient compression techniques such as sparsification, quantization, and sign-based schemes are commonly used. While these methods reduce communication overhead, they alter the optimization dynamics by introducing bias and additional variance (Alistarh et al., 2017). Understanding the trade-off between efficiency and convergence guarantees under compression remains a central question.

3. **Adaptivity.** Many successful optimizers in deep learning, such as Adam, AdaGrad, or SignSGD, incorporate some form of normalization or adaptivity in their update rules. Adaptivity has been empirically shown to mitigate the detrimental effects of noise and ill-conditioning (Safaryan & Richtárik, 2021; Staib et al., 2019), yet a rigorous understanding of *why* adaptivity helps in distributed and noisy scenarios is still incomplete. In particular, the interaction between adaptivity and the statistical properties of the gradient noise is far from fully understood.

Despite substantial work studying each component separately, their *joint* interplay remains underexplored, especially under realistic assumptions on the loss landscape.

[†]Eduard Gorbunov, Aurelien Lucchi, and Antonio Orvieto share equal senior supervision. [1]University of Basel, Basel, Switzerland [2]University of Oslo, Oslo, Norway [3]Max Planck Institute for Intelligent Systems, Germany [4]ELLIS Institute Tübingen, Germany [5]Tübingen AI Center, Germany [6]MBZUAI, Abu Dhabi, United Arab Emirates. Correspondence to: Enea Monzio Compagnoni <enea.monziocompagnoni@unibas.ch>.

*Proceedings of the 43rd International Conference on Machine Learning*, Seoul, South Korea. PMLR 306, 2026. Copyright 2026 by the author(s).

Most theoretical results rely on *L-smoothness*, i.e., the assumption that the gradient of the objective function is globally Lipschitz continuous (Bubeck, 2015). While this simplifies analysis, it fails to capture the complexities of practical problems, including those encountered in nonconvex optimization for DL. In contrast, $(L_0, L_1)$-*smoothness* allows the norm of the Hessian of the loss to grow at most affinely with its gradient norm: This relaxes the aforementioned regularity condition (Zhang et al., 2020b) and is a much more realistic alternative. Similarly, while most of the literature relies on the assumption that the gradient noise is bounded or has bounded variance, more realistic models, such as affine variance and heavy-tailed noise, are increasingly adopted in recent works.

In this paper, SDEs are *models* of discrete optimizers: they are not intended to reproduce the iterate-by-iterate trajectory, but rather to capture selected properties of the dynamics. Since there is no "universally correct" SDE, we focus on one key property, the learning-rate restrictions that ensure stability for distributed stochastic optimizers under $(L_0, L_1)$-smoothness, when there may exist *no single constant stepsize* that is stable uniformly over all initializations.

**On the limitations of classic models from the literature.** Following the standard approach, we first considered *first-order* SDEs, which are naturally well-suited to handle a broad range of noise models, e.g., Gaussian, affine, or even heavy-tailed, and historically led to new conceptual (Su et al., 2014) and practical (Jastrzebski et al., 2018) insights. However, these models can be misleading: As we prove in Sec. 4.1, they do not prescribe any learning rate constraints and incorrectly suggest that constant-stepsize SGD converges unconditionally. A natural next step is to move to the *second-order* model from the literature (Li et al., 2017). As we show in Sec. 4.2, this is even more problematic: it additionally predicts *accelerated convergence* at large stepsizes, where SGD, in fact, diverges. Beyond this, neither of these models captures the more subtle mechanisms that arise under $(L_0, L_1)$-smoothness, where *no universal stepsize can guarantee stability uniformly across initializations*.

**Our approach: stability-corrected SDEs.** To address this, we derive *stability-faithful* SDE models by modifying the drift to carefully include curvature-dependent information. The resulting models are *first-order weak approximations* which correctly encode the stability thresholds and align closely with the dynamics of their respective optimizers.

We develop a comprehensive analysis of DCSGD and DSignSGD under $(L_0, L_1)$-smoothness with flexible gradient noise assumptions encompassing affine variance and heavy-tailed noise. In settings already examined in the literature, such as $(L_0, L_1)$-smoothness with affine variance,[1]

---

[1] This setting is studied for Normalized SGD and AdaGrad (Faw et al., 2023; Chen et al., 2023), **and not** for DCSGD or DSignSGD.

our results are consistent with established findings. In previously unexplored regimes, where $(L_0, L_1)$-smoothness, affine variance, heavy-tailed noise, and gradient compression are brought together under a unified framework, our analysis provides novel insights that advance the understanding of the interaction of these factors.

**Contributions.**

1. **Conceptual: Limitations of models from the literature.** We identify concrete ways in which classical first- and second-order SDE models can fail to capture the correct discrete-time stepsize/stability behavior;

2. **Technical: Stability-faithful continuous-time models under $(L_0, L_1)$-smoothness.** We *formally* derive first-order SDE models that correctly capture the learning rate restrictions and stability thresholds *also* under $(L_0, L_1)$-smoothness, which cannot be done by classic SDEs;

3. **Technical: Unified SDE analysis of compression and nonstandard noise.** We prove convergence bounds for the models of DCSGD and DSignSGD under $(L_0, L_1)$-smoothness and batch noise assumptions more general than those commonly used in the literature, namely, affine-variance noise for DCSGD and heavy-tailed noise for DSignSGD;

4. **Practical: Normalization strength for DCSGD.** We demonstrate that the degree of normalization required for DCSGD to converge is precisely determined by the interplay between the compression rate, the structure of gradient noise, and the smoothness constants of the loss;

5. **Practical: Robustness of DSignSGD under heavy tails.** We show that an *adaptive* method such as DSignSGD converges even under heavy-tailed noise with standard assumptions on the learning rate scheduler.

## 2. Related work

**SDE Approximations in Optimization.** Continuous-time models in the form of differential equations are a well-established tool to study discrete-time optimizers, e.g. (Helmke & Moore, 1994; Kushner & Yin, 2003). It was (Li et al., 2017) that first introduced a *rigorous* theoretical framework to derive SDEs that faithfully model the stochastic behavior intrinsic to optimization algorithms widely employed in machine learning. Since then, such SDE-based formulations have been applied across several domains, including *stochastic optimal control* for tuning stepsizes (Li et al., 2017; 2019) and batch sizes (Zhao et al., 2022). Notably, SDEs have been instrumental in analyzing *convergence bounds* and *stationary distributions* (Compagnoni et al., 2023; 2024; 2025b), *scaling laws* (Jastrzebski et al., 2018; Compagnoni et al., 2025a;b; 2026), *implicit regularization* effects (Smith et al., 2021; Compagnoni et al., 2023), and *implicit preconditioning* (Xiao et al., 2025).

*Table 1.* Comparison of existing convergence results for stochastic methods applied to $(L_0, L_1)$-smooth problems. All results are derived for non-convex problems, and the bounds are given in expectation unless stated otherwise. All works assume bounded noise or bounded variance unless stated otherwise. Abbreviations: "HT" = heavy-tailed noise, "Affine var." = affine variance.

| Reference | Dynamics | Noise | | $(L_0, L_1)$-smooth | Compression |
| | | HT | Affine var. | | |
|---|---|---|---|---|---|
| (Zhang et al., 2020b;a) (Zhao et al., 2021) (Crawshaw et al., 2022) (Koloskova et al., 2023) (Li et al., 2023a)[1] [2] (Hübler et al., 2024) (Li et al., 2023b)[1] [3] (Gaash et al., 2025)[1] [3] | Discrete | ✗ | ✗ | ✓ | ✗ |
| (Faw et al., 2023)[1] [2] (Wang et al., 2023)[1] [2] (Chen et al., 2023) | Discrete | ✗ | ✓ | ✓ | ✗ |
| (Khirirat et al., 2024) | Discrete | ✗ | ✗ | ✓ | ✓ |
| (Chezhegov et al., 2025)[1] [3] | Discrete | ✓ | ✗ | ✓ | ✗ |
| (Compagnoni et al., 2025a) | Continuous | ✓ | ✗ | ✗ | ✓ |
| **This work** | Continuous | ✓ | ✓ | ✓ | ✓ |

[1] High-probability convergence analysis.
[2] Convergence bounds have inverse-power dependence on the failure probability.
[3] Derived for convex problems.

**Why continuous-time models are useful.** As mentioned above, continuous-time modeling via ODEs/SDEs is a classical tool in optimization and has seen significant growth in recent years (Li et al., 2017): The main advantage is getting access to sophisticated tools such as Itô calculus, which turns SDEs for the iterates into SDEs for the loss, enabling Lyapunov-style stability arguments to derive convergence bounds. To the best of our knowledge, no prior work has used SDEs to analyze optimizers under $(L_0, L_1)$-smoothness. This represents a significant gap, as this assumption has emerged as an alternative to $L$-smoothness to model nonconvex landscapes.

**A note on modeling.** Before proceeding, we clarify what we mean by *model*: It is a deliberately simplified mathematical surrogate tailored to a specific question; it is not expected to be universally accurate, but rather predictive within the regime relevant to that question. An example comes from physics: General Relativity describes macroscopic gravity, but it is not a quantum theory. In contrast, quantum mechanics excels in the microscopic regime, yet does not describe spacetime geometry. Both are valuable, and their limitations outside their domains are a reminder that the "right" model depends on the phenomenon of interest.

$(L_0, L_1)$**-smoothness, normalization, and compression.** The $(L_0, L_1)$-smoothness condition (Zhang et al., 2020b) relaxes global $L$-smoothness and has motivated analyses of first-order methods beyond bounded-variance settings,

e.g., for Normalized SGD/AdaGrad/Adam/SignSGD variants (Chen et al., 2023; Faw et al., 2023; Wang et al., 2023; Li et al., 2023b; Crawshaw et al., 2022). For compressed communication under relaxed smoothness, Khirirat et al. (2024) provides a representative analysis, but existing results do not offer a unified treatment that simultaneously captures compression together with affine-variance growth and heavy-tailed regimes under $(L_0, L_1)$-smoothness. Table 1 summarizes the closest guarantees and clarifies our positioning. Finally, we refer the interested reader to Appendix A for an extended discussion on the related works.

**Positioning.** We identify a modeling gap relevant to step-size stability: classic first- and second-order modified equations can contradict discrete-time stability behavior, especially under $(L_0, L_1)$-smoothness. We address this by deriving stability-corrected first-order SDEs and using them to provide, to the best of our knowledge, the first unified SDE-based analysis that is both stability-faithful under $(L_0, L_1)$-smoothness and suited to analyze the joint effect of distributed compression with general noise regimes, including affine-variance growth and heavy tails.

## 3. Preliminaries

**Distributed Setup.** Let us consider the problem of minimizing an objective function expressed as an average of $N$ functions: $\min_{x \in \mathbb{R}^d} \left[ f(x) := \frac{1}{N} \sum_{j=1}^{N} f_j(x) \right]$, where each $f_j : \mathbb{R}^d \to \mathbb{R}$ is lower bounded and twice continuously dif-

ferentiable, and represents the loss over the local data of the $i$-th client. In our stochastic setup, each client only has access to gradient estimates: let $n_i$ be the number of datapoints accessible to client $i$; at a given $x \in \mathbb{R}^d$, client $i$ estimates $\nabla f(x)$ using a batch of data $\gamma_i \subseteq \{1, \ldots, n_i\}$, sampled uniformly with replacement and uncorrelated from the previously sampled batches. Under homogeneous data/client assumption, $\nabla f_{i,\gamma_i}(x)$ can be modeled as a perturbation of the global gradient: $\nabla f_{i,\gamma_i}(x) = \nabla f(x) + Z_i(x)^2$. Finally, $S_0 := f(X_0) - f(X_*)$.

**Noise assumptions.** We assume the sampling process and client configurations are such that, for all $x \in \mathbb{R}^d$ and each client pair $(i, j)$ with $i \neq j$, $Z_i(x)$ is independent of $Z_j(x)$. Regarding the noise structure, we always assume that at each $x \in \mathbb{R}^d$, $Z_i(x)$ is absolutely continuous and has a coordinate-wise symmetric distribution. For context, we highlight that numerous works in the literature assume a much more restrictive assumption, e.g. that $Z_i(x)$ are Gaussian (Ahn et al., 2012; Chen et al., 2014; Mandt et al., 2016; 2017; Zhu et al., 2019; Wu et al., 2020; Xie et al., 2021), and (Li et al., 2017; Mertikopoulos & Staudigl, 2018; Raginsky & Bouvrie, 2012; Zhu et al., 2019; Mandt et al., 2016; Ahn et al., 2012; Jastrzebski et al., 2018) even assume the covariance matrix of the batch noise to be constant: we refer the reader to (Jastrzebski et al., 2018) for the intuition behind this modeling choices. If $Z_i(x) \in L^1(\mathbb{R}^d)$, we assume $\mathbb{E}[Z_i(x)] = 0$; if $Z_i(x) \in L^2(\mathbb{R}^d)$, we denote $\Sigma_i(x) := \text{Cov}(Z_i(x))$, where $L^p(\mathbb{R}^d)$ denotes the space of random variables with finite $p$-th moment.

Next, we define our two structural assumptions. The first one strictly concerns the global landscape; the second concerns how global landscape features affect the noise distribution of each client.

**Definition 3.1.** $f$ is $(L_0, L_1)$-smooth $(L_0, L_1 \geq 0)$ if, $\forall x \in \mathbb{R}^d$, $\left\|\nabla^2 f(x)\right\|_2 \leq L_0 + L_1\|\nabla f(x)\|_2$.

**Definition 3.2** (Extension of the assumptions from (Schmidt & Le Roux, 2013))**.** The gradient noise for client $i$ has *affine* $(\sigma_{0,i}^2, \sigma_{1,i}^2)$-variance if $\|\Sigma_i(x)\|_\infty \leq \sigma_{0,i}^2 + \sigma_{1,i}^2\|\nabla f(x)\|_2^2$. If $\sigma_{1,i} = 0$, the noise has bounded variance.

Finally, we define which compressors we use to reduce the communication costs of gradients.

**Definition 3.3.** An unbiased compressor is a stochastic map $\mathcal{C}_\xi : \mathbb{R}^d \to \mathbb{R}^d$ such that $(a)$ $\mathbb{E}_\xi[\mathcal{C}_\xi(x)] = x$ and $(b)$ $\mathbb{E}_\xi\left[\|\mathcal{C}_\xi(x) - x\|_2^2\right] \leq \omega\|x\|_2^2$ for a compression rate $\omega \geq 0$.

**SDE approximations.** The following definition presents the most commonly used notion that formalizes how an SDE can be a "reliable surrogate" to model an optimizer. It is drawn from the field of numerical analysis of SDEs (see (Milshtein, 1986)), and it quantifies the disparity between the discrete and the continuous processes.

---

[2] Our SDE framework can in principle be extended to heterogeneous objectives.

**Definition 3.4.** A continuous-time stochastic process $(X_t)_{t \in [0,T]}$ is an $\alpha$-order weak approximation of a discrete stochastic process $(x_k)_{k=0}^{\lfloor T/\eta \rfloor}$ if for every polynomial growth function $g$, $\exists C > 0$, independent of $\eta$, such that $\max_{k=0,\ldots,\lfloor T/\eta \rfloor} |\mathbb{E}g(x_k) - \mathbb{E}g(X_{k\eta})| \leq C\eta^\alpha$.

We will often refer to 1-order and 2-order weak approximations as *first-* and *second*-order SDEs. Here, "*first*-order" refers to the order of the weak-approximation error in $\eta$, not to the absence of Hessian terms in the drift.

**Scope of SDE guarantees and results.** As is standard in the literature using SDEs for optimization, e.g., (Li et al., 2017; 2019; Luo et al., 2024; Orvieto & Lucchi, 2019b), theoretical guarantees are stated for the *continuous-time models* rather than for the discrete optimizers. The link to the underlying algorithm is provided by weak-approximation (or shadowing) results, which quantify how closely the statistics of the discrete iterates and of the SDE match over finite horizons, with an error controlled by the stepsize (cf. Def. 3.4). In addition, several recent works empirically validate that such SDEs do track the respective optimizers on modern DNNs and datasets, supporting their use as models (Compagnoni et al., 2025a;b; Marshall et al., 2025).

**Example: classical vs. new SDEs for SGD.** Thm 1 of (Li et al., 2017) originally formally derived the first- and second-order SDEs for single-node SGD with stepsize $\eta$. As we denote the covariance batch noise with $\Sigma(x) = \frac{1}{n}\sum_{i=1}^n (\nabla f(x) - \nabla f_i(x))(\nabla f(x) - \nabla f_i(x))^T$, the *first*-order one reads

$$dX_t = -\nabla f(X_t)dt + \sqrt{\eta}\sqrt{\Sigma(X_t)}dW_t, \quad (1)$$

while the *2nd*-order SDE adds a curvature term to the drift,

$$dX_t = -\nabla f(X_t)dt - \frac{\eta}{2}\nabla^2 f(X_t)\nabla f(X_t)dt + \sqrt{\eta}\sqrt{\Sigma(X_t)}dW_t. \quad (2)$$

In this work, we show that both Eq. 1 and Eq. 2 miss discrete-time stepsize stability restrictions, especially under $(L_0, L_1)$-smoothness. We therefore introduce a new *first*-order SDE obtained by modifying the drift term:

$$dX_t = -\nabla f(X_t)dt + \frac{\eta}{2}\nabla^2 f(X_t)\nabla f(X_t)dt + \sqrt{\eta}\sqrt{\Sigma(X_t)}dW_t. \quad (3)$$

Theorem C.5 formally proves that Eq. 3 is a *first*-order weak approximation for SGD in the sense of Def. 3.4, so the modification is not merely heuristic.

**Algorithms and their SDEs.** For a constant stepsize $\eta$ and scheduler $\eta_k$, we study DCSGD defined as

$$x_{k+1} = x_k - \frac{\eta\eta_k}{N}\sum_{i=1}^N \mathcal{C}_{\xi_i}(\nabla f_{i,\gamma_i}(x_k)), \quad (4)$$

with unbiased compressors $\mathcal{C}_{\xi_i}$ with SDE models in Eq. 98–117 and DSignSGD defined as

$$x_{k+1} = x_k - \frac{\eta\eta_k}{N}\sum_{i=1}^N \text{sign}(\nabla f_{i,\gamma_i}(x_k)), \quad (5)$$

with SDEs in Eq. 137–145.

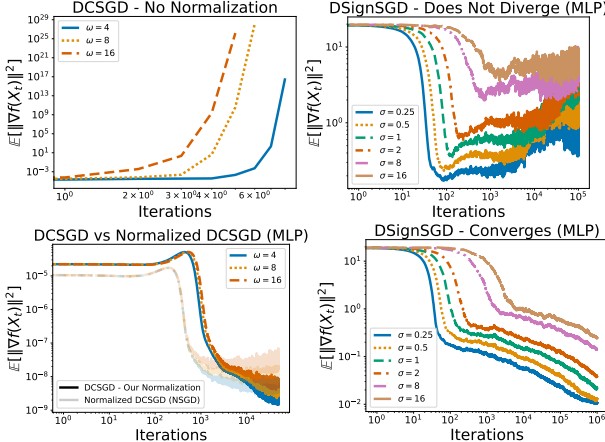

*Figure 1.* **Sanity checks for our stability prescriptions on a noisy, compressed MLP training. Left (DCSGD + unbiased sparsification).** We train an MLP with $N{=}8$ clients, inject additive Gaussian gradient noise with affine variance $Z_t \sim \mathcal{N}(0, \sigma^2 \|g_t\|_2^2 I)$, and then apply unbiased random sparsification at compression levels $\omega$ (larger $\omega$ means more aggressive sparsification). Without any scheduler/normalization, DCSGD becomes unstable and the divergence worsens as $\omega$ increases (top-left). Using the adaptive normalization suggested by Thm. 4.2 (Eq. 15) stabilizes training and yields convergence for all $\omega$ (bottom-left); We also report a baseline that applies plain Normalized SGD under the same noise and compression, which exhibits a (here less stable) profile. **Right (DSignSGD under heavy tails).** We inject Student's $t$ gradient noise with $\nu{=}1$ for different scale values $\sigma$. With a constant step-size, DSignSGD remains stable but does not converge (top-right), whereas the diminishing schedule prescribed by Thm. 4.3 (here $\eta_k = 1/\sqrt{k+1}$) yields convergence across noise scales (bottom-right). See Appendix E for full implementation details.

## 4. Theoretical Results

**Step-size convention.** Recall that, in the continuous-time setup, the dynamics of the iterates is modeled by a stochastic process $X_t$ solution to an SDE model. To separate *adaptivity* from *scheduling*, we *parameterize* the stepsize as $\eta\eta_t$, where $\eta > 0$ is a base scale and $(\eta_t)_{t\geq 0}$ is a deterministic scheduler. We assume the standard Robbins-Monro conditions: defining $\phi_t^{(i)} := \int_0^t (\eta_s)^i ds$, we require $\phi_t^{(1)} \to \infty$ and $\phi_t^{(2)}/\phi_t^{(1)} \to 0$ as $t \to \infty$. A typical choice is $\eta_t = (1+t)^{-a}$ for $a \in (0,1)$ (the boundary cases $a \in \{\frac{1}{2}, 1\}$ are also admissible with minor modifications).

**Overview** Our insights concern the conditions on the learning rate $\eta\eta_t$ for convergence, where $\eta_t$ is a predetermined scheduler. We aim to determine how factors such as compression, noise structure, and adaptivity influence the level of normalization required to guarantee convergence. First, we show how first- and second-order continuous-time models from the literature lead to misleading conclusions, as they fail to capture the stability thresholds of the learning rate of GD. Then, we justify the derivation of new models that capture this aspect of the dynamics. Finally, we present Thm. 4.2 and Thm. 4.3, which are derived under these new formulations and empirically validated in Fig. 1. These ex-

periments should be read as **mechanism validation** rather than competitive benchmarking: their purpose is to test whether the instability and stabilization patterns predicted by the SDE analysis appear in controlled neural-network settings. To further check that these qualitative mechanisms persist beyond the MLP setup, Appendix E.4 reports additional sanity checks on ResNet-18 and ViT models trained on CIFAR-10 in a distributed setting with $N = 8$ clients.

### 4.1. On the Failure of Classic First-Order SDE Models

Consistent with the literature, we start our analysis and derive a convergence bound for DCSGD from its *first*-order SDE: On the one hand, this result is very insightful and certainly captures important aspects of the dynamics. On the other hand, we quickly figure out its limitations as it fails to capture the fact that Gradient Descent on an $L$-smooth loss only converges if $\eta\eta_t < \frac{2}{L}$.

**Theorem 4.1.** *(DCSGD, unbiased compression, affine variance) Let $f$ be $(L_0, L_1)$-smooth, and each client has $(\sigma_{0,i}^2, \sigma_{1,i}^2)$-variance. Define $\overline{\sigma_0^2} := \frac{1}{N}\sum_{i=1}^N \sigma_{0,i}^2$, $\overline{\sigma_1^2} := \frac{1}{N}\sum_{i=1}^N \sigma_{1,i}^2$, $\overline{\sigma_0^2\omega} := \frac{1}{N}\sum_{i=1}^N \sigma_{i,0}^2\omega_i$, and $\overline{\sigma_1^2\omega} := \frac{1}{N}\sum_{i=1}^N \sigma_{i,1}^2\omega_i$. For $\epsilon \in (0,1)$, assume $\eta\eta_t$ is smaller than*

$$\frac{2\epsilon}{(L_0 + L_1 \mathbb{E}[\|\nabla f(X_t)\|_2]) \frac{\overline{\omega} + d(\overline{\sigma_1^2\omega} + \overline{\sigma_1^2})}{N} + \frac{L_1 d(\overline{\sigma_0^2} + \overline{\sigma_0^2\omega})}{N}}. \quad (6)$$

*Then, for a random time $\hat{t}$ (indep. of $X_t$) with distribution $\frac{\eta_t}{\phi_t^{(1)}}$, we have that $\mathbb{E}\left[\|\nabla f(X_{\hat{t}})\|_2^2\right]$ is smaller than*

$$\frac{S_0}{(1-\epsilon)\phi_t^{(1)}} + \phi_t^{(2)} \frac{L_0(\overline{\omega} + d(\overline{\sigma_1^2\omega} + \overline{\sigma_1^2})) + L_1 d(\overline{\sigma_0^2} + \overline{\sigma_0^2\omega})}{2N(1-\epsilon)\phi_t^{(1)}} \xrightarrow{t \to \infty} 0.$$

**Takeaways.** This result highlights the role of the regularity of the loss landscape and its interaction with both gradient noise and compression. Eq. 6 makes explicit how stability tightens as the system becomes noisier, higher-dimensional, and more aggressively compressed. A way to read it is: **(i) More clients help:** increasing $N$ relaxes the constraint by countering noise and compression effects; **(ii) Dimension hurts:** larger $d$ tightens the condition, especially under affine variance ($\overline{\sigma_1^2} > 0$); **(iii) Compression hurts:** $\overline{\omega} > 0$ behaves as an additional distortion term and tightens the stability region; **(iv) $(L_0, L_1)$ matters:** the constraint couples $L_1$ with noise and compression; when $L_1 > 0$, stability becomes more sensitive to the local geometry through $\mathbb{E}\|\nabla f(X_t)\|_2$.

At the same time, the bound reveals the core **modeling limitation**: in the noiseless, uncompressed, $L$-smooth case ($\sigma_{0,i} = \sigma_{1,i} = \omega_i = L_1 = 0$), the denominator in Eq. 6 vanishes and the condition does not restrict $\eta\eta_t$, contradicting the basic discrete-time requirement $\eta\eta_t < 2/L_0$ for

GD/SGD stability (Ahn et al., 2022). This motivates going beyond classical first-order surrogates.

Similarly, in Theorem D.9, we leverage the first-order SDE of DSignSGD to derive the convergence bound of DSignSGD. While it recovers the results from (Compagnoni et al., 2025a) when $L_1 = \sigma_1 = 0$, it also predicts no restrictions on the learning rate in the noiseless scenario.

Finally, we highlight that this bound is not implementable in practice, as constants such as $L_0$, $L_1$, $\sigma_{0,i}$, $\sigma_{1,i}$, and even $\mathbb{E}\left[\|\nabla f(X_t)\|_2\right]$ are not known a priori. While this might look like a limitation, **this is common** in the literature (Gorbunov et al., 2025): As with classical $\eta < 2/L$-type conditions, only identify stability regions and show how different factors driving the dynamics influence it.

### 4.2. Second-Order Models Fail: We Need New Models

For didactic reasons, we now showcase a clear example of how first- and second-order models for GD fail at capturing its learning-rate restrictions. To keep the discussion transparent, we use the noiseless single-node case; Similar issues arise in the stochastic case and motivate our new SDE: We formalize this in Sec. C and Thm. C.5.

**A quadratic sanity check.** Consider $f(x) = \frac{\lambda x^2}{2}$ with $\lambda > 0$ in one dimension. Discrete GD with constant stepsize $\eta$ satisfies $x_{k+1} = (1 - \eta\lambda)x_k$: it is stable iff $\eta < 2/\lambda$.

The first-order ODE $dX_t = -\nabla f(X_t)dt = -\lambda X_t dt$ yields $f(X_t) = f(X_0)e^{-2\lambda t}$, i.e., convergence for *all* $\eta$: it does not even "see" $\eta$. One might expect that moving from the first-order to the *second-order* ODE might fix the issue. Surprisingly, this is not the case. The classical second-order ODE from the literature,

$$dX_t = -\nabla f(X_t)dt - \frac{\eta}{2}\nabla^2 f(X_t)\nabla f(X_t)dt, \quad (7)$$

implies $f(X_t) = f(X_0)e^{-2\lambda(1+\lambda\eta/2)t}$: it again predicts unconditional convergence and even suggests that larger $\eta$ accelerates convergence, where instead GD would diverge if the learning rate were large enough.

**A quartic sanity check: no universal stepsize under $(L_0, L_1)$-smoothness.** The quadratic example above highlights that classical ODE/SDE surrogates can miss the *global* $L$-smooth stepsize restriction. Under $(L_0, L_1)$-smoothness, an even sharper pathology occurs: there may be *no single constant stepsize* that guarantees stability *uniformly over initializations*. A minimal example is the one-dimensional quartic $f(x) = \frac{x^4}{4}$, which is not $L$-smooth. In this case, a GD step with constant stepsize $\eta$ reads

$$x_{k+1} = x_k - \eta\nabla f(x_k) = x_k - \eta x_k^3 = x_k(1 - \eta x_k^2).$$

Hence, one needs $\eta < 2/x_k^2$ to prevent instantaneous expansion, meaning that stability depends on the *current scale* of the iterate. In particular, there is *no universal constant* $\eta > 0$ that stabilizes GD for all initialization: for any $\eta$, choosing $|x_0| > \sqrt{2/\eta}$ makes GD expand at the first step.

The first-order ODE is $dX_t = -X_t^3 dt$, which yields an explicit globally convergent trajectory $X_t = (X_0^{-2} + 2t)^{-1/2}$ and thus predicts convergence *independently* of $\eta$. The classical second-order ODE, $dX_t = -X_t^3 dt - \frac{3\eta}{2}X_t^5 dt$, makes the loss decrease even *faster* as $\eta$ grows (the additional term is purely damping), again contradicting the discrete-time instability for large initialization.

**Takeaway:** The modeling failure of classic ODE models is twofold: **(i)** Failure to restrict the learning rate to ensure stability; **(ii)** Failure to reproduce divergence at large learning rates. While the first is non-contestable, the second might feel a bit stretched, as ODEs are more reliable when the learning rate is small, and one could argue we pushed the use of ODEs outside of their "validity region". However, our analysis of the quartic function shows that even for an infinitesimally small learning rate, both the ODEs of GD predict convergence independently of the learning rate, while GD actually has a restriction on the learning rate depending on the initialization. Therefore, neither first- nor second-order ODEs of GD correctly model the dynamics of GD on $(L_0, L_1)$-smooth functions even at infinitesimally small learning rates: More details in Sec. C.3.2.

**Comparison With Discrete-Time Analysis** Here, we take a step back and closely compare the dynamics of the loss function in discrete-time with that in continuous time as prescribed by the ODEs of GD. Using a second-order Taylor expansion around $x_k$ along the GD step gives

$$\frac{f(x_{k+1}) - f(x_k)}{\eta} = -\|\nabla f(x_k)\|^2 + O_{x_k}(\eta^2) \quad (8)$$
$$+ \tfrac{\eta}{2}\nabla f(x_k)^\top \nabla^2 f(x_k)\nabla f(x_k).$$

Here, $O_{x_k}(\eta^2)$ denotes a term bounded by $C(x_k)\eta^2$ for small enough $\eta$, with $C(x_k)$ independent of $\eta$. However, the first-order ODE of GD implies that

$$df(X_t) = -\|\nabla f(X_t)\|_2^2 dt. \quad (9)$$

We notice that this continuous-time loss drift is completely missing the $O(\eta)$ correction highlighted in purple color in Eq. 8. The natural step is to use the 2nd-order ODE, which implies that

$$df(X_t) = -\|\nabla f(X_t)\|_2^2 dt \quad (10)$$
$$- \tfrac{\eta}{2}\nabla f(X_t)^\top \nabla^2 f(X_t)\nabla f(X_t)dt.$$

While this ODE of the loss *does* incorporate some second-order information highlighted in purple color, we notice that its sign is flipped with respect to the discrete-time loss drift in Eq. 8. This flipped sign is exactly the factor responsible for the failures of this second-order ODE.

*Table 2.* Comparison of the learning rate constraints on $\eta\eta_t$ derived from classic SDEs vs. our SDEs. Each row corresponds to the theorem pairs: DCSGD (Thm. 4.1 vs. Thm. 4.2) and DSignSGD (Thm. D.9 vs. Thm. 4.3). Below, $G :=$ $(L_0 + L_1\mathbb{E}[\|\nabla f(X_t)\|_2])$.

| Setting | Constraint (Classic vs. Ours) |
|---|---|
| DCSGD | Classic: $\dfrac{2\epsilon}{G\frac{\overline{\omega}+d(\overline{\sigma_1^2\omega}+\overline{\sigma_1^2})}{N} + \frac{L_1 d\left(\overline{\sigma_0^2}+\overline{\sigma_0^2\omega}\right)}{N}}$ 
 Ours: $\dfrac{2\epsilon}{G\left(1 + \frac{\overline{\omega}+d(\overline{\sigma_1^2\omega}+\overline{\sigma_1^2})}{N}\right) + \frac{L_1 d\left(\overline{\sigma_0^2}+\overline{\sigma_0^2\omega}\right)}{N}}$ |
| DSignSGD | Classic: $\frac{\ell_\nu}{K}$ s.t. $K = \dfrac{L_1 d\sigma_{\mathcal{H},1}}{2N}$ 
 Ours: $\frac{\ell_\nu}{K}$ s.t. $K = \dfrac{L_1 d\sigma_{\mathcal{H},1}}{2N} + \sqrt{d}\left(L_0 + L_1\right)M_\nu$ |

**Deriving a New Model: An Ansatz Approach.** Therefore, we understand that choosing the right model for the iterates is critical to capture the aspects of the dynamics under analysis. Inspired by a classic approach in physics, we propose an *ansatz* for an ODE of the iterates of GD and seek one that models the loss dynamics more closely, which is exactly the information needed to recover the correct stepsize stability threshold. For $\alpha \in \mathbb{R}$, we propose:

$$dX_t = -\nabla f(X_t)dt + \alpha\eta\nabla^2 f(X_t)\nabla f(X_t)dt, \quad (11)$$

which implies that the loss dynamics is driven by

$$df(X_t) = -\|\nabla f(X_t)\|_2^2 dt \quad\quad (12)$$
$$+ \alpha\eta\nabla f(X_t)^\top \nabla^2 f(X_t)\nabla f(X_t)dt.$$

Matching the induced loss drift with the discrete-time generator expansion in Eq. 8 (i.e., comparing $\frac{d}{dt}f(X_t)$ with $\frac{f(x_{k+1})-f(x_k)}{\eta}$ up to $O(\eta)$) suggests choosing $\alpha = \frac{1}{2}$, which is the unique choice within the ansatz family that recovers the exact quadratic stability threshold. Therefore, we obtain

$$dX_t = -\nabla f(X_t)dt + \frac{\eta}{2}\nabla^2 f(X_t)\nabla f(X_t)dt, \quad (13)$$

is our candidate ODE for GD: We formalize this in Section C, and Theorem C.5 extends this formalization to the stochastic setting. Importantly, in the **quadratic case** studied above, it implies that

$$f(X_t) = f(X_0)e^{-2\lambda\left(1-\frac{\lambda\eta}{2}\right)t}, \quad\quad (14)$$

which, consistently with GD, converges only if $\eta < \frac{2}{\lambda}$. In the **quartic case**, this ODE implies that

$$df(X_t) = \left(-X_t^6 + \frac{3\eta}{2}X_t^8\right)dt,$$

which matches the discrete-time loss expansion

$$\frac{f(x_{k+1}) - f(x_k)}{\eta} = -x_k^6 + \frac{3}{2}\eta x_k^8 + O(\eta^2).$$

Crucially, this model predicts that for sufficiently large $|X_t|$ (roughly when $\eta X_t^2 \gtrsim 1$), the drift becomes *repulsive* and the loss can increase, mirroring the fact that on $(L_0, L_1)$-smooth landscapes admissible stepsizes must shrink with the local gradient scale, rather than being globally constant or deterministic.

**Conclusion:** This analysis suggests that a higher order of a continuous-time model does not necessarily translate into it better modeling the discrete-time dynamics, not even in the simplest cases, and even less in the $(L_0, L_1)$-smoothness setting. In particular, we find that appropriate **first**-order SDEs are more faithful than both the classic first- **and** second-order models when it comes to accurately capturing the stability of the optimizers.

### 4.3. Results Derived via Our SDEs

In this subsection, we report the convergence bounds for newly derived models of DCSGD and DSignSGD. Compared to standard first- and second-order models, our models reveal the interaction between learning rate schedules, loss landscape, batch noise, and compression in a way that is consistent with the discrete dynamics of known cases in the literature. Before presenting these results, Table 2 summarizes how the constraint on $\eta\eta_t$ changes when moving from leveraging the classic SDEs to ours. The orange color indicates terms that *only* appear due to the use of our SDEs.

**Theorem 4.2.** *(DCSGD, unbiased compression, affine variance) Let $f$ be $(L_0, L_1)$-smooth, and each client have $(\sigma_{0,i}^2, \sigma_{1,i}^2)$-variance. Define $\overline{\sigma_0^2} := \frac{1}{N}\sum_{i=1}^N \sigma_{0,i}^2$, $\overline{\sigma_1^2} := \frac{1}{N}\sum_{i=1}^N \sigma_{1,i}^2$, $\overline{\sigma_0^2\omega} := \frac{1}{N}\sum_{i=1}^N \sigma_{i,0}^2\omega_i$, and $\overline{\sigma_1^2\omega} := \frac{1}{N}\sum_{i=1}^N \sigma_{i,1}^2\omega_i$. For $\epsilon \in (0,1)$, assume $\eta\eta_t$ is smaller than*

$$\frac{2\epsilon}{(L_0 + L_1\mathbb{E}[\|\nabla f(X_t)\|_2])\left(1 + \frac{\overline{\omega}+d(\overline{\sigma_1^2\omega}+\overline{\sigma_1^2})}{N}\right) + \frac{L_1 d\left(\overline{\sigma_0^2}+\overline{\sigma_0^2\omega}\right)}{N}}.$$
$$(15)$$

*Then, for a random time $\hat{t}$ (indep. of $X_t$) with distribution $\frac{\eta_t}{\phi_t^{(1)}}$, we have that $\mathbb{E}\left[\|\nabla f(X_{\hat{t}})\|_2^2\right]$ is smaller than*

$$\frac{1}{(1-\epsilon)\phi_t^{(1)}}\left(S_0 + \phi_t^{(2)}\frac{\eta(L_0+L_1)d(\overline{\sigma_0^2}+\overline{\sigma_0^2\omega})}{2N}\right) \overset{t\to\infty}{\to} 0.$$
$$(16)$$

**Takeaways.** The key qualitative difference from Thm. 4.1 is the *baseline term* 1 inside the stability constraint (Eq. 15), which restores the correct noiseless $L$-smooth restriction: setting $\sigma_{0,i} = \sigma_{1,i} = \omega_i = L_1 = 0$ yields $\eta\eta_t < 2/L_0$. More broadly: **(i)** The constraint quantifies how **compression** and **affine variance** amplify the effective "adaptivity pressure": the more aggressive the compression (larger $\overline{\omega}$) and/or the stronger the variance growth ($\overline{\sigma_1^2}$), the more the stepsize must be normalized. **(ii)** Under $(L_0, L_1)$-smoothness, stability is inherently **geometry dependent** (through $\mathbb{E}\|\nabla f(X_t)\|_2$), reflecting that there may be no

universal constant stepsize guaranteeing stability uniformly over the trajectory. **(iii)** Although Eq. 15 is a sufficient condition rather than a closed-form tuning rule, it plays the same conceptual role as the classical $\eta < 2/L$ condition: it delineates a stability region and reveals how the right amount of normalization necessary to ensure convergence is dictated jointly by the compression rate, the variance structure of the noise, and the geometry of the landscape. In this sense, this result offers practitioners concrete guidance on when and how to stabilize DCSGD in challenging regimes. **(iv)** The insights derived are in accordance with some known in the literature: 1. In the noiseless setup $\overline{\sigma_0} = \overline{\sigma_1} = 0$, normalizing the update step naturally emerges as a condition for convergence, in accordance with (Gorbunov et al., 2025); 2. When $L_1 \overline{\sigma_1^2} > 0$, stronger adaptivity is required, in line with insights derived from analyses of Adagrad (Wang et al., 2023) and NSGD (Chen et al., 2023). **(v) Novelty.** To the best of our knowledge, Eq. 15 is the first stability region that simultaneously captures $(L_0, L_1)$-geometry, *unbiased* gradient compression, and *affine-variance* noise growth for DCSGD; Additionally, existing $(L_0, L_1)$ results typically treat either affine-variance without compression or compression under more restrictive noise assumptions (cf. Table 1).

**DSignSGD, structured noise, unbounded expected value.** To provide informative results for the convergence of DSignSGD under heavy-tailed batch noise, we additionally assume structured noise following a student-$t$ distribution: $\nabla f_{\gamma_i}(x) = \nabla f(x) + \sqrt{\Sigma_i} Z_i$ s.t. $Z_i \sim t_\nu(0, I_d)$, $\nu$ are the degrees of freedom, and *scale matrices*[3] $\Sigma_i = \text{diag}(\sigma_{1,i}^2, \cdots, \sigma_{d,i}^2)$. Note that if $\nu = 1$, the *expected value* of $Z_i$ is *unbounded*, thus modeling much more pathological noise than simple affine $(\sigma_0^2, \sigma_1^2)$-variance.

**Theorem 4.3.** *Let $f$ be $(L_0, L_1)$-smooth, $\Sigma_i \leq \sigma_{max,i}^2$, $\sigma_{\mathcal{H},1}$ be the harmonic mean of $\{\sigma_{max,i}\}$, and $K := \left( \frac{L_1 d \sigma_{\mathcal{H},1}}{2N} + \sqrt{d}(L_0 + L_1) M_\nu \right)$. Then, for $\eta \eta_t < \frac{\ell_\nu}{K}$ and random time $\tilde{t}$ (indep. of $X_t$) with distribution $\frac{\eta_t \ell_\nu - \eta_t^2 \eta K}{\phi_t^{(1)} \ell_\nu - \phi_t^{(2)} \eta K}$, we have that $\mathbb{E}\left[ \|\nabla f(X_{\tilde{t}})\|_2^2 \right]$ is smaller than*

$$\frac{\sigma_{\mathcal{H},1}}{\phi_t^{(1)} \ell_\nu - \phi_t^{(2)} \eta K} \left[ S_0 + \phi_t^{(2)} \eta (L_0 + L_1) \left[ \frac{d}{2N} + \frac{M_\nu \sqrt{d}}{\sigma_{\mathcal{H},1}} \right] \right] \tag{17}$$

*and converges to 0 as $t \to \infty$.*

**Takeaways.** Heavier tails (smaller $\nu$) and larger noise scales (larger $\sigma_{max,i}$) tighten the admissible stepsize through $K$. The crucial difference from the first-order SDE analysis (Theorem D.9) is the appearance of an additional baseline term: Even when $\sigma_{max,i} = 0$, the bound still enforces a nontrivial restriction $\eta \eta_t \lesssim 1/\sqrt{d}(L_0 + L_1)$. Finally, unlike DCSGD (see Eq. 15), DSignSGD does *not* require the step-

---

[3] These are *not* covariance matrices, but we use the same notation to facilitate comparability.

size to scale inversely with $\|\nabla f\|$: The sign update already induces an implicit normalization, which is consistent with the stability behavior in Fig. 1.

## 5. Conclusion

**Conceptual contribution: stability-faithful continuous-time models.** We developed continuous-time models for distributed stochastic optimizers under $(L_0, L_1)$-smoothness that are explicitly *stability-faithful*. A key conceptual finding is that, in this regime, standard SDEs used in the literature can be qualitatively misleading: both the first- and second-order SDEs may fail to reproduce discrete-time stepsize/stability restrictions, and can even predict benign or accelerated behavior where the underlying algorithm is unstable. Our viewpoint is to prioritize *stability fidelity*, by modifying the curvature-dependent drift so that the dynamics of our novel models recover the correct stability thresholds.

**Technical contribution: new SDE surrogates and convergence guarantees beyond classical assumptions.** Technically, this paper provides (to the best of our knowledge) the *first* SDE-based treatment of stochastic optimization under $(L_0, L_1)$-smoothness. Using our stability-corrected SDEs, we derive bounds under *unbiased compression* and general noise models that go beyond bounded variance, including *affine-variance growth* and *heavy-tailed* regimes. The resulting stepsize/stability prescriptions make the dependence on geometry, compression strength, dimension, and the noise structure explicit, clarifying precisely which mechanisms shrink the stable region and why classical models can miss this interaction under $(L_0, L_1)$-smoothness.

**Practical implications and outlook.** From a practical standpoint, our analysis yields a concrete stability message: for DCSGD, *noise- and compression-dependent normalization* emerges as the natural mechanism to remain stable, and our novel results quantify how the appropriate normalization level is jointly determined by the compression rate, the noise-growth parameters, and $(L_0, L_1)$-geometry. In contrast, DSignSGD's elementwise normalization makes it robust to heavy-tailed noise, enabling convergence under standard diminishing stepsize schedules even in regimes where the noise can have unbounded expectation. Experiments on DNNs corroborate these qualitative predictions.

More broadly, our contribution is methodological: it shows how to build SDE surrogates that are faithful to the stability restrictions of the discrete optimizer in a regime where classical SDEs can be qualitatively misleading. This matters because SDEs are already used to study convergence bounds, learning-rate schedules, batch-size control, and scaling laws (Orvieto & Lucchi, 2019a; Zhao et al., 2022; Jastrzebski et al., 2018; Compagnoni et al., 2024; 2025a;b). If the continuous-time surrogate predicts convergence while

missing the discrete-time stability region, then prescriptions derived from it may push the actual optimizer outside its stable regime. Our corrected models open the door to revisiting these questions under $(L_0, L_1)$-smoothness with the relevant stability restrictions built in. For instance, batch-size schemes based on classical SDE control problems (Zhao et al., 2022) should incorporate constraints preventing ratios such as $\eta/B(t)$ from leaving the stable regime; learning-rate schedules derived from continuous-time convergence bounds on suboptimality or gradient norm (Orvieto & Lucchi, 2019a; Compagnoni et al., 2024; 2025a;b) should be interpreted together with admissible stepsize regions, since otherwise they may misinform how to set learning-rate schedules; and SDE-based scaling laws, such as the $\eta/B$ linear scaling rule (Jastrzebski et al., 2018) and square-root scaling rules for adaptive methods (Compagnoni et al., 2025b), should be viewed as valid only within the stability limits imposed by the corrected dynamics, for instance in terms of admissible batch-size and stepsize ranges.

Finally, our contribution is intentionally foundational: rather than proposing new optimizers, we provide a unified, stability-faithful SDE framework for understanding how noise, compression, and normalization interact under relaxed smoothness assumptions, and we expect it to support future extensions (e.g., heterogeneous clients, error-feedback and biased compressors) and analogous analyses of modern methods that incorporate implicit second-order structure, such as momentum/NAG and adaptive optimizers (AdamW, RMSprop).

**Limitations.** Our analysis focuses on homogeneous clients, server-aggregated distributed optimization, and unbiased or signed compression. Fully decentralized topologies, heterogeneous client distributions, error-feedback mechanisms, and general biased compressors are outside the scope of the present work. Moreover, our guarantees are stated for the continuous-time surrogate models, and our experiments are intended as mechanism validation rather than competitive benchmarking. We provide a more detailed discussion in App. D.4.

## Acknowledgments

Enea Monzio Compagnoni, Rustem Islamov, and Aurelien Lucchi acknowledge the financial support of the Swiss National Foundation, SNF grant No 207392.

## Impact Statement

This paper develops a theoretical framework for understanding the stability and convergence of distributed stochastic optimization under relaxed smoothness assumptions, non-standard gradient noise, and communication compression. The intended impact is to improve the design and tuning of communication-efficient training methods, which can reduce communication overhead, compute cost, and potentially energy consumption in large-scale learning.

This work is primarily methodological and theoretical: it does not introduce new datasets, collect personal data, or propose a new deployed system. We do not anticipate direct negative societal impacts specific to this contribution beyond those broadly associated with advances in optimization that make training large-scale models more efficient. Such efficiency improvements can enable both beneficial applications (e.g., more accessible training and faster research iteration) and harmful ones (e.g., lowering barriers to training models for misuse). Any downstream deployment should therefore follow established best practices for evaluating safety, privacy, and fairness.

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

*Table 3.* Comparison of existing convergence results for stochastic methods applied to $(L_0, L_1)$-smooth problems. All results are derived for non-convex problems, and the bounds are given in expectation unless stated otherwise. All works assume bounded noise or bounded variance unless stated otherwise. Abbreviations: "HT" = heavy-tailed noise, "Affine var." = affine variance.

| Reference | Dynamics | Noise | | $(L_0, L_1)$-smooth | Compression |
| | | HT | Affine var. | | |
| --- | --- | --- | --- | --- | --- |
| (Zhang et al., 2020b;a) (Zhao et al., 2021) (Crawshaw et al., 2022) (Koloskova et al., 2023) (Li et al., 2023a)[1][2] (Hübler et al., 2024) (Li et al., 2023b)[1][3] (Gaash et al., 2025)[1][3] | Discrete | ✗ | ✗ | ✓ | ✗ |
| (Faw et al., 2023)[1][2] (Wang et al., 2023)[1][2] (Chen et al., 2023) | Discrete | ✗ | ✓ | ✓ | ✗ |
| (Khirirat et al., 2024) | Discrete | ✗ | ✗ | ✓ | ✓ |
| (Chezhegov et al., 2025)[1][3] | Discrete | ✓ | ✗ | ✓ | ✗ |
| (Compagnoni et al., 2025a) | Continuous | ✓ | ✗ | ✗ | ✓ |
| **This work** | Continuous | ✓ | ✓ | ✓ | ✓ |

[1] High-probability convergence analysis.
[2] Convergence bounds have inverse-power dependence on the failure probability.
[3] Derived for convex problems.

## A. Additional Related work

**SDE Approximations in Optimization.** Continuous-time models in the form of differential equations are a well-established tool to study discrete-time optimizers, e.g. (Helmke & Moore, 1994; Kushner & Yin, 2003). Recent works also derived differential equations to model SGD under heavy-tailed batch noise (Simsekli et al., 2019), and (Zhou et al., 2020) derived a Lévy-driven stochastic differential equation to model the non-Gaussianity of the noise. It was (Li et al., 2017) that first introduced a *rigorous* theoretical framework to derive SDEs that faithfully model the stochastic behavior intrinsic to optimization algorithms widely employed in machine learning. Since then, such SDE-based formulations have been applied across several domains, including *stochastic optimal control* for tuning stepsizes (Li et al., 2017; 2019) and batch sizes (Zhao et al., 2022). Notably, SDEs have been instrumental in analyzing *convergence bounds* and *stationary distributions* (Compagnoni et al., 2023; 2024; 2025b), *scaling laws* (Jastrzebski et al., 2018; Compagnoni et al., 2025b;a), *implicit regularization* effects (Smith et al., 2021; Compagnoni et al., 2023), and *implicit preconditioning* (Xiao et al., 2025; Marshall et al., 2025). We refer the interested reader to (Orvieto & Lucchi, 2019b;a) for a didactic introduction to this topic, especially for how Itô calculus is used to derive these results.

We contribute to this line by highlighting a key gap: both the classic *first*-order and the *second-order* SDEs from the literature can yield conclusions that contradict the discrete-time dynamics of SGD. While this is somewhat expected from a first-order model, it is surprising that a higher-order one also fails, possibly even more. While previous studies did derive second-order SDEs for various optimizers (Li et al., 2017; 2019; Luo et al., 2024), they did not exploit them to obtain theoretical insights and thus overlooked these limitations. To remediate this, we derive new *first*-order SDEs whose drift is modified to recover the relevant stability thresholds of discrete optimizers, and we use them to study the joint effect of compression and general noise in distributed learning.

**Interplay of noise, compression, and adaptivity under $(L_0, L_1)$-smoothness.** Previous research has extensively studied the effect of batch noise, compression, and adaptivity on the convergence of optimizers. Batch noise significantly influences stochastic gradient algorithms, affecting their convergence speed and stability (Simsekli et al., 2019; Zhang et al., 2020b; Kunstner et al., 2024; Compagnoni et al., 2025b). Noise characteristics such as heavy-tailed distributions have been shown to profoundly impact the optimization trajectories, necessitating robust algorithmic strategies (Şimşekli et al., 2019; Gorbunov et al., 2021). Compression methods, including unbiased techniques such as sparsification and quantization (Alistarh et al., 2017; Stich et al., 2018; Mishchenko et al., 2024) and biased approaches such as SignSGD (Bernstein et al., 2018; Balles & Hennig, 2018), are critical for reducing communication overhead in distributed training. These compression techniques come with theoretical guarantees under various smoothness assumptions (Alistarh et al., 2017; Gorbunov et al., 2020; Mishchenko et al., 2024; Compagnoni et al., 2025a), and recent results also develop linear-rate or near-optimal behavior under generalized/$(L_0, L_1)$-smoothness (Vankov et al., 2025; Tyurin, 2024). Adaptive methods such as SignSGD normalize gradient elements to cope effectively with large or heavy-tailed gradient noise, thus demonstrating improved empirical robustness (Safaryan & Richtárik, 2021; Compagnoni et al., 2025b;a; Kornilov et al., 2025).

However, most of the works mentioned above rely on restrictive assumptions such as $L$-smoothness, i.e., the $L$-Lipschitz continuity of the gradient. To relax this condition, Zhang et al. (2020b) introduces and empirically validates the $(L_0, L_1)$-*smoothness* assumption, which allows the norm of the Hessian to be bounded by an affine function of the gradient norm, thereby significantly expanding the class of admissible problems. A growing body of work now analyzes (stochastic) *first*-order methods under $(L_0, L_1)$ or more "generalized-smoothness" assumptions, including Clip-SGD and related clipping schemes (Zhang et al., 2020b;c; Koloskova et al., 2023; Reisizadeh et al., 2025; Gorbunov et al., 2025; Vankov et al., 2025; Gaash et al., 2025; Pethick et al., 2025), Normalized SGD and variants with normalization-based schedules (Zhao et al., 2021; Chen et al., 2023; Hübler et al., 2024; Yang et al., 2024), SignSGD (Crawshaw et al., 2022), AdaGrad (Faw et al., 2023; Wang et al., 2023), Adam (Li et al., 2023b; Zhang et al., 2024), and SGD (Li et al., 2023a). Beyond these, there are accelerated and proximal/mirror-descent developments under generalized or $(L_0, L_1)$-smoothness (Tyurin, 2025; Yu et al., 2025a; Tovmasyan et al., 2025; Yu et al., 2025b), nonlinearly preconditioned methods (Oikonomidis et al., 2025), results on escaping saddle points (Cao et al., 2025), zero-/first-order complexity under generalized smoothness (Lobanov & Gasnikov, 2025), and decentralized/federated formulations with generalized smoothness and local steps (Demidovich et al., 2025; Jiang et al., 2025). For compressed communication, Khirirat et al. (2024) proposed and analyzed a momentum-based variant of normalized EF21-SGD (Richtárik et al., 2021) under bounded variance. Additional generalized-smoothness analyses further connect normalization, compression, and relaxed smoothness guarantees (Lobanov et al., 2024; Tyurin, 2024; Yang et al., 2024).

**Positioning.** To the best of our knowledge, there is no unified SDE-based framework that (a) is *stability-faithful* under $(L_0, L_1)$-smoothness and (b) is then used to analyze the joint effect of distributed compression and general noise regimes, including affine variance and heavy tails. This is the gap addressed by the present work (Table 3).

## B. Theoretical Framework

In this section, we introduce the theoretical framework, assumptions, and notations used to formally derive the SDE models used in this paper.

**Definition B.1.** Let $\mathcal{G}$ denote the set of continuous functions $g : \mathbb{R}^d \to \mathbb{R}$ of at most polynomial growth, namely such that there exist positive integers $k_1, k_2 > 0$ such that $|g(x)| < k_1(1 + \|x\|_2^2)^{k_2}$, for all $x \in \mathbb{R}^d$.

To simplify the notation, we will write

$$b(x + \eta) = b_0(x) + \eta b_1(x) + O(\eta^2),$$

whenever there exists $g \in \mathcal{G}$, independent of $\eta$, such that

$$|b(x + \eta) - b_0(x) - \eta b_1(x)| \le g(x)\eta^2.$$

We now introduce the definition of weak approximation, which formalizes in which sense the solution to an SDE, which is a continuous-time random process, models a discrete-time optimizer.

**Definition B.2.** A continuous-time stochastic process $(X_t)_{t \in [0,T]}$ is an $\alpha$-order weak approximation of a discrete stochastic process $(x_k)_{k=0}^{\lfloor T/\eta \rfloor}$ if for every polynomial growth function $g$, there exists a positive constant $C$, independent of $\eta$, such that

$\max_{k=0,\ldots,\lfloor T/\eta \rfloor} |\mathbb{E}g(x_k) - \mathbb{E}g(X_{k\eta})| \leq C\eta^\alpha$. We will often refer to 1-order and 2-order weak approximations as *first-* and *second-*order SDEs.

This framework focuses on approximation in a *weak sense*, meaning in distribution rather than path-wise. Since $\mathcal{G}$ contains all polynomials, all the moments of both processes become closer at a rate of $\eta^\alpha$ and thus their distributions. Thus, while the processes exhibit similar average behavior, their sample paths may differ significantly, justifying the term weak approximation.

The key ingredient for deriving the SDE is given by the following result (see Theorem 1, (Li et al., 2017)), which provides sufficient conditions to get a weak approximation in terms of the single step increments of both $X_t$ and $x_k$. Before stating the theorem, we list the regularity assumption under which we are working.

**Assumptions:** Assume that the following conditions are satisfied:

- $f, f_i \in \mathcal{C}_b^8(\mathbb{R}^d, \mathbb{R})$;

- $f, f_i$ and its partial derivatives up to order 7 belong to $\mathcal{G}$;

- $\nabla f, \nabla f_i$ satisfy the following Lipschitz condition: there exists $L > 0$ such that

$$\|\nabla f(u) - \nabla f(v)\|_2 + \sum_{i=1}^d \|\nabla f_i(u) - \nabla f_i(v)\|_2 \leq L \|u - v\|_2 \, ;$$

- $\nabla f, \nabla f_i$ satisfy the following growth condition: there exists $M > 0$ such that

$$\|\nabla f(x)\|_2 + \sum_{i=1}^n \|\nabla f_i(x)\|_2 \leq M(1 + \|x\|_2).$$

*Remark* B.3. Although these assumptions are very strong, they mainly reflect limitations of the available proof techniques for weak approximation results. In particular, they can be so restrictive that even simple objectives, such as quadratic functions, may fall outside their formal scope. For this reason, we view the weak-approximation theorems as a rigorous guarantee under additional regularity, but not as a prerequisite for the usefulness of the resulting SDE models. In practice, the same weak-approximation heuristics can be applied far beyond the class of functions covered by these assumptions, and extensive empirical evidence indicates that the resulting modified equations closely track the dynamics of the corresponding optimizers across a range of architectures, including ResNets, ViTs, MLPs, and CNNs (Compagnoni et al., 2025b).

**Lemma B.4.** *Let $0 < \eta < 1$. Consider a stochastic process $X_t, t \geq 0$ satisfying the SDE*

$$dX_t = b(X_t)dt + \sqrt{\eta}\sigma(X_t)dW_t, \qquad X_0 = x \tag{18}$$

*where $b, \sigma$ together with their derivatives belong to $\mathcal{G}$. Define the one-step difference $\Delta = X_\eta - x$, and indicate the $i$-th component of $\Delta$ with $\Delta_i$. Then we have*

1. *$\mathbb{E}\Delta_i = b_i\eta + \frac{1}{2}\left[\sum_{j=1}^d b_j \partial_j b_i\right]\eta^2 + O(\eta^3) \qquad \forall i = 1,\ldots,d;$*

2. *$\mathbb{E}\Delta_i\Delta_j = \left[b_i b_j + \sigma\sigma_{ij}^\top\right]\eta^2 + O(\eta^3) \qquad \forall i,j = 1,\ldots,d;$*

3. *$\mathbb{E}\prod_{j=1}^s \Delta_{i_j} = O(\eta^3) \qquad \forall s \geq 3, i_j = 1,\ldots,d.$*

*All functions above are evaluated at $x$.*

**Theorem B.5.** *Let $0 < \eta < 1$, $\tau > 0$ and set $T = \lfloor \tau/\eta \rfloor$. Let Assumption B hold and let $X_t$ be a stochastic process as in Lemma B.4. Define $\bar{\Delta} = x_1 - x$ to be the increment of the discrete-time algorithm, and indicate the $i$-th component of $\bar{\Delta}$ with $\bar{\Delta}_i$. If in addition there exist $K_1, K_2, K_3, K_4 \in \mathcal{G}$ so that*

1. *$|\mathbb{E}\Delta_i - \mathbb{E}\bar{\Delta}_i| \leq K_1(x)\eta^2, \qquad \forall i = 1,\ldots,d;$*

2. $|\mathbb{E}\Delta_i\Delta_j - \mathbb{E}\bar{\Delta}_i\bar{\Delta}_j| \leq K_2(x)\eta^2, \qquad \forall i, j = 1, \ldots, d;$

3. $|\mathbb{E}\prod_{j=1}^s \Delta_{i_j} - \mathbb{E}\prod_{j=1}^s \bar{\Delta}_{i_j}| \leq K_3(x)\eta^2, \qquad \forall s \geq 3, \forall i_j = 1, \ldots, d;$

4. $\mathbb{E}\prod_{j=1}^s |\bar{\Delta}_{i_j}| \leq K_4(x)\eta^2, \qquad \forall i_j = 1, \ldots, d.$

*Then, there exists a constant $C$ so that for all $k = 0, 1, \ldots, N$ we have*

$$|\mathbb{E}g(X_{k\eta}) - \mathbb{E}g(x_k)| \leq C\eta. \tag{19}$$

*We say Eq. 18 is an order 1 weak approximation of the update step of $x_k$.*

## C. New ODEs and SDEs for GD and SGD

### C.1. Comparison With Discrete-Time Analysis

In this section, we closely compare the dynamics of the loss function in discrete-time with that in continuous time as prescribed by the ODEs of GD. Consider GD with constant stepsize $\eta > 0$:

$$x_{t+1} = x_t - \eta\nabla f(x_t). \tag{20}$$

Using a second-order Taylor expansion around $x_t$ gives

$$f(x_{t+1}) - f(x_t) = -\eta\|\nabla f(x_t)\|^2 + \tfrac{\eta^2}{2}\nabla f(x_t)^\top \nabla^2 f(x_t) \nabla f(x_t) + O_{x_t}(\eta^3). \tag{21}$$

and therefore the normalized one-step loss drift (the discrete-time generator applied to $f$)

$$\frac{f(x_{t+1}) - f(x_t)}{\eta} = -\|\nabla f(x_t)\|^2 + \tfrac{\eta}{2}\nabla f(x_t)^\top \nabla^2 f(x_t) \nabla f(x_t) + O_{x_t}(\eta^2). \tag{22}$$

Here, $O_{x_t}(\eta^r)$ denotes a term bounded by $C(x_t)\eta^r$ for small enough $\eta$, with $C(x_t)$ independent of $\eta$.

*Remark (generator vs. finite increment).* Eq. 22 compares the *discrete-time generator* of GD applied to $f$ (a difference quotient) with the continuous-time drift $\frac{d}{dt}f(X_t)$. Matching the finite increment $f(X_{t+\eta}) - f(X_t)$ would instead introduce additional $\ddot{f}$ terms and corresponds to classical modified-equation analysis; this is *not* the notion of matching we use in this paper.

However, the first-order ODE of GD implies that

$$df(X_t) = -\|\nabla f(X_t)\|_2^2 dt. \tag{23}$$

We immediately notice that this continuous-time loss drift is completely missing the $O(\eta)$ correction highlighted in purple color in Eq. 22. The natural step is to shift to the second-order ODE, which implies that

$$df(X_t) = -\|\nabla f(X_t)\|_2^2 dt - \tfrac{\eta}{2}\nabla f(X_t)^\top \nabla^2 f(X_t)\nabla f(X_t)dt. \tag{24}$$

While this ODE of the loss does incorporate some second-order information highlighted in purple color, we notice that its sign is flipped with respect to the discrete-time loss drift in Eq. 22. This flipped sign is exactly the factor responsible for the failures of this second-order ODE.

**Deriving a New Model: An Ansatz Approach.** Therefore, we understand that choosing the right model is critical to capture the aspects of the dynamics under analysis. Inspired by a classic approach in mathematical physics, we propose an ansatz for an ODE of the iterates of GD and look for one that models the loss dynamics more closely. For a real number $\alpha$, we propose:

$$dX_t = -\nabla f(X_t)dt + \alpha\eta\nabla^2 f(X_t)\nabla f(X_t)dt, \tag{25}$$

which implies that the loss dynamics is driven by

$$df(X_t) = -\|\nabla f(X_t)\|_2^2 dt + \alpha\eta\nabla f(X_t)^\top \nabla^2 f(X_t)\nabla f(X_t)dt, \tag{26}$$

Matching the induced loss drift with the discrete-time generator expansion in Eq. 22 (i.e., comparing $\frac{d}{dt}f(X_t)$ with $\frac{f(x_{t+1})-f(x_t)}{\eta}$ up to $O(\eta)$) suggests choosing $\alpha = \frac{1}{2}$, which recovers the exact quadratic stability threshold. Therefore, we obtain

$$dX_t = -\nabla f(X_t)dt + \frac{\eta}{2}\nabla^2 f(X_t)\nabla f(X_t)dt, \tag{27}$$

is our new candidate ODE for GD.

## C.2. New Models

First, we define two new models for GD and SGD. Then, we introduce a technical lemma and proceed to prove that our new models are **first**-order models for (S)GD.

**Definition C.1.** Based on the discussion above, we define the new ODE model for GD:

$$dX_t = -\nabla f(X_t)dt + \frac{\eta}{2}\nabla^2 f(X_t)\nabla f(X_t)dt, \tag{28}$$

and the new SDE model for SGD:

$$dX_t = -\nabla f(X_t)dt + \frac{\eta}{2}\nabla^2 f(X_t)\nabla f(X_t)dt + \sqrt{\eta}\sqrt{\Sigma(X_t)}dW_t. \tag{29}$$

*Remark* C.2. Notice that, contrary to the second-order ODE and SDE from the literature (i.e., the classical modified-equation models), our stability-faithful models place a $+$ rather than a $-$ in front of the $\frac{\eta}{2}\nabla^2 f(X_t)\nabla f(X_t)$ correction. This sign choice is deliberate: it matches the discrete-time generator expansion of the loss and, in particular, recovers the exact quadratic stepsize stability threshold. Matching the finite increment $f(X_{t+\eta}) - f(X_t)$ instead leads back to the classical modified equation with the opposite sign; this is not our objective here.

**Theorem C.3.** *Under the dynamics $\dot{x} = F(x)$ such that $F \in C^3(\mathbb{R})$, fix $t$. One has the expansion*

$$x(t + \eta) = x + \eta F + \frac{\eta^2}{2}F'F + \frac{\eta^3}{6}\left(F''F^2 + (F')^2 F\right) + O(\eta^4),$$

*where all derivatives of $F$ are with respect to $x$, evaluated at $x(t)$.*

*Proof.* By Taylor's theorem about $t$,

$$x(t + \eta) = x(t) + \eta x'(t) + \frac{\eta^2}{2}x''(t) + \frac{\eta^3}{6}x'''(t) + O(\eta^4).$$

Note that:

$$x'(t) = F(x(t)), \quad x''(t) = F'(x(t))F(x(t)), \quad x'''(t) = F''(x(t))F(x(t))^2 + \left(F'(x(t))\right)^2 F(x(t)).$$

$\square$

**Theorem C.4** (ODE approximations of Gradient Descent)**.** *Consider gradient descent (GD) with constant stepsize $\eta > 0$. The following ODEs are all weak-approximations of GD:*

1. *The first-order approximation from the literature:*

$$dX_t = -\nabla f(X_t)\, dt. \tag{30}$$

2. *The second-order approximation from the literature:*

$$dX_t = -\nabla f(X_t)\, dt - \frac{\eta}{2}\nabla^2 f(X_t)\nabla f(X_t)\, dt. \tag{31}$$

3. *Our newly proposed first-order approximation:*

$$dX_t = -\nabla f(X_t)\, dt + \frac{\eta}{2}\nabla^2 f(X_t)\nabla f(X_t)\, dt. \tag{32}$$

*Proof.* For simplicity, we consider gradient descent in one dimension, as generalizing to higher dimensions follows the same steps:

$$x_{k+1} = x_k - \eta f'(x_k).$$

We now seek a flow of the form

$$F(x) = -f'(x) + \alpha f'(x) f''(x),$$

and just substitute in the expressions in the previous result. Then, we will study the error as a function of $\alpha$. Note that we want to compute

$$x(t + \eta) = x + \eta F + \frac{\eta^2}{2} F'F + \frac{\eta^3}{6} \left( F''F^2 + (F')^2 F \right) + O(\eta^4).$$

We have:

$$F = -f' + \alpha f' f'', \qquad F' = -f'' + \alpha\big((f'')^2 + f' f'''\big), \qquad F'' = -f''' + \alpha\big(3 f'' f''' + f' f''''\big).$$

So

$$x(t + \eta) = x + \eta\Big(-f' + \alpha f' f''\Big)$$

$$+ \frac{\eta^2}{2}\Big[f' f'' - \alpha\big(f'^2 f''' + 2 f'(f'')^2\big) + \alpha^2\big(f'^2 f'' f''' + f'(f'')^3\big)\Big]$$

$$+ \frac{\eta^3}{6}\Big[ -\big(f'^2 f''' + f'(f'')^2\big)$$

$$+ \alpha\big(f'^3 f'''' + 7 f'^2 f'' f''' + 3 f'(f'')^3\big)$$

$$- \alpha^2\big(2 f'^3 f'' f'''' + f'^3 (f''')^2 + 11 f'^2 (f'')^2 f''' + 3 f'(f'')^4\big)$$

$$+ \alpha^3\big(f'^3 (f'')^2 f'''' + f'^3 f''(f''')^2 + 5 f'^2 (f'')^3 f''' + f'(f'')^5\big)\Big]$$

$$+ O(\eta^4).$$

Assume now that $\alpha = \beta\eta$, we get

$$x(t + \eta) = x - \eta f'$$

$$+ \eta^2\left(\beta + \tfrac{1}{2}\right) f' f''$$

$$- \frac{\eta^3}{6}\Big[(3\beta + 1) f'^2 f''' + (6\beta + 1) f'(f'')^2\Big]$$

$$+ O(\eta^4),$$

For $\alpha = 0$ we get gradient flow and hence

$$x(t + \eta) = x - \eta f' + \tfrac{1}{2} \eta^2 f' f'' - \tfrac{1}{6} \eta^3 \left( f'^2 f''' + f'(f'')^2 \right) + O(\eta^4),$$

which is the first-order ODE from the literature.

For $\alpha = -\eta/2$

$$x(t + \eta) = x - \eta f' + \eta^3 \left( \tfrac{1}{12} f'^2 f''' + \tfrac{1}{3} f'(f'')^2 \right) + O(\eta^4),$$

which is the second-order ODE from the literature.

Finally, for $\alpha = \eta/2$,

$$x(t + \eta) = x - \eta f' + \eta^2 f' f'' - \tfrac{5}{12}\eta^3 f'^2 f''' - \tfrac{2}{3}\eta^3 f'(f'')^2 + O(\eta^4),$$

which is our newly proposed first-order ODE.

$\square$

The following theorem formalizes that our new SDE model from Eq. 29 is formally a first-order weak approximation for SGD.

**Theorem C.5** (SDE approximations of Stochastic Gradient Descent)**.** *Consider stochastic gradient descent with constant stepsize $\eta > 0$. Its continuous-time approximations are given by the following SDEs:*

1. *The first-order approximation from the literature:*

$$dX_t = -\nabla f(X_t)\, dt + \sqrt{\eta \Sigma(X_t)} dW_t. \tag{33}$$

2. *The second-order approximation from the literature:*

$$dX_t = -\nabla f(X_t)\, dt - \frac{\eta}{2} \nabla^2 f(X_t)\, \nabla f(X_t)\, dt + \sqrt{\eta \Sigma(X_t)} dW_t. \tag{34}$$

3. *Our newly proposed first-order approximation:*

$$dX_t = -\nabla f(X_t)\, dt + \frac{\eta}{2} \nabla^2 f(X_t)\, \nabla f(X_t)\, dt + \sqrt{\eta \Sigma(X_t)} dW_t. \tag{35}$$

Here is the formal proof:

*Proof.* We work under the framework and assumptions of Section B. Let $(x_k)_{k \geq 0}$ denote the SGD iterates with constant stepsize $\eta > 0$:

$$x_{k+1} = x_k - \eta\, g(x_k, \xi_{k+1}),$$

where $\{\xi_k\}_{k \geq 1}$ are i.i.d. random variables and $g(\cdot, \xi)$ is an unbiased stochastic gradient estimator:

$$\mathbb{E}[g(x, \xi)] = \nabla f(x), \qquad \mathrm{Cov}(g(x, \xi)) = \Sigma(x).$$

We write the gradient noise as

$$\zeta(x, \xi) \coloneqq g(x, \xi) - \nabla f(x),$$

so that $\mathbb{E}[\zeta(x, \xi)] = 0$ and $\mathbb{E}[\zeta(x, \xi)\zeta(x, \xi)^\top] = \Sigma(x)$, with bounded moments up to order 3 (consistent with (Li et al., 2017) and Assumption B).

Throughout, we fix $0 < \eta < 1$ and condition on $x_k = x$.

**Step 1: One-step moments of SGD.**  Define the one-step increment of SGD by

$$\bar{\Delta} \coloneqq x_{k+1} - x_k = -\eta\, \nabla f(x_k) - \eta\, \zeta(x_k, \xi_{k+1}).$$

Conditioned on $x_k = x$, we thus have

$$\bar{\Delta} = -\eta\, \nabla f(x) - \eta\, \zeta,$$

where we write $\zeta \coloneqq \zeta(x, \xi_{k+1})$.

Writing components as $\bar{\Delta}_i$, $f_i = \partial_i f$, and $\Sigma_{ij}$ for the $(i, j)$-th entry of $\Sigma(x)$, we get:

$$\mathbb{E}[\bar{\Delta}_i \mid x_k = x] = -\eta\, \mathbb{E}\big[f_i(x) + \zeta_i \mid x_k = x\big] = -\eta\, f_i(x),$$

$$\mathbb{E}[\bar{\Delta}_i \bar{\Delta}_j \mid x_k = x] = \eta^2\, \mathbb{E}\big[(f_i(x) + \zeta_i)(f_j(x) + \zeta_j) \mid x_k = x\big]$$
$$= \eta^2\big(f_i(x) f_j(x) + \Sigma_{ij}(x)\big).$$

Moreover, for any $s \geq 3$ and indices $i_1, \ldots, i_s$, we have

$$\mathbb{E}\Big[\prod_{\ell=1}^{s} \bar{\Delta}_{i_\ell} \,\Big|\, x_k = x\Big] = O(\eta^s) = O(\eta^3),$$

thanks to the bounded higher moments of $\zeta$ and Assumption B. In particular, there exists $K_4 \in \mathcal{G}$ such that

$$\mathbb{E}\Big[\prod_{\ell=1}^{s} \big|\bar{\Delta}_{i_\ell}\big| \,\Big|\, x_k = x\Big] \leq K_4(x)\, \eta^3 \leq K_4(x)\, \eta^2,$$

for all $0 < \eta < 1$ and all $s \geq 3$. Hence condition (4) in Theorem B.5 holds.

**Step 2: One-step moments of the SDE family.** We now put the three candidate SDE models in Theorem C.5 into the template of Lemma B.4 and Theorem B.5.

Fix $\alpha \in \mathbb{R}$ and consider the SDE

$$dX_t = b_\alpha(X_t)\, dt + \sqrt{\eta}\, \sigma(X_t)\, dW_t, \tag{36}$$

with

$$b_\alpha(x) := -\nabla f(x) + \alpha\, \eta\, \nabla^2 f(x)\, \nabla f(x), \qquad \sigma(x)\sigma(x)^\top = \Sigma(x).$$

We will later set

$$\alpha = 0, -\tfrac{1}{2}, \tfrac{1}{2}$$

to recover the three SDEs in the statement.

Let $X_0 = x$ and define the one-step increment

$$\Delta := X_\eta - x,$$

with components $\Delta_i$. By Lemma B.4 applied to 36, we have

$$\mathbb{E}[\Delta_i] = b_{\alpha,i}(x)\, \eta + \frac{1}{2} \sum_{j=1}^{d} b_{\alpha,j}(x)\, \partial_j b_{\alpha,i}(x)\, \eta^2 + O(\eta^3), \tag{37}$$

$$\mathbb{E}[\Delta_i \Delta_j] = \Big[b_{\alpha,i}(x) b_{\alpha,j}(x) + \big(\sigma\sigma^\top\big)_{ij}(x)\Big]\eta^2 + O(\eta^3)$$

$$= \Big[b_{\alpha,i}(x) b_{\alpha,j}(x) + \Sigma_{ij}(x)\Big]\eta^2 + O(\eta^3), \tag{38}$$

$$\mathbb{E}\Big[\prod_{\ell=1}^{s} \Delta_{i_\ell}\Big] = O(\eta^3), \qquad \forall s \geq 3. \tag{39}$$

All functions above belong to $\mathcal{G}$ by Assumption B.

Now plug $b_\alpha(x) = -\nabla f(x) + \alpha\eta\, \nabla^2 f(x)\, \nabla f(x)$ into these expressions. Write

$$h(x) := \nabla^2 f(x)\, \nabla f(x), \qquad h_i(x) = \sum_{j=1}^{d} \partial_{ij} f(x)\, \partial_j f(x).$$

Then

$$b_{\alpha,i}(x) = -f_i(x) + \alpha\, \eta\, h_i(x).$$

For the *first* moment, using 37, we expand to order $\eta^2$:

$$\mathbb{E}[\Delta_i] = \Big(-f_i(x) + \alpha\eta\, h_i(x)\Big)\eta + \frac{1}{2} \sum_{j=1}^{d} \Big(-f_j(x) + O(\eta)\Big) \partial_j\Big(-f_i(x) + O(\eta)\Big)\eta^2 + O(\eta^3)$$

$$= -\eta f_i(x) + \alpha\, \eta^2 h_i(x) + \frac{1}{2} \sum_{j=1}^{d} f_j(x)\, \partial_j f_i(x)\, \eta^2 + O(\eta^3)$$

$$= -\eta f_i(x) + \Big(\alpha + \tfrac{1}{2}\Big) h_i(x)\, \eta^2 + O(\eta^3).$$

In particular,

$$\mathbb{E}[\Delta_i] - \mathbb{E}[\bar{\Delta}_i] = \left(\alpha + \tfrac{1}{2}\right) h_i(x)\,\eta^2 + O(\eta^3) = O(\eta^2). \tag{40}$$

For the *second* moment, from 38 we note that

$$b_{\alpha,i}(x)b_{\alpha,j}(x) = \left(-f_i(x) + O(\eta)\right)\left(-f_j(x) + O(\eta)\right) = f_i(x)f_j(x) + O(\eta),$$

hence

$$\mathbb{E}[\Delta_i \Delta_j] = \left(f_i(x)f_j(x) + \Sigma_{ij}(x)\right)\eta^2 + O(\eta^3), \tag{41}$$

so that

$$\mathbb{E}[\Delta_i \Delta_j] - \mathbb{E}[\bar{\Delta}_i \bar{\Delta}_j] = O(\eta^3) \le K_2(x)\,\eta^2,$$

for some $K_2 \in \mathcal{G}$ and all $0 < \eta < 1$. Similarly, combining 39 with the bound on higher moments of $\bar{\Delta}$ from Step 1, we obtain for $s \ge 3$

$$\left|\mathbb{E}\prod_{\ell=1}^{s}\Delta_{i_\ell} - \mathbb{E}\prod_{\ell=1}^{s}\bar{\Delta}_{i_\ell}\right| = O(\eta^3) \le K_3(x)\,\eta^2$$

for some $K_3 \in \mathcal{G}$, and we have already seen that

$$\mathbb{E}\prod_{\ell=1}^{s}|\bar{\Delta}_{i_\ell}| \le K_4(x)\,\eta^2$$

for appropriate $K_4 \in \mathcal{G}$. Thus, all four conditions of Theorem B.5 are satisfied for 36, for any fixed $\alpha \in \mathbb{R}$.

**Step 3: Weak order and identification of the three SDEs.** By Theorem B.5, for any fixed $\alpha \in \mathbb{R}$ the SDE

$$dX_t = b_\alpha(X_t)\,dt + \sqrt{\eta}\,\sigma(X_t)\,dW_t$$

is an *order* 1 *weak approximation* of the SGD recursion, in the sense of Definition B.2.

It remains to identify the three specific choices of $\alpha$ that correspond to the SDEs in the statement.

- **Case $\alpha = 0$.** Then $b_0(x) = -\nabla f(x)$, so the SDE 36 becomes

$$dX_t = -\nabla f(X_t)\,dt + \sqrt{\eta\,\Sigma(X_t)}\,dW_t,$$

which is exactly the *first*-order SDE approximation from the literature in Eq. 33. By Theorem B.5, this is an order 1 weak approximation of SGD.

- **Case $\alpha = -\tfrac{1}{2}$.** Then

$$b_{-1/2}(x) = -\nabla f(x) - \frac{\eta}{2}\,\nabla^2 f(x)\,\nabla f(x),$$

and 36 becomes

$$dX_t = -\nabla f(X_t)\,dt - \frac{\eta}{2}\,\nabla^2 f(X_t)\,\nabla f(X_t)\,dt + \sqrt{\eta\,\Sigma(X_t)}\,dW_t,$$

which is exactly Eq. 34, the classical *second*-order SDE from the literature. In this case, 40 shows that

$$\mathbb{E}[\Delta_i] - \mathbb{E}[\bar{\Delta}_i] = O(\eta^3),$$

so the drift matches to one order higher; combined with the analysis in (Li et al., 2017), this yields a second-order weak approximation. In particular, it is also an order 1 weak approximation.

- **Case $\alpha = \tfrac{1}{2}$.** Then

$$b_{1/2}(x) = -\nabla f(x) + \frac{\eta}{2}\,\nabla^2 f(x)\,\nabla f(x),$$

and 36 becomes

$$dX_t = -\nabla f(X_t)\,dt + \frac{\eta}{2}\,\nabla^2 f(X_t)\,\nabla f(X_t)\,dt + \sqrt{\eta\,\Sigma(X_t)}\,dW_t,$$

which is exactly our new SDE model in Eq. 35. As shown above, all four conditions of Theorem B.5 hold, so this SDE is also an *order* 1 *weak approximation* of SGD.

This proves that all three SDEs listed in Theorem C.5 are weak approximations of SGD in the sense of Definition B.2: the first and third are first-order weak approximations, while the second one is the classical second-order stochastic modified equation from the literature.

$\square$

## C.3. Comparing ODEs - An Insight Perspective

In this section, we showcase how models from the literature fail to properly model the dynamics of GD, especially regarding the constraints on the learning rate to ensure convergence. In contrast, we show that our model is in accordance with GD.

### C.3.1. QUADRATIC FUNCTION

For didactic reasons, we now compare the proofs for a convergence bound on the loss value $f(x)$ when the loss is a 1-dimensional convex quadratic function $\frac{\lambda x^2}{2}$. **To avoid overloading the proof with technicalities intrinsic in Itô calculus**, we restrict the analysis to the noiseless and single-node case. The *first*-order ODE is

$$dX_t = -\nabla f(X_t)dt = -\lambda X_t dt, \tag{42}$$

which implies that

$$df(X_t) = -2\lambda f(X_t)dt \implies f(X_t) = f(X_0)e^{-2\lambda t} \overset{t\to\infty}{\to} 0, \tag{43}$$

somewhat implying that GD converges independently of the constant $L$ and of the learning rate $\eta$. Much differently, the *second*-order ODE *from the literature* is

$$dX_t = -\nabla f(X_t)dt - \tfrac{\eta}{2}\nabla^2 f(X_t)\nabla f(X_t)dt, \tag{44}$$

which implies that

$$df(X_t) = -\|\nabla f(X_t)\|_2^2 dt - \tfrac{\eta}{2}\nabla f(X_t)^\top \nabla^2 f(X_t)\nabla f(X_t)dt = -2\lambda f(X_t)dt - \frac{\eta}{2}\lambda X_t^\top \lambda\lambda X_t \tag{45}$$

$$= -2\lambda\left(1 + \frac{\lambda\eta}{2}\right)f(X_t)dt \implies f(X_t) = f(X_0)e^{-2\lambda\left(1+\frac{\lambda\eta}{2}\right)t} \overset{t\to\infty}{\to} 0, \tag{46}$$

which is also inconsistent with the discrete-time analysis since we get convergence for any $\eta > 0$.

Now, we try to leverage our new ODE derived in Theorem C.4 and get that:

$$df(X_t) = -\|\nabla f(X_t)\|_2^2 dt + \tfrac{\eta}{2}\nabla f(X_t)^\top \nabla^2 f(X_t)\nabla f(X_t)dt = -2\lambda f(X_t)dt + \frac{\eta}{2}\lambda X_t^\top \lambda\lambda X_t \tag{47}$$

$$= -2\lambda\left(1 - \frac{\lambda\eta}{2}\right)f(X_t)dt \implies f(X_t) = f(X_0)e^{-2\lambda\left(1-\frac{\lambda\eta}{2}\right)t} \overset{t\to\infty}{\to} 0, \tag{48}$$

which only converges if $\eta < \frac{2}{\lambda}$. This is consistent with the analysis in discrete time.

**Conclusion:** First of all, it is immediately apparent that while *first*-order approximations may lead to relevant insights, they prevent us from having a full picture. Second, we demonstrated that the classic *second*-order SDE also led us to results that are inconsistent with the discrete-time analysis. Finally, our model provides a qualitatively faithful description of the true GD dynamics.

### C.3.2. QUARTIC FUNCTION

Here, we compare the three ODEs listed above as they describe the optimization of a quartic function $f(x) = \frac{x^4}{4}$: We find that the classic ones both fail. First of all, a single step of gradient descent with stepsize $\eta$ reads

$$x_{k+1} = x_k - \eta\nabla f(x_k) = x_k - \eta x_k^3,$$

meaning that if $\eta > \frac{2}{x_k^2}$ the dynamics *explodes*. In particular,

$$\frac{f(x_{k+1}) - f(x_k)}{\eta} = -x_k^6 + \frac{3}{2}\eta x_k^8 + O(\eta^2). \tag{49}$$

Using the first-order ODE, we obtain

$$dX_t = -X_t^3 dt \implies f(X_t) = \frac{1}{4(2t + X_0^{-2})^2} \tag{50}$$

This model predicts universal convergence with a polynomial rate, but it does *not* capture the exploding behaviour observed in GD. Using the second-order ODE, we obtain

$$dX_t = -X_t^3 dt - \tfrac{3\eta}{2} X_t^5 dt \implies df(X_t) = -X_t^6 dt - \tfrac{3\eta}{2} X_t^8 dt, \tag{51}$$

from which we understand that since the additional term is *negative*, this ODE suggests *faster convergence* for larger $\eta$. Using our new ODE, we obtain

$$dX_t = -X_t^3 dt + \tfrac{3\eta}{2} X_t^5 dt \implies df(X_t) = -X_t^6 dt + \tfrac{3\eta}{2} X_t^8 dt, \tag{52}$$

which matches the discrete-time expansion of the loss difference quotient up to $O(\eta^2)$. Importantly, it captures the phenomenon that the learning rate $\eta$ needs to scale inversely to the norm of the iterates for GD to converge.

**Conclusion.** On the quartic loss, the first-order ODE predicts convergence for all $\eta$, missing the instability. The second-order ODE from the literature predicts *accelerated convergence* for larger $\eta$, in direct contradiction with GD. In contrast, our new ODE reproduces the key phenomenon: the learning rate $\eta$ needs to scale inversely to the norm of the iterates for GD to converge. Hence, our model provides a qualitatively faithful description of the true GD dynamics.

### C.4. Diffusion Approximation for the Loss in SGD

In this section, we propose an alternative approach to the derivation of a continuous-time model for SGD. Rather than modeling the iterates and use the Itô Lemma to study the SDE of the loss function, we try a new approach: We directly investigate the possibility of directly modeling the dynamics of the loss. Consider stochastic gradient descent (SGD) with constant stepsize $\eta > 0$:

$$x_{t+1} = x_t - \eta g_t, \qquad g_t = \nabla f(x_t) + \zeta_t, \tag{53}$$

where $f : \mathbb{R}^d \to \mathbb{R}$ is smooth, $\zeta_t$ is the gradient noise satisfying

$$\mathbb{E}[\zeta_t \mid x_t] = 0, \qquad \mathrm{Cov}(\zeta_t \mid x_t) = \Sigma(x_t).$$

We study the dynamics of the *loss process* $Y_t := f(x_t)$.

**Step 1. Taylor expansion of the loss.** Using a second-order Taylor expansion around $x_t$, for $h = -\eta g_t$ we have

$$
\begin{aligned}
f(x_{t+1}) &= f(x_t + h) \\
&= f(x_t) + \nabla f(x_t)^\top h + \tfrac{1}{2} h^\top \nabla^2 f(x_t) h + O(\|h\|^3).
\end{aligned} \tag{54}
$$

Substituting $h = -\eta g_t$ gives

$$f(x_{t+1}) - f(x_t) = -\eta \nabla f(x_t)^\top g_t + \tfrac{\eta^2}{2} g_t^\top \nabla^2 f(x_t) g_t + O(\eta^3). \tag{55}$$

**Step 2. Expansion of stochastic terms.** Expanding with $g_t = \nabla f(x_t) + \zeta_t$ yields

$$
\begin{aligned}
f(x_{t+1}) - f(x_t) = &-\eta \|\nabla f(x_t)\|^2 - \eta \nabla f(x_t)^\top \zeta_t \\
&+ \tfrac{\eta^2}{2} \nabla f(x_t)^\top \nabla^2 f(x_t) \nabla f(x_t) \\
&+ \tfrac{\eta^2}{2} \zeta_t^\top \nabla^2 f(x_t) \zeta_t + \eta^2 \nabla f(x_t)^\top \nabla^2 f(x_t) \zeta_t + O(\eta^3).
\end{aligned} \tag{56}
$$
$$ \tag{57}$$

**Step 3. Drift and volatility.** Taking the conditional expectation given $x_t$,

$$\mathbb{E}[f(x_{t+1}) - f(x_t) \mid x_t] = -\eta \|\nabla f(x_t)\|^2$$
$$+ \tfrac{\eta^2}{2} \nabla f(x_t)^\top \nabla^2 f(x_t) \nabla f(x_t) + \tfrac{\eta^2}{2} \operatorname{tr}\left(\nabla^2 f(x_t) \Sigma(x_t)\right) + O(\eta^3). \tag{58}$$

The stochastic fluctuations arise from the linear terms in $\zeta_t$,

$$-\eta \nabla f(x_t)^\top \zeta_t + \eta^2 \nabla f(x_t)^\top \nabla^2 f(x_t) \zeta_t,$$

whose leading-order contribution is

$$-\eta \nabla f(x_t)^\top \zeta_t.$$

This term has conditional variance

$$\operatorname{Var}\left(-\eta \nabla f(x_t)^\top \zeta_t \,\big|\, x_t\right) = \eta^2 \nabla f(x_t)^\top \Sigma(x_t) \nabla f(x_t).$$

**Step 4. Continuous-time limit.** Rescaling time by $s = t\eta$ and letting $\eta \to 0$, the increments 57 converge in distribution to the diffusion

$$dY_s = \left(-\|\nabla f(X_s)\|^2 + \tfrac{\eta}{2} \nabla f(X_s)^\top \nabla^2 f(X_s) \nabla f(X_s) + \tfrac{\eta}{2} \operatorname{tr}\left(\nabla^2 f(X_s) \Sigma(X_s)\right)\right) ds + G(X_s)\, dW_s, \tag{59}$$

where $W_s$ is a standard Brownian motion and the scalar volatility $G(x)$ is defined by

$$G(x)^2 = \nabla f(x)^\top \Sigma(x)\, \nabla f(x). \tag{60}$$

Interestingly, this SDE is the same one that one gets by applying Itô's Lemma on $f(X_t)$ under the dynamics of our newly proposed SDE in Eq. 29, which consolidates the intuition that our model properly captures the dynamics of SGD faithfully.

# D. Theoretical Results

**Assumptions and notation.** In line with (Compagnoni et al., 2025a), we assume that the stochastic gradient of the $i$-th client is given by $\nabla f_{\gamma_i}(x) = \nabla f(x) + Z_i(x)$, where $Z_i(x)$ denotes the gradient noise and $Z_i(x)$ is independent of $Z_j(x)$ for $i \neq j$. If $Z_i(x) \in L^1(\mathbb{R}^d)$, we assume $\mathbb{E}[Z_i(x)] = 0$, and if $Z_i(x) \in L^2(\mathbb{R}^d)$, we assume $Cov(Z_i(x)) = \Sigma_i(x)$ (we omit the size of the batch $\gamma$ unless relevant) s.t. $\sqrt{\Sigma_i(x)}$ is bounded, Lipschitz, satisfies affine growth, and together with its derivatives, it grows at most polynomially fast (Definition 2.5 in (Malladi et al., 2022)). Importantly, we assume that all $Z_i(x)$ have a smooth and bounded probability density function whose derivatives are all integrable: A common assumption in the literature is for $Z_i(x)$ to be Gaussian (Ahn et al., 2012; Chen et al., 2014; Mandt et al., 2016; 2017; Zhu et al., 2019; Wu et al., 2020; Xie et al., 2021): See (Jastrzebski et al., 2018) for the justification why this could be the case. Differently, our assumption allows for heavy-tailed distributions such as the Student's t. It is important to point out that (Li et al., 2017; Mertikopoulos & Staudigl, 2018; Raginsky & Bouvrie, 2012; Zhu et al., 2019; Mandt et al., 2016; Ahn et al., 2012; Jastrzebski et al., 2018) use a Gaussian noise with a constant covariance matrix to model batch noise.

## D.1. Distributed SGD

### D.1.1. FIRST ORDER SDE

The following is the *first*-order SDE model of DSGD (see Theorem 3.2 in (Compagnoni et al., 2025a)). Let us consider the stochastic process $X_t \in \mathbb{R}^d$ defined as the solution of

$$dX_t = -\nabla f(X_t) dt + \sqrt{\frac{\eta}{N}} \sqrt{\hat{\Sigma}(X_t)} dW_t, \tag{61}$$

where $\hat{\Sigma}(x) := \frac{1}{N} \sum_{i=1}^N \Sigma_i(x)$ is the average of the covariance matrices of the $N$ clients.

**Theorem D.1.** *Let $f$ be $(L_0, L_1)$-smooth, $\|\Sigma_i(x)\|_\infty < \sigma_{0,i}^2 + \sigma_{1,i}^2 \|\nabla f(x)\|_2^2$, the learning rate scheduler $\eta_t$ s.t. $\phi_t^{(i)} = \int_0^t (\eta_s)^i ds$, $\phi_t^{(1)} \overset{t \to \infty}{\to} \infty$, $\frac{\phi_t^{(2)}}{\phi_t^{(1)}} \overset{t \to \infty}{\to} 0$, $\overline{\sigma_0^2} := \frac{1}{N} \sum_{i=1}^N \sigma_{0,i}^2$, and $\overline{\sigma_1^2} := \frac{1}{N} \sum_{i=1}^N \sigma_{1,i}^2$. Then, for $0 < \epsilon < 1$,*

$$\eta \eta_t < \frac{2N\epsilon}{d \left(\overline{\sigma_1^2} L_0 + \overline{\sigma_0^2} L_1 + L_1 \overline{\sigma_1^2} \mathbb{E}\left[\|\nabla f(X_t)\|_2\right]\right)}, \tag{62}$$

*and for a random time $\hat{t}$ with distribution $\frac{\eta_t}{\phi_t^{(1)}}$, we have that*

$$\mathbb{E}\left[\|\nabla f(X_{\hat{t}})\|_2^2\right] \leq \frac{1}{\phi_t^{(1)}(1-\epsilon)}\left(f(X_0) - f(X_*) + \phi_t^{(2)}\frac{\eta d(L_0 + L_1)(\overline{\sigma_0^2} + \overline{\sigma_1^2})}{2N}\right) \overset{t\to\infty}{\Rightarrow} 0. \tag{63}$$

*Proof.* Using Itô's Lemma and using a learning rate scheduler $\eta_t$ during the derivation of the SDE, we have

$$d(f(X_t) - f(X_*)) = -\eta_t\|\nabla f(X_t)\|_2^2 dt + \mathcal{O}(\text{Noise}) + (\eta_t)^2\frac{\eta}{2N}\text{Tr}(\nabla^2 f(X_t)\tilde{\Sigma}(X_t))dt \tag{64}$$

$$\leq -\eta_t\|\nabla f(X_t)\|_2^2 dt + \mathcal{O}(\text{Noise}) \tag{65}$$

$$+ (\eta_t)^2\frac{\eta(\overline{\sigma_0^2} + \overline{\sigma_1^2}\|\nabla f(X_t)\|_2^2)d(L_0 + L_1\|\nabla f(X_t)\|)}{2N}dt, \tag{66}$$

where we used that $\text{Tr}\left(\nabla^2 f(x)\tilde{\Sigma}(x)\right) \leq d\|\nabla^2 f(x)\|_\infty\|\tilde{\Sigma}(x)\|_\infty$ together with the smoothness and noise assumptions. Importantly, $\mathcal{O}(\text{Noise}) = \sqrt{\tilde{\Sigma}(X_t)}\nabla f(X_t)dW_t$.

**Phase 1:** If $\|\nabla f(X_t)\| \leq 1$, we have that

$$d(f(X_t) - f(X_*)) \leq -\eta_t\|\nabla f(X_t)\|_2^2 dt + (\eta_t)^2\frac{\eta(\overline{\sigma_0^2} + \overline{\sigma_1^2})d(L_0 + L_1)}{2N}dt + \mathcal{O}(\text{Noise}), \tag{67}$$

**Phase 2:** If $\|\nabla f(X_t)\| > 1$, we have

$$d(f(X_t) - f(X_*)) = -\eta_t\|\nabla f(X_t)\|_2^2 dt + \mathcal{O}(\text{Noise}) + (\eta_t)^2\frac{\eta}{2N}\text{Tr}(\nabla^2 f(X_t)\tilde{\Sigma}(X_t))dt \tag{68}$$

$$\leq -\eta_t\|\nabla f(X_t)\|_2^2 dt + \mathcal{O}(\text{Noise}) \tag{69}$$

$$+ (\eta_t)^2\frac{\eta(\overline{\sigma_0^2} + \overline{\sigma_1^2}\|\nabla f(X_t)\|_2^2)d(L_0 + L_1\|\nabla f(X_t)\|)}{2N}dt \tag{70}$$

$$= -\eta_t\|\nabla f(X_t)\|_2^2\left(1 - \frac{\eta_t\eta d}{2N}\left(\overline{\sigma_1^2}L_0 + \overline{\sigma_0^2}L_1 + L_1\overline{\sigma_1^2}\|\nabla f(X_t)\|_2\right)\right)dt \tag{71}$$

$$+ (\eta_t)^2\frac{\eta\overline{\sigma_0^2}dL_0}{2N}dt + \mathcal{O}(\text{Noise}). \tag{72}$$

By taking a worst-case scenario approach, we merge these two bounds into a single one:

$$d(f(X_t) - f(X_*)) \leq -\eta_t\|\nabla f(X_t)\|_2^2\left(1 - \frac{\eta_t\eta d}{2N}\left(\overline{\sigma_1^2}L_0 + \overline{\sigma_0^2}L_1 + L_1\overline{\sigma_1^2}\|\nabla f(X_t)\|_2\right)\right)dt \tag{73}$$

$$+ (\eta_t)^2\frac{\eta d(L_0 + L_1)(\overline{\sigma_0^2} + \overline{\sigma_1^2})}{2N}dt + \mathcal{O}(\text{Noise}). \tag{74}$$

Therefore, for $0 < \epsilon < 1$ we have that if

$$1 - \frac{\eta_t\eta d}{2N}\left(\overline{\sigma_1^2}L_0 + \overline{\sigma_0^2}L_1 + L_1\overline{\sigma_1^2}\|\nabla f(X_t)\|_2\right) > 1 - \epsilon, \tag{75}$$

or, equivalently

$$\eta\eta_t < \frac{2N\epsilon}{d\left(\overline{\sigma_1^2}L_0 + \overline{\sigma_0^2}L_1 + L_1\overline{\sigma_1^2}\|\nabla f(X_t)\|_2\right)}, \tag{76}$$

we have that

$$d(f(X_t) - f(X_*)) \leq -\eta_t\|\nabla f(X_t)\|_2^2(1-\epsilon)\,dt + (\eta_t)^2\frac{\eta d(L_0 + L_1)(\overline{\sigma_0^2} + \overline{\sigma_1^2})}{2N}dt + \mathcal{O}(\text{Noise}). \tag{77}$$

Therefore,

$$\eta_t \|\nabla f(X_t)\|_2^2 (1 - \epsilon) \, dt \leq -d(f(X_t) - f(X_*)) + (\eta_t)^2 \frac{\eta d(L_0 + L_1)(\overline{\sigma_0^2} + \overline{\sigma_1^2})}{2N} dt + \mathcal{O}(\text{Noise}). \tag{78}$$

Dividing by $1 - \epsilon$, integrating over time, and using the martingality of the noise term under the expected value,

$$\int_0^t \eta_s \mathbb{E}\|\nabla f(X_s)\|_2^2 ds \leq \frac{1}{1 - \epsilon} \left( f(X_0) - f(X_*) + \phi_t^{(2)} \frac{\eta d(L_0 + L_1)(\overline{\sigma_0^2} + \overline{\sigma_1^2})}{2N} \right). \tag{79}$$

Dividing by $\phi_t^{(1)}$ and by the Law of the Unconscious Statistician, we have that

$$\mathbb{E}\left[\|\nabla f(X_{\hat{t}})\|_2^2\right] \leq \frac{1}{\phi_t^{(1)}(1 - \epsilon)} \left( f(X_0) - f(X_*) + \phi_t^{(2)} \frac{\eta d(L_0 + L_1)(\overline{\sigma_0^2} + \overline{\sigma_1^2})}{2N} \right) \overset{t \to \infty}{\to} 0, \tag{80}$$

where $\hat{t}$, is a random time with distribution $\frac{\eta_{\hat{t}}}{\phi_t^{(1)}}$.

Finally, for practical reasons, we leverage the distributed setting to tighten the requirements on the learning rate scheduler to make it experimentally viable (see Section E.5 for the details), and require

$$\eta \eta_t < \frac{2N\epsilon}{d \left( \overline{\sigma_1^2} L_0 + \overline{\sigma_0^2} L_1 + L_1 \overline{\sigma_1^2} \mathbb{E}\left[\|\nabla f(X_t)\|_2\right] \right)}. \tag{81}$$

$\square$

### D.1.2. OUR NEW FIRST-ORDER SDE FOR DSGD

The following is the *first*-order SDE model of DSGD and is a straightforward generalization of Theorem 3.2 in (Compagnoni et al., 2025a) and Remark C.2. Let us consider the stochastic process $X_t \in \mathbb{R}^d$ defined as the solution of

$$dX_t = -\nabla f(X_t)dt + \frac{\eta}{2}\nabla^2 f(X_t)\nabla f(X_t)dt + \sqrt{\frac{\eta}{N}}\sqrt{\hat{\Sigma}(X_t)}dW_t, \tag{82}$$

where $\hat{\Sigma}(x) := \frac{1}{N}\sum_{i=1}^N \Sigma_i(x)$ is the average of the covariance matrices of the $N$ clients.

**Theorem D.2.** *Let $f$ be $(L_0, L_1)$-smooth, $\|\Sigma_i(x)\|_\infty < \sigma_{0,i}^2 + \sigma_{1,i}^2\|\nabla f(x)\|_2^2$, the learning rate scheduler $\eta_t$ s.t. $\phi_t^{(i)} = \int_0^t (\eta_s)^i ds$, $\phi_t^{(1)} \overset{t \to \infty}{\to} \infty$, $\frac{\phi_t^{(2)}}{\phi_t^{(1)}} \overset{t \to \infty}{\to} 0$, $\overline{\sigma_0^2} := \frac{1}{N}\sum_{i=1}^N \sigma_{0,i}^2$, and $\overline{\sigma_1^2} := \frac{1}{N}\sum_{i=1}^N \sigma_{1,i}^2$. Then, for $0 < \epsilon < 1$,*

$$\eta \eta_t < \frac{2\epsilon}{L_0 + L_1\mathbb{E}\left[\|\nabla f(X_t)\|\right] + \frac{d}{N}\left(\overline{\sigma_1^2}L_0 + \overline{\sigma_0^2}L_1 + L_1\overline{\sigma_1^2}\mathbb{E}\left[\|\nabla f(X_t)\|\right]\right)}, \tag{83}$$

*and for a random time $\hat{t}$ with distribution $\frac{\eta_t}{\phi_t^{(1)}}$, we have that*

$$\mathbb{E}\left[\|\nabla f(X_{\hat{t}})\|_2^2\right] \leq \frac{1}{\phi_t^{(1)}(1 - \epsilon)} \left( f(X_0) - f(X_*) + \frac{\eta \phi_t^{(2)}}{2N}(L_0 + L_1)d\overline{\sigma_0^2} \right) \overset{t \to \infty}{\to} 0. \tag{84}$$

*Proof.* Using Itô's Lemma and using a learning rate scheduler $\eta_t$ during the derivation of the SDE, we have that for $\mathcal{O}(\text{Noise}) = \sqrt{\tilde{\Sigma}(X_t)}\nabla f(X_t)dW_t$,

$$d(f(X_t) - f(X_*)) = -\eta_t\|\nabla f(X_t)\|_2^2 dt + \frac{\eta\eta_t^2}{2}(\nabla f(X_t))^\top \nabla^2 f(X_t)\nabla f(X_t)dt \tag{85}$$

$$+ \mathcal{O}(\text{Noise}) + (\eta_t)^2 \frac{\eta}{2N}\text{Tr}(\nabla^2 f(X_t)\tilde{\Sigma}(X_t))dt \tag{86}$$

$$\leq -\eta_t\|\nabla f(X_t)\|_2^2 dt + \frac{\eta\eta_t^2}{2}(L_0 + L_1\|\nabla f(X_t)\|)\|\nabla f(X_t)\|^2 dt \tag{87}$$

$$+ \mathcal{O}(\text{Noise}) + (\eta_t)^2 \frac{\eta(\overline{\sigma_0^2} + \overline{\sigma_1^2}\|\nabla f(X_t)\|_2^2)d(L_0 + L_1\|\nabla f(X_t)\|)}{2N} dt. \tag{88}$$

**Phase** 1**:** If $\|\nabla f(X_t)\| \leq 1$,

$$d(f(X_t) - f(X_*)) \leq -\|\nabla f(X_t)\|_2^2 \left( \eta_t - \frac{\eta \eta_t^2}{2}(L_0 + L_1\|\nabla f(X_t)\|_2) \left( 1 + \frac{d\overline{\sigma_1^2}}{N} \right) \right) dt \tag{89}$$

$$+ \frac{\eta \eta_t^2}{2N} \cdot (L_0 + L_1) d\overline{\sigma_0^2} dt + \mathcal{O}(\text{Noise}). \tag{90}$$

**Phase** 2**:** If $\|\nabla f(X_t)\| > 1$, we have

$$d(f(X_t) - f(X_*)) \leq -\eta_t \|\nabla f(X_t)\|_2^2 dt + \frac{\eta \eta_t^2}{2}(L_0 + L_1\|\nabla f(X_t)\|)\|\nabla f(X_t)\|^2 dt \tag{91}$$

$$+ \mathcal{O}(\text{Noise}) + (\eta_t)^2 \frac{\eta(\overline{\sigma_0^2} + \overline{\sigma_1^2}\|\nabla f(X_t)\|_2^2) d(L_0 + L_1\|\nabla f(X_t)\|)}{2N} dt \tag{92}$$

$$= -\eta_t\|\nabla f(X_t)\|_2^2 \left[ 1 - \frac{\eta_t\eta}{2} \left[ (L_0 + L_1\|\nabla f(X_t)\|) \left[ 1 + \frac{d\overline{\sigma_1^2}}{N} \right] + \frac{d\overline{\sigma_0^2}L_1}{N} \right] \right] dt$$

$$+ (\eta_t)^2 \frac{\eta \overline{\sigma_0^2} dL_0}{2N} dt + \mathcal{O}(\text{Noise}). \tag{93}$$

By taking a worst-case scenario approach, we merge these two bounds into a single one:

$$d(f(X_t) - f(X_*)) \leq -\eta_t\|\nabla f(X_t)\|_2^2 \left[ 1 - \frac{\eta_t\eta}{2} \left[ (L_0 + L_1\|\nabla f(X_t)\|) \left[ 1 + \frac{d\overline{\sigma_1^2}}{N} \right] + \frac{d\overline{\sigma_0^2}L_1}{N} \right] \right] dt$$

$$+ (\eta_t)^2 \frac{\eta}{2N}(L_0 + L_1) d\overline{\sigma_0^2} dt + \mathcal{O}(\text{Noise}). \tag{94}$$

With arguments that follow the same steps we detailed in the proof of Theorem D.1, for $0 < \epsilon < 1$, we have that if

$$\eta \eta_t < \frac{2\epsilon}{L_0 + L_1\|\nabla f(X_t)\| + \frac{d}{N} \left( \overline{\sigma_1^2}L_0 + \overline{\sigma_0^2}L_1 + L_1\overline{\sigma_1^2}\|\nabla f(X_t)\|_2 \right)}, \tag{95}$$

by integrating over time and by the Law of the Unconscious Statistician, we have that

$$\mathbb{E}\left[\|\nabla f(X_{\hat{t}})\|_2^2\right] \leq \frac{1}{\phi_t^{(1)}(1 - \epsilon)} \left( f(X_0) - f(X_*) + \frac{\eta \phi_t^{(2)}}{2N}(L_0 + L_1) d\overline{\sigma_0^2} \right) \overset{t\to\infty}{\to} 0, \tag{96}$$

where $\hat{t}$, is a random time with distribution $\frac{\eta_{\hat{t}}}{\phi_t^{(1)}}$.

Finally, for practical reasons, we leverage the distributed setting to tighten the requirements on the learning rate scheduler to make it experimentally viable, and rather require

$$\eta \eta_t < \frac{2\epsilon}{L_0 + L_1\mathbb{E}\left[\|\nabla f(X_t)\|\right] + \frac{d}{N} \left( \overline{\sigma_1^2}L_0 + \overline{\sigma_0^2}L_1 + L_1\overline{\sigma_1^2}\mathbb{E}\left[\|\nabla f(X_t)\|\right] \right)}. \tag{97}$$

$\square$

*Remark* D.3. This condition compares a (deterministic) step-size schedule to a bound that involves $\mathbb{E}[\|\nabla f(X_t)\|]$, which in turn depends on the (random) training trajectory. As such, it should be understood as a *sufficient* stability criterion that generally cannot be certified *a priori*. Our goal is not to propose an immediately implementable rule, but rather to provide quantitative guidance: the bound makes explicit how the key factors in the dynamics (e.g., curvature, noise, compression) jointly shape the stability region.

## D.2. Distributed Compressed SGD with Unbiased Compression

### D.2.1. FIRST ORDER SDE

The following is the *first*-order SDE model of DCSGD (see Theorem 3.6 in (Compagnoni et al., 2025a)). Let us consider the stochastic process $X_t \in \mathbb{R}^d$ defined as the solution of

$$dX_t = -\nabla f(X_t)dt + \sqrt{\frac{\eta}{N}}\sqrt{\tilde{\Sigma}(X_t)}dW_t, \tag{98}$$

where for $\Phi_{\xi_i, \gamma_i}(x) := \mathcal{C}_{\xi_i}(\nabla f_{\gamma_i}(x)) - \nabla f_{\gamma_i}(x)$

$$\tilde{\Sigma}(x) = \frac{1}{N}\sum_{i=1}^{N}\left(\mathbb{E}_{\xi_i \gamma_i}\left[\Phi_{\xi_i, \gamma_i}(x)\Phi_{\xi_i, \gamma_i}(x)^\top\right] + \Sigma_i(x)\right). \tag{99}$$

**Theorem D.4.** *Let $f$ be $(L_0, L_1)$-smooth, the learning rate scheduler $\eta_t$ such that $\phi_t^{(i)} = \int_0^t (\eta_s)^i ds$, $\phi_t^{(1)} \overset{t\to\infty}{\Rightarrow} \infty$, $\frac{\phi_t^{(2)}}{\phi_t^{(1)}} \overset{t\to\infty}{\Rightarrow} 0$, and $\overline{\sigma^2 \omega} := \frac{1}{N}\sum_{i=1}^N \sigma_i^2 \omega_i$. Then, for $0 < \epsilon < 1$,*

$$\eta\eta_t < \frac{2N\epsilon}{\overline{\omega}L_0 + \left(\overline{\sigma^2}d + d\overline{\sigma^2\omega}\right)L_1 + \overline{\omega}L_1\mathbb{E}\left[\|\nabla f(X_t)\|_2\right]}, \tag{100}$$

*and for a random time $\hat{t}$ with distribution $\frac{\eta_t}{\phi_t^{(1)}}$, we have that*

$$\mathbb{E}\left[\|\nabla f(X_{\hat{t}})\|_2^2\right] \le \frac{1}{\phi_t^{(1)}(1-\epsilon)}\left(f(X_0) - f(X_*) + \phi_t^{(2)}\frac{\eta(L_0+L_1)d\left(\overline{\sigma^2}+\overline{\sigma^2\omega}\right)}{2N}\right) \overset{t\to\infty}{\Rightarrow} 0. \tag{101}$$

*Proof.* Since it holds that

$$\mathbb{E}_{\xi_i, \gamma_i}\|(\mathcal{C}_{\xi_i}(\nabla f_{\gamma_i}(x)) - \nabla f(x))\|_2^2 \le \omega_i\|\nabla f(x)\|_2^2 + d\sigma_i^2(\omega_i + 1),$$

we have that for $\mathcal{O}(\text{Noise}) = \sqrt{\tilde{\Sigma}(X_t)}\nabla f(X_t)dW_t$,

$$d(f(X_t) - f(X_*)) = -\eta_t\|\nabla f(X_t)\|_2^2 dt + \mathcal{O}(\text{Noise}) \tag{102}$$

$$+ (\eta_t)^2\frac{\eta(L_0+L_1\|\nabla f(X_t)\|_2)}{2N}\left(\frac{1}{N}\sum_{i=1}^N \mathbb{E}_{\xi_i, \gamma_i}\|(\mathcal{C}_{\xi_i}(\nabla f_{\gamma_i}(x)) - \nabla f(x))\|_2^2\right)dt \tag{103}$$

$$\le -\eta_t\|\nabla f(X_t)\|_2^2 dt + \mathcal{O}(\text{Noise}) \tag{104}$$

$$+ (\eta_t)^2\frac{\eta(L_0+L_1\|\nabla f(X_t)\|_2)}{2N}\left(\overline{\omega}\|\nabla f(X_t)\|_2^2 + \overline{\sigma^2}d + d\overline{\sigma^2\omega}\right)dt. \tag{105}$$

**Phase 1:** If $\|\nabla f(X_t)\|_2 \le 1$, then we have that

$$d(f(X_t) - f(X_*)) \le -\|\nabla f(X_t)\|_2^2\left(\eta_t - \frac{\eta(L_0+L_1)\overline{\omega}}{2N}(\eta_t)^2\right)dt \tag{106}$$

$$+ (\eta_t)^2\frac{\eta(L_0+L_1)d}{2N}\left(\overline{\sigma^2}+\overline{\sigma^2\omega}\right)dt + \mathcal{O}(\text{Noise}). \tag{107}$$

**Phase 2:** If $\|\nabla f(X_t)\|_2 > 1$, we have that

$$d(f(X_t) - f(X_*)) \le -\eta_t\|\nabla f(X_t)\|_2^2 dt + \mathcal{O}(\text{Noise}) \tag{108}$$

$$+ (\eta_t)^2\frac{\eta(L_0+L_1\|\nabla f(X_t)\|_2)}{2N}\left(\overline{\omega}\|\nabla f(X_t)\|_2^2 + \overline{\sigma^2}d + d\overline{\sigma^2\omega}\right)dt \tag{109}$$

$$\le -\eta_t\|\nabla f(X_t)\|_2^2\left(1 - \frac{\eta_t\eta}{2N}\left(\overline{\omega}L_0 + d\left(\overline{\sigma^2}+\overline{\sigma^2\omega}\right)L_1 + \overline{\omega}L_1\|\nabla f(X_t)\|_2\right)\right)dt \tag{110}$$

$$+ \eta_t^2\frac{\eta L_0 d}{2N}\left(\overline{\sigma^2}+\overline{\sigma^2\omega}\right)dt + \mathcal{O}(\text{Noise}). \tag{111}$$

By taking a worst-case scenario approach, we merge these two bounds into a single one. With arguments that follow the same steps we detailed in the proof of Theorem D.1, we have that for $0 < \epsilon < 1$, we have that if

$$\eta\eta_t < \frac{2N\epsilon}{\overline{\omega}L_0 + d\left(\overline{\sigma^2} + \overline{\sigma^2\omega}\right)L_1 + \overline{\omega}L_1\|\nabla f(X_t)\|_2}, \tag{112}$$

by integrating over time and by the Law of the Unconscious Statistician, we have that

$$\mathbb{E}\left[\|\nabla f(X_{\hat{t}})\|_2^2\right] \leq \frac{1}{\phi_t^{(1)}(1-\epsilon)}\left(f(X_0) - f(X_*) + \phi_t^{(2)}\frac{\eta(L_0 + L_1)d\left(\overline{\sigma^2} + \overline{\sigma^2\omega}\right)}{2N}\right) \overset{t\to\infty}{\Rightarrow} 0, \tag{113}$$

where $\hat{t}$, is a random time with distribution $\frac{\eta_{\hat{t}}}{\phi_t^{(1)}}$.

Finally, for practical reasons, we leverage the distributed setting to tighten the requirements on the learning rate scheduler to make it experimentally viable, and rather require

$$\eta\eta_t < \frac{2N\epsilon}{\overline{\omega}L_0 + \left(\overline{\sigma^2}d + d\overline{\sigma^2\omega}\right)L_1 + \overline{\omega}L_1\mathbb{E}\left[\|\nabla f(X_t)\|_2\right]}. \tag{114}$$

$$\square$$

Finally, one can generalize this result to cover the $(\sigma_0^2, \sigma_1^2)$-Variance.

**Theorem D.5.** *Let $f$ be $(L_0, L_1)$-smooth, $\max(\Sigma_i(x)) < \sigma_{i,0}^2 + \sigma_{i,1}^2\|\nabla f(x)\|_2^2$, the learning rate scheduler $\eta_t$ such that $\phi_t^{(i)} = \int_0^t (\eta_s)^i ds$, $\phi_t^{(1)} \overset{t\to\infty}{\to} \infty$, $\frac{\phi_t^{(2)}}{\phi_t^{(1)}} \overset{t\to\infty}{\to} 0$, $\overline{\sigma_0^2} := \frac{1}{N}\sum_{i=1}^N \sigma_{0,i}^2$, $\overline{\sigma_1^2} := \frac{1}{N}\sum_{i=1}^N \sigma_{1,i}^2$, $\overline{\sigma_0^2\omega} := \frac{1}{N}\sum_{i=1}^N \sigma_{i,0}^2\omega_i$, and $\overline{\sigma_1^2\omega} := \frac{1}{N}\sum_{i=1}^N \sigma_{i,1}^2\omega_i$. Then, for $0 < \epsilon < 1$,*

$$\eta\eta_t < \frac{2N\epsilon}{L_0(\overline{\omega} + d(\overline{\sigma_1^2\omega} + \overline{\sigma_1^2})) + L_1 d\left(\overline{\sigma_0^2} + \overline{\sigma_0^2\omega}\right) + L_1(\overline{\omega} + d(\overline{\sigma_1^2\omega} + \overline{\sigma_1^2}))\mathbb{E}\left[\|\nabla f(X_t)\|_2\right]}, \tag{115}$$

*and for a random time $\hat{t}$ with distribution $\frac{\eta_t}{\phi_t^{(1)}}$, we have that*

$$\mathbb{E}\left[\|\nabla f(X_{\hat{t}})\|_2^2\right] \leq \frac{1}{(1-\epsilon)\phi_t^{(1)}}\left(f(X_0) - f(X_*) + \phi_t^{(2)}\frac{L_0(\overline{\omega} + d(\overline{\sigma_1^2\omega} + \overline{\sigma_1^2})) + L_1 d\left(\overline{\sigma_0^2} + \overline{\sigma_0^2\omega}\right)}{2N}\right) \overset{t\to\infty}{\Rightarrow} 0. \tag{116}$$

### D.2.2. OUR NEW FIRST-ORDER SDE FOR DCSGD

The following is the *first*-order SDE model of DCSGD and is a straightforward generalization of Theorem 3.6 in (Compagnoni et al., 2025a) and Remark C.2. Let us consider the stochastic process $X_t \in \mathbb{R}^d$ defined as the solution of

$$dX_t = -\nabla f(X_t)dt + \frac{\eta}{2}\nabla^2 f(X_t)\nabla f(X_t)dt + \sqrt{\frac{\eta}{N}}\sqrt{\tilde{\Sigma}(X_t)}dW_t, \tag{117}$$

where for $\Phi_{\xi_i,\gamma_i}(x) := \mathcal{C}_{\xi_i}\left(\nabla f_{\gamma_i}(x)\right) - \nabla f_{\gamma_i}(x)$

$$\tilde{\Sigma}(x) = \frac{1}{N}\sum_{i=1}^N \left(\mathbb{E}_{\xi_i\gamma_i}\left[\Phi_{\xi_i,\gamma_i}(x)\Phi_{\xi_i,\gamma_i}(x)^\top\right] + \Sigma_i(x)\right). \tag{118}$$

**Theorem D.6.** *Let $f$ be $(L_0, L_1)$-smooth, the learning rate scheduler $\eta_t$ such that $\phi_t^{(i)} = \int_0^t (\eta_s)^i ds$, $\phi_t^{(1)} \overset{t\to\infty}{\to} \infty$, $\frac{\phi_t^{(2)}}{\phi_t^{(1)}} \overset{t\to\infty}{\to} 0$, and $\overline{\sigma^2\omega} := \frac{1}{N}\sum_{i=1}^N \sigma_i^2\omega_i$. Then, for $0 < \epsilon < 1$,*

$$\eta\eta_t < \frac{2\epsilon}{L_0 + L_1\mathbb{E}\left[\|\nabla f(X_t)\|_2\right] + \frac{\overline{\omega}L_0 + d\left(\overline{\sigma^2} + \overline{\sigma^2\omega}\right)L_1 + \overline{\omega}L_1\mathbb{E}\left[\|\nabla f(X_t)\|_2\right]}{N}}, \tag{119}$$

*and for a random time $\hat{t}$ with distribution $\frac{\eta_t}{\phi_t^{(1)}}$, we have that*

$$\mathbb{E}\left[\|\nabla f(X_{\hat{t}})\|_2^2\right] \leq \frac{1}{\phi_t^{(1)}(1-\epsilon)}\left(f(X_0) - f(X_*) + \phi_t^{(2)}\frac{\eta(L_0 + L_1)d}{2N}\left(\overline{\sigma^2} + \overline{\sigma^2\omega}\right)\right) \overset{t\to\infty}{\Rightarrow} 0. \tag{120}$$

*Proof.* Since it holds that

$$\mathbb{E}_{\xi_i, \gamma_i} \|(\mathcal{C}_{\xi_i}(\nabla f_{\gamma_i}(x)) - \nabla f(x))\|_2^2 \leq \omega_i \|\nabla f(x)\|_2^2 + d\sigma_i^2(\omega_i + 1),$$

we have that for $\mathcal{O}(\text{Noise}) = \sqrt{\tilde{\Sigma}(X_t)} \nabla f(X_t) dW_t$,

$$d(f(X_t) - f(X_*)) = -\eta_t \|\nabla f(X_t)\|_2^2 dt + \frac{\eta \eta_t^2}{2} (\nabla f(X_t))^\top \nabla^2 f(X_t) \nabla f(X_t) dt + \mathcal{O}(\text{Noise}) \tag{121}$$

$$+ \frac{\eta \eta_t^2}{2} \frac{(L_0 + L_1 \|\nabla f(X_t)\|_2)}{N} \left( \frac{1}{N} \sum_{i=1}^N \mathbb{E}_{\xi_i, \gamma_i} \|(\mathcal{C}_{\xi_i}(\nabla f_{\gamma_i}(x)) - \nabla f(x))\|_2^2 \right) dt \tag{122}$$

$$\leq -\eta_t \|\nabla f(X_t)\|_2^2 dt + \frac{\eta \eta_t^2}{2} (L_0 + L_1 \|\nabla f(X_t)\|) \|\nabla f(X_t)\|^2 dt + \mathcal{O}(\text{Noise}) \tag{123}$$

$$+ \frac{\eta \eta_t^2}{2} \frac{(L_0 + L_1 \|\nabla f(X_t)\|_2)}{N} \left( \overline{\omega} \|\nabla f(X_t)\|_2^2 + \overline{\sigma^2} d + d\overline{\sigma^2 \omega} \right) dt. \tag{124}$$

**Phase 1:** If $\|\nabla f(X_t)\|_2 \leq 1$, then we have that

$$d(f(X_t) - f(X_*)) \leq -\|\nabla f(X_t)\|_2^2 \left( \eta_t - \frac{\eta_t^2 \eta}{2} (L_0 + L_1) \left( 1 + \frac{\overline{\omega}}{N} \right) \right) dt \tag{125}$$

$$+ (\eta_t)^2 \frac{\eta(L_0 + L_1)d}{2N} \left( \overline{\sigma^2} + \overline{\sigma^2 \omega} \right) dt + \mathcal{O}(\text{Noise}). \tag{126}$$

**Phase 2:** If $\|\nabla f(X_t)\|_2 > 1$, we have that

$$d(f(X_t) - f(X_*)) \leq -\eta_t \|\nabla f(X_t)\|_2^2 dt + \frac{\eta \eta_t^2}{2} (L_0 + L_1 \|\nabla f(X_t)\|) \|\nabla f(X_t)\|^2 dt + \mathcal{O}(\text{Noise}) \tag{127}$$

$$+ (\eta_t)^2 \frac{\eta(L_0 + L_1 \|\nabla f(X_t)\|_2)}{2N} \left( \overline{\omega} \|\nabla f(X_t)\|_2^2 + \overline{\sigma^2} d + d\overline{\sigma^2 \omega} \right) dt \tag{128}$$

$$\leq -\eta_t \|\nabla f(X_t)\|_2^2 \left[ 1 - \frac{\eta_t \eta}{2} \left[ (L_0 + L_1 \|\nabla f(X_t)\|_2) \left[ 1 + \frac{\overline{\omega}}{N} \right] + \frac{d \left( \overline{\sigma^2} + \overline{\sigma^2 \omega} \right) L_1}{N} \right] \right]$$

$$+ \eta_t^2 \frac{\eta L_0 d}{2N} \left( \overline{\sigma^2} + \overline{\sigma^2 \omega} \right) + \mathcal{O}(\text{Noise}). \tag{129}$$

By taking a worst-case scenario approach, we merge these two bounds into a single one. With arguments that follow the same steps we detailed in the proof of Theorem D.1, we have that for $0 < \epsilon < 1$, we have that if

$$\eta \eta_t < \frac{2\epsilon}{L_0 + L_1 \|\nabla f(X_t)\|_2 + \frac{\overline{\omega} L_0 + d(\overline{\sigma^2} + \overline{\sigma^2 \omega}) L_1 + \overline{\omega} L_1 \|\nabla f(X_t)\|_2}{N}}, \tag{130}$$

by integrating over time and by the Law of the Unconscious Statistician, we have that

$$\mathbb{E}\left[ \|\nabla f(X_{\hat{t}})\|_2^2 \right] \leq \frac{1}{\phi_t^{(1)}(1 - \epsilon)} \left( f(X_0) - f(X_*) + \phi_t^{(2)} \frac{\eta(L_0 + L_1)d}{2N} \left( \overline{\sigma^2} + \overline{\sigma^2 \omega} \right) \right) \overset{t \to \infty}{\to} 0, \tag{131}$$

where $\hat{t}$, is a random time with distribution $\frac{\eta_{\hat{t}}}{\phi_t^{(1)}}$.

Finally, for practical reasons, we leverage the distributed setting to tighten the requirements on the learning rate scheduler to make it experimentally viable, and rather require

$$\eta \eta_t < \frac{2\epsilon}{L_0 + L_1 \mathbb{E}\left[ \|\nabla f(X_t)\|_2 \right] + \frac{\overline{\omega} L_0 + d(\overline{\sigma^2} + \overline{\sigma^2 \omega}) L_1 + \overline{\omega} L_1 \mathbb{E}[\|\nabla f(X_t)\|_2]}{N}}. \tag{132}$$

$\square$

Finally, one can generalize this result to cover the $(\sigma_0^2, \sigma_1^2)$-Variance.

**Theorem D.7.** *Let $f$ be $(L_0, L_1)$-smooth, $\max(\Sigma_i(x)) < \sigma_{i,0}^2 + \sigma_{i,1}^2 \|\nabla f(x)\|_2^2$, the learning rate scheduler $\eta_t$ such that $\phi_t^{(i)} = \int_0^t (\eta_s)^i ds$, $\phi_t^{(1)} \overset{t \to \infty}{\to} \infty$, $\frac{\phi_t^{(2)}}{\phi_t^{(1)}} \overset{t \to \infty}{\to} 0$, $\overline{\sigma_0^2} := \frac{1}{N} \sum_{i=1}^N \sigma_{0,i}^2$, $\overline{\sigma_1^2} := \frac{1}{N} \sum_{i=1}^N \sigma_{1,i}^2$, $\overline{\sigma_0^2 \omega} := \frac{1}{N} \sum_{i=1}^N \sigma_{i,0}^2 \omega_i$, and $\overline{\sigma_1^2 \omega} := \frac{1}{N} \sum_{i=1}^N \sigma_{i,1}^2 \omega_i$. Then, for $0 < \epsilon < 1$,*

$$\eta \eta_t < \frac{2\epsilon}{L_0 + L_1 \mathbb{E}\left[\|\nabla f(X_t)\|_2\right] + \frac{L_0(\overline{\omega} + d(\overline{\sigma_1^2 \omega} + \overline{\sigma_1^2})) + L_1 d\left(\overline{\sigma_0^2} + \overline{\sigma_0^2 \omega}\right) + L_1(\overline{\omega} + d(\overline{\sigma_1^2 \omega} + \overline{\sigma_1^2}))\mathbb{E}[\|\nabla f(X_t)\|_2]}{N}}, \tag{133}$$

*and for a random time $\hat{t}$ with distribution $\frac{\eta_t}{\phi_t^{(1)}}$, we have that*

$$\mathbb{E}\left[\|\nabla f(X_{\hat{t}})\|_2^2\right] \leq \frac{1}{(1-\epsilon)\phi_t^{(1)}} \left(f(X_0) - f(X_*) + \phi_t^{(2)} \frac{\eta(L_0 + L_1)d(\overline{\sigma_0^2} + \overline{\sigma_0^2 \omega})}{2N}\right) \overset{t \to \infty}{\to} 0. \tag{134}$$

### D.3. Distributed SignSGD

#### D.3.1. FIRST ORDER SDE

The following is the *first*-order SDE model of DSignSGD (see Theorem 3.10 in (Compagnoni et al., 2025a)). Let us consider the stochastic process $X_t \in \mathbb{R}^d$ defined as the solution of

$$dX_t = -\frac{1}{N} \sum_{i=1}^N (1 - 2\mathbb{P}(\nabla f_{\gamma_i}(X_t) < 0)) \, dt + \sqrt{\frac{\eta}{N}} \sqrt{\overline{\Sigma}(X_t)} dW_t. \tag{135}$$

where

$$\overline{\Sigma}(X_t) := \frac{1}{N} \sum_{i=1}^N \overline{\Sigma_i}(X_t), \tag{136}$$

and $\overline{\Sigma_i}(x) = \mathbb{E}[\xi_{\gamma_i}(x)\xi_{\gamma_i}(x)^\top]$ where $\xi_{\gamma_i}(x) := \text{sign}(\nabla f_{\gamma_i}(x)) - 1 + 2\mathbb{P}(\nabla f_{\gamma_i}(x) < 0)$ the noise in the sample sign $(\nabla f_{\gamma_i}(x))$.

**Corollary D.8** (Corollary C.10 in (Compagnoni et al., 2025a)). *If the stochastic gradients are $\nabla f_{\gamma_i}(x) = \nabla f(x) + \sqrt{\Sigma_i} Z_i$ such that $Z_i \sim t_\nu(0, I_d)$ does not depend on $x$, $\nu$ are the degrees of freedom, and scale matrices $\Sigma_i = \text{diag}(\sigma_{1,i}^2, \cdots, \sigma_{d,i}^2)$. Then, the SDE of DSignSGD is*

$$dX_t = -\frac{2}{N} \sum_{i=1}^N \Xi_\nu \left(\Sigma_i^{-\frac{1}{2}} \nabla f(X_t)\right) dt + \sqrt{\frac{\eta}{N}} \sqrt{\tilde{\Sigma}(X_t)} dW_t. \tag{137}$$

*where $\Xi_\nu(x)$ is defined as $\Xi_\nu(x) := x \frac{\Gamma\left(\frac{\nu+1}{2}\right)}{\sqrt{\pi\nu}\Gamma\left(\frac{\nu}{2}\right)} {}_2F_1 \left(\frac{1}{2}, \frac{\nu+1}{2}; \frac{3}{2}; -\frac{x^2}{\nu}\right)$, ${}_2F_1(a, b; c; x)$ is the hypergeometric function, and*

$$\tilde{\Sigma}(X_t) := I_d - \frac{4}{N} \sum_{i=1}^N \left(\Xi_\nu \left(\Sigma_i^{-\frac{1}{2}} \nabla f(X_t)\right)\right)^2. \tag{138}$$

**Remark (Beyond Student's $t$ noise).** We specialize our analysis to the Student's $t$ noise to obtain a closed-form expression for the scalar function $\Xi_\nu$ governing the drift of the DSignSGD surrogate (and hence explicit constants $M_\nu, \ell_\nu$). More generally, the same SDE construction applies to any coordinate-wise symmetric noise with an absolutely continuous density: the drift depends on the one-dimensional CDF through $\Xi(x) := F(x) - \frac{1}{2}$, and the constants in the stability bound can be defined as $M := \sup_x \Xi'(x)$ and $\ell := 2\Xi'(0)$ whenever these quantities are finite. The Student's $t$ family is a convenient heavy-tailed instantiation that also covers regimes with undefined mean (e.g., $\nu \leq 1$).

**Theorem D.9.** *Let $f$ be $(L_0, L_1)$-smooth, $\eta_t$ a learning rate scheduler such that $\phi_t^{(i)} = \int_0^t (\eta_s)^i ds$, $\phi_t^{(1)} \overset{t \to \infty}{\to} \infty$, $\frac{\phi_t^{(2)}}{\phi_t^{(1)}} \overset{t \to \infty}{\to} 0$, $\Sigma_i \leq \sigma_{max,i}^2$, $\sigma_{\mathcal{H},1}$ be the harmonic mean of $\{\sigma_{max,i}\}$, and $\ell_\nu := 2\Xi_\nu'(0) > 0$ a constant. Then, for a scheduler $\eta \eta_t < \frac{2N\ell_\nu}{\sigma_{\mathcal{H},1} d L_1}$ and a random time $\tilde{t}$ with distribution $\frac{\eta_t \ell_\nu \sigma_{\mathcal{H},1}^{-1} - \eta_t^2 \frac{\eta L_1 d}{2N}}{\phi_t^{(1)} \ell_\nu \sigma_{\mathcal{H},1}^{-1} - \phi_t^{(2)} \frac{\eta L_1 d}{2N}}$, we have that*

$$\mathbb{E}\|\nabla f(X_{\tilde{t}})\|_2^2 \leq \frac{1}{\phi_t^{(1)} \ell_\nu \sigma_{\mathcal{H},1}^{-1} - \phi_t^{(2)} \frac{\eta L_1 d}{2N}} \left(f(X_0) - f(X_*) + \frac{\eta(L_0 + L_1)d\phi_t^{(2)}}{2N}\right) \overset{t \to \infty}{\to} 0. \tag{139}$$

*Proof.* By Itô Lemma on $f(X_t) - f(X_*)$, we have that for $\mathcal{O}(\text{Noise}) = \sqrt{\Sigma(X_t)}\nabla f(X_t)dW_t$,

$$d(f(X_t) - f(X_*)) \leq -\ell_\nu \sigma_{\mathcal{H},1}^{-1}\eta_t\|\nabla f(X_t)\|_2^2 dt + \frac{\eta\eta_t^2 d}{2N}(L_0 + L_1\|\nabla f(X_t)\|_2)dt + \mathcal{O}(\text{Noise}) \tag{140}$$

**Phase 1:** $\|\nabla f(X_t)\|_2 \leq 1$:

$$d(f(X_t) - f(X_*)) \leq -\ell_\nu \sigma_{\mathcal{H},1}^{-1}\eta_t\|\nabla f(X_t)\|_2^2 dt + \frac{\eta\eta_t^2 d}{2N}(L_0 + L_1)dt + \mathcal{O}(\text{Noise}). \tag{141}$$

**Phase 2:** $\|\nabla f(X_t)\|_2 > 1$:

$$d(f(X_t) - f(X_*)) \leq -\ell_\nu \sigma_{\mathcal{H},1}^{-1}\eta_t\|\nabla f(X_t)\|_2^2 dt + \frac{\eta\eta_t^2 dL_1\|\nabla f(X_t)\|_2^2}{2N} + \frac{\eta\eta_t^2 dL_0}{2N}dt + \mathcal{O}(\text{Noise}). \tag{142}$$

By taking the worst case of these two phases, we have that

$$d(f(X_t) - f(X_*)) \leq -\ell_\nu \sigma_{\mathcal{H},1}^{-1}\eta_t\|\nabla f(X_t)\|_2^2 dt + \frac{\eta\eta_t^2 dL_1\|\nabla f(X_t)\|_2^2}{2N}dt + \frac{\eta\eta_t^2 d}{2N}(L_0 + L_1)dt + \mathcal{O}(\text{Noise}). \tag{143}$$

With arguments that follow the same steps we detailed in the proof of Theorem D.1, we have that

$$\mathbb{E}\|\nabla f(X_{\tilde{t}})\|_2^2 \leq \frac{1}{\phi_t^{(1)}\ell_\nu \sigma_{\mathcal{H},1}^{-1} - \phi_t^{(2)}\frac{d\eta L_1}{2N}}\left(f(X_0) - f(X_*) + \frac{\eta(L_0 + L_1)d\phi_t^{(2)}}{2N}\right) \overset{t\to\infty}{\to} 0. \tag{144}$$

$\square$

### D.3.2. OUR NEW FIRST-ORDER SDE FOR DSIGNSGD

The following is the *first*-order SDE model of DSignSGD and is a straightforward generalization of Corollary C.10 in (Compagnoni et al., 2025a) and Remark C.2. We observe that $\Xi'_\nu(x)$ is bounded by the positive finite constant $M_\nu$.

$$dX_t = -\frac{2}{N}\sum_{i=1}^{N}\Xi_\nu\left(\Sigma_i^{-\frac{1}{2}}\nabla f(X_t)\right)dt$$

$$+ \frac{\eta}{N}\sum_{i=1}^{N}\Sigma_i^{-\frac{1}{2}}\nabla^2 f(X_t)\left(\Xi'_\nu\left(\Sigma_i^{-\frac{1}{2}}\nabla f(X_t)\right) \circ \Xi_\nu\left(\Sigma_i^{-\frac{1}{2}}\nabla f(X_t)\right)\right)dt$$

$$+ \sqrt{\frac{\eta}{N}}\sqrt{\tilde{\Sigma}(X_t)}dW_t. \tag{145}$$

**Theorem D.10.** *Let $f$ be $(L_0, L_1)$-smooth, $\Sigma_i \leq \sigma_{max,i}^2$, $\sigma_{\mathcal{H},1}$ be the harmonic mean of $\{\sigma_{max,i}\}$, $M_\nu := \sup\{\Xi'_\nu(x)\} > 0$ and $\ell_\nu := 2\Xi'_\nu(0) > 0$ constants, and $K := \left(\frac{L_1}{2N} + \frac{(L_0+L_1)\sigma_{\mathcal{H},1}^{-1}M_\nu}{\sqrt{d}}\right)$. Then, for a scheduler $\eta\eta_t < \frac{\ell_\nu K^{-1}}{\sigma_{\mathcal{H},1}d}$ and a random time $\tilde{t}$ with distribution $\frac{\eta_t\ell_\nu \sigma_{\mathcal{H},1}^{-1} - \eta_t^2 K}{\phi_t^{(1)}\ell_\nu \sigma_{\mathcal{H},1}^{-1} - \phi_t^{(2)}K}$, we have that*

$$\mathbb{E}\|\nabla f(X_{\tilde{t}})\|_2^2 \leq \frac{1}{\phi_t^{(1)}\ell_\nu \sigma_{\mathcal{H},1}^{-1} - \phi_t^{(2)}K}\left(f(X_0) - f(X_*) + \phi_t^{(2)}\eta(L_0 + L_1)d\left(\frac{1}{2N} + \frac{M_\nu}{\sigma_{\mathcal{H},1}\sqrt{d}}\right)\right) \overset{t\to\infty}{\to} 0. \tag{146}$$

*Proof.* By Itô Lemma on $f(X_t) - f(X_*)$, we have that for $\mathcal{O}(\text{Noise}) = \sqrt{\tilde{\Sigma}(X_t)}\nabla f(X_t)dW_t$,

$$d(f(X_t) - f(X_*)) \leq -\ell_\nu \sigma_{\mathcal{H},1}^{-1}\eta_t\|\nabla f(X_t)\|_2^2 dt + \eta\eta_t^2\sigma_{\mathcal{H},1}^{-1}(L_0 + L_1\|\nabla f(X_t)\|_2)M_\nu\|\nabla f(X_t)\|_1 dt \tag{147}$$

$$+ \frac{\eta\eta_t^2 d}{2N}(L_0 + L_1\|\nabla f(X_t)\|_2)dt + \mathcal{O}(\text{Noise}). \tag{148}$$

**Phase 1:** $\|\nabla f(X_t)\|_2 \leq 1$:

$$d(f(X_t) - f(X_*)) \leq -\ell_\nu \sigma_{\mathcal{H},1}^{-1}\eta_t\|\nabla f(X_t)\|_2^2 dt + \eta\eta_t^2\sigma_{\mathcal{H},1}^{-1}(L_0 + L_1)M_\nu\sqrt{d}dt \tag{149}$$

$$+ \frac{\eta\eta_t^2 d}{2N}(L_0 + L_1)dt + \mathcal{O}(\text{Noise}). \tag{150}$$

**Phase 2:** $\|\nabla f(X_t)\|_2 > 1$: Since $\|\nabla f(X_t)\|_1 < \sqrt{d}\|\nabla f(X_t)\|_2 < \sqrt{d}\|\nabla f(X_t)\|_2^2$, we have that

$$d(f(X_t) - f(X_*)) \leq -\ell_\nu \sigma_{\mathcal{H},1}^{-1} \eta_t \|\nabla f(X_t)\|_2^2 dt + \eta \eta_t^2 \sigma_{\mathcal{H},1}^{-1} (L_0 + L_1) M_\nu \sqrt{d} \|\nabla f(X_t)\|_2^2 dt \tag{151}$$

$$+ \frac{\eta \eta_t^2 d L_1 \|\nabla f(X_t)\|_2^2}{2N} + \frac{\eta \eta_t^2 d L_0}{2N} dt + \mathcal{O}(\text{Noise}). \tag{152}$$

By taking the worst case of these two phases, we have that

$$d(f(X_t) - f(X_*)) \leq -\ell_\nu \sigma_{\mathcal{H},1}^{-1} \eta_t \|\nabla f(X_t)\|_2^2 dt + \eta \eta_t^2 \sigma_{\mathcal{H},1}^{-1} (L_0 + L_1) M_\nu \sqrt{d} \|\nabla f(X_t)\|_2^2 dt \tag{153}$$

$$+ \frac{\eta \eta_t^2 d L_1 \|\nabla f(X_t)\|_2^2}{2N} dt + \eta \eta_t^2 (L_0 + L_1) d \left( \frac{1}{2N} + \frac{M_\nu}{\sigma_{\mathcal{H},1} \sqrt{d}} \right) dt + \mathcal{O}(\text{Noise}). \tag{154}$$

With arguments that follow the same steps we detailed in the proof of Theorem D.1, we have that

$$\mathbb{E}\|\nabla f(X_{\tilde{t}})\|_2^2 \leq \frac{1}{\phi_t^{(1)} \ell_\nu \sigma_{\mathcal{H},1}^{-1} - \phi_t^{(2)} K} \left( f(X_0) - f(X_*) + \phi_t^{(2)} \eta (L_0 + L_1) d \left( \frac{1}{2N} + \frac{M_\nu}{\sigma_{\mathcal{H},1} \sqrt{d}} \right) \right) \overset{t \to \infty}{\to} 0. \tag{155}$$

$\square$

## D.4. Limitations

Our analysis focuses on *homogeneous* client distributions to isolate the effects of noise, compression, and adaptivity without the additional complexity of data heterogeneity. Our implementation-oriented normalization discussion also assumes a server-aggregated topology, in which a central server collects client updates and can aggregate a scalar norm estimate. Fully decentralized topologies are outside the scope of this work and would require separate communication and estimation mechanisms. Extending the results to heterogeneous settings—where clients may have different tail indices, variance structures, or asymmetric noise distributions—is an important direction for future work. We also restrict attention to *unbiased* and *signed* gradient compression, while many practical distributed optimizers employ general *biased* compressors or use *error-feedback* mechanisms to recover convergence guarantees. Extending our SDE framework to these settings would require augmenting the continuous-time state with the error-feedback memory, or introducing suitable bias-correction terms in the drift.

Additionally, we focus on finite-sum minimization, as is common in the SDE literature for stochastic optimization (Jastrzebski et al., 2018), and do not tackle questions related to generalization (Smith et al., 2020).

Finally, our contribution is intentionally foundational: rather than proposing new optimizers, we build a rigorous, unified framework that captures the joint effects of noise, compression, and adaptivity for distributed methods under $(L_0, L_1)$-smoothness. We view this work as a basis for future extensions (e.g., heterogeneous clients, error-feedback, and general biased compressors) and for subsequent analyses that further systematize large-scale stochastic optimization.

**Acknowledgments.** We acknowledge the use of OpenAI's ChatGPT as a writing assistant to help us rephrase and refine parts of the manuscript. All technical content, derivations, and scientific contributions remain the sole responsibility of the authors.

## E. Experiments

Our experiments are intentionally minimalistic: They are designed to validate the fidelity of the derived insights and to illustrate the qualitative phenomena predicted by our theory, rather than to benchmark performance on specific tasks. This aligns with the theoretical nature of our contribution.

### E.1. DCSGD - Figure 2 - (Left Column)

We optimize $f(x) = \frac{\sum_{j=1}^{1000} (x_j)^4}{4}$ as we inject Gaussian noise with mean 0 and variance $\sigma^2 \|\nabla f(x)\|_2^2$ on the gradient. The learning rate is $\eta = 0.1$, $\sigma = 0.1$, we use *random sparsification* with $\omega \in \{4, 8, 16\}$, and we average over 1000 runs. In the top figure, we use no scheduler, while in the bottom one we use a scheduler as per Eq. 15.

### E.2. DSignSGD - Figure 2 - (Right Column)

We optimize $f(x) = \frac{x^4}{4}$ as we inject student's t noise with $\nu = 1$ and scale parameters $\sigma$ on the gradient. The learning rate is $\eta = 0.1$, $\sigma \in \{0.25, 0.5, 1, 2, 8, 16\}$, and we average over 10000 runs. In the top figure, we use no scheduler, while in the bottom one we use a scheduler as per Theorem 4.3, e.g. $\eta_t = \frac{1}{\sqrt{t+1}}$.

### E.3. Experiments on a MLP (Figure 1)

**Architecture, data, and metric.** We consider fully connected neural network $h_\theta : \mathbb{R}^{20} \to \mathbb{R}$ with a single hidden layer of width 64 and ReLU activation. The network has the form

$$h_\theta(x) = W_2 \, \phi(W_1 x + b_1) + b_2, \tag{156}$$

where $W_1 \in \mathbb{R}^{64 \times 20}$, $W_2 \in \mathbb{R}^{1 \times 64}$, $b_1 \in \mathbb{R}^{64}$, $b_2 \in \mathbb{R}$, and $\phi$ denotes the ReLU activation. We collect all parameters into a single vector $\theta \in \mathbb{R}^d$. The loss is the mean squared error

$$f(\theta) = \frac{1}{n} \sum_{i=1}^{n} \big(h_\theta(x_i) - y_i\big)^2, \tag{157}$$

with a fixed dataset of $n = 4096$ examples. Inputs are $x_i \in \mathbb{R}^{20}$ sampled as $x_i \sim \mathcal{N}(0, I_{20})$, and labels are generated from a linear teacher

$$y_i = x_i^\top w_\star + \varepsilon_i, \qquad w_\star \sim \mathcal{N}(0, I_{20}), \quad \varepsilon_i \sim \mathcal{N}(0, 0.1^2). \tag{158}$$

The dataset $(x_i, y_i)_{i=1}^{n}$ is sampled once and reused for all methods and repetitions, while the number of clients is $N = 8$. At iteration $t$, we compute the full-batch gradient $g_t = \nabla_\theta f(\theta_t) \in \mathbb{R}^d$ and monitor the quantity

$$\|g_t\|_2^2. \tag{159}$$

For each setting, we approximate the expectation by averaging this quantity over multiple independent runs.

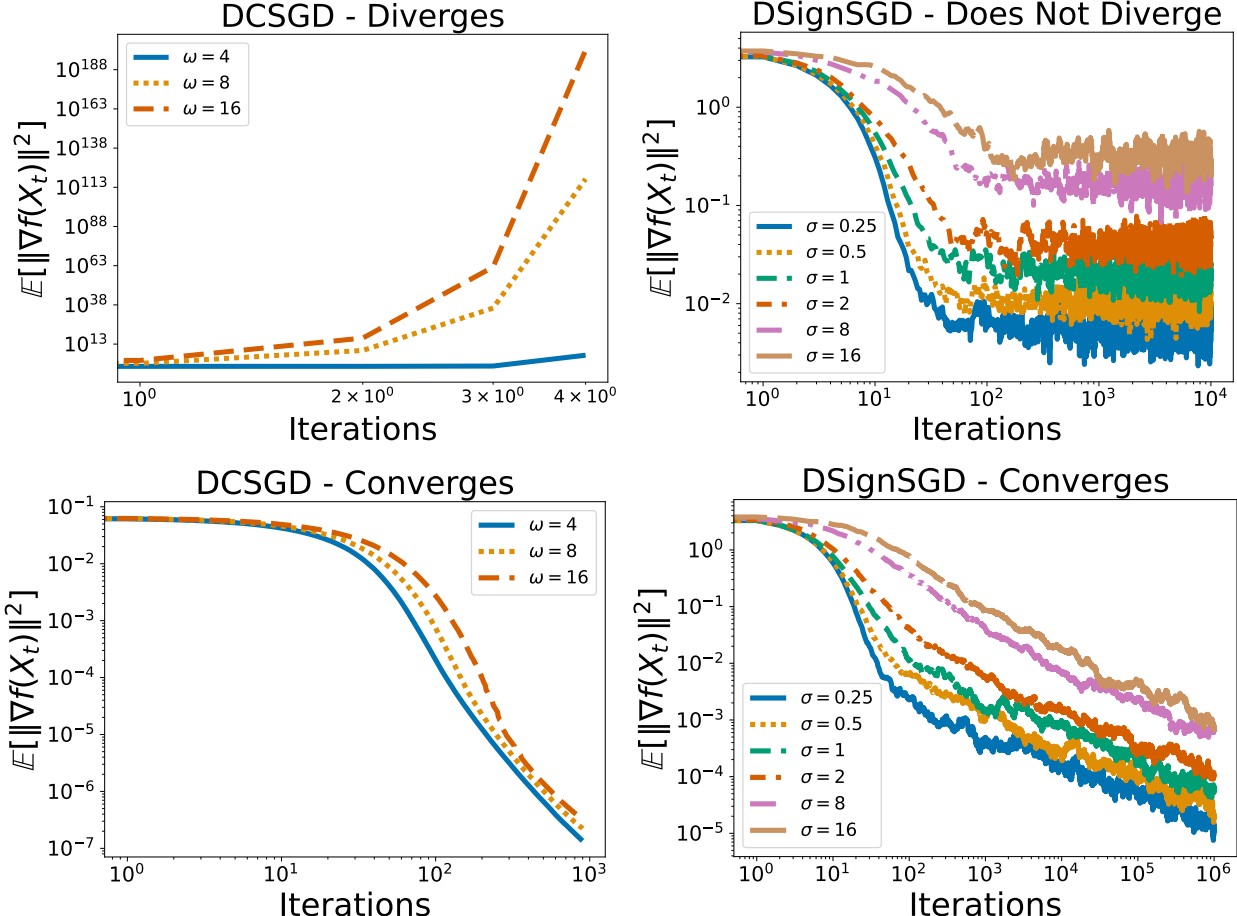

*Figure 2.* We optimize $f(x) = \frac{\sum_{j=1}^{1000}(x_j)^4}{4}$ with batch noise of variance $\sigma^2 \|\nabla f(x)\|_2^2$ and use *Random Sparsification* for different compression rates $\omega$: as per Thm. 4.2, DCSGD diverges faster and faster for larger values of $\omega$ when the normalization proposed in Eq. 15 **is not employed** (Top-Left) but always converges if it **is employed** (Bottom-Left). We optimize $f(x) = \frac{x^4}{4}$ with batch noise of **unbounded expected value** and for different *scale parameters* $\sigma$: DSignSGD does not converge to 0 *without* a proper learning rate scheduler as prescribed by Thm. 4.3 (Top-Right), but does converge *with* (Bottom-Right). See Appendix E for all implementation details.

**Noise injection and compression.** In all experiments, we inject additive Gaussian gradient noise and then apply random sparsification. At each iteration $t$, we sample $Z_t \sim \mathcal{N}(0, \sigma^2 \|g_t\|_2^2 I_d)$ with $\sigma = 0.1$ and form the noisy gradient $g_t + Z_t$. For a given sparsification probability $p \in \{\frac{4}{5}, \frac{8}{9}, \frac{16}{17}\}$ we draw an i.i.d. mask $m_t \in \{0, 1\}^d$ with

$$\mathbb{P}\big[(m_t)_i = 1\big] = 1 - p, \tag{160}$$

and define the unbiased random sparsifier

$$C_p(v) = \frac{v \odot m_t}{1 - p}, \tag{161}$$

so that $\mathbb{E}[C_p(v) \mid v] = v$. In the plots, we simply label these three compression levels as $\omega \in \{4, 8, 16\}$, with larger $\omega$ corresponding to more aggressive sparsification (larger $p$).

**DCSGD without scheduler** Here, we use DCSGD with learning rate $\eta = 0.01$, noise level $\sigma = 0.1$, and sparsification probabilities $p \in \{\frac{4}{5}, \frac{8}{9}, \frac{16}{17}\}$. For each value of $p$, we run $T_{\text{div}} = 10$ iterations and repeat the experiment over $n_{\text{runs}}^{\text{div}} = 100$ independent initializations of the MLP. We report the average of $\|g_t\|_2^2$ over these runs as a function of the iteration index.

**DCSGD with our scheduler** Here, we use DCSGD with learning rate $\eta = 0.01$ and use an adaptive scheduler $\eta_t$ as per Eq. 15 where $\sigma_0 = 0$, and we assume $L_0 = L_1 = 1$, because these constants are not actually known. As before, we use $\sigma = 0.1$ and $p \in \{\frac{4}{5}, \frac{8}{9}, \frac{16}{17}\}$. For each value of $p$, we run $T_{\text{conv}} = 50000$ iterations and repeat the experiment over $n_{\text{runs}}^{\text{conv}} = 5$ independent initializations. We then average $\|g_t\|_2^2$ over these runs.

**Normalized SGD with compression** This experiment provides a baseline where we apply plain Normalized DCSGD, i.e., we normalize the compressed gradient. We use learning rate $\eta = 0.01$, noise level $\sigma = 0.1$, sparsification probabilities $p \in \{\frac{4}{5}, \frac{8}{9}, \frac{16}{17}\}$, and a small constant $\varepsilon = 10^{-8}$ added for numerical stability. The horizon and number of runs are the same as in the convergent DCSGD experiment, that is $T_{\text{conv}} = 50000$ iterations and $n_{\text{runs}}^{\text{conv}} = 5$ independent initializations for each value of $p$. As before, we track and report the averaged trajectories of $\|g_t\|_2^2$.

**DSignSGD** Here, we apply DSignSGD as we inject Student's t noise with $\nu = 1$ and scale parameters $\sigma$ on the gradient. The learning rate is $\eta = 0.01$, $\sigma \in \{0.25, 0.5, 1, 2, 8, 16\}$, and we average over 5 runs. In the top figure, we use no scheduler, while in the bottom one we use a scheduler as per Theorem 4.3, $\eta_t = \frac{1}{\sqrt{t+1}}$. As before, we track and report the averaged trajectories of $\|g_t\|_2^2$.

## E.4. Experiments on ResNet-18 and ViT (Figures 3 and 4)

**Architecture, data, and metric.** We complement the MLP experiments with additional sanity checks on ResNet-18 and a simple ViT models trained on CIFAR-10 in a distributed setting with $N = 8$ clients. CIFAR-10 is split uniformly across clients, and all curves are averaged over 5 seeds. As in the previous section, these experiments are not intended as competitive benchmarks. Their goal is to test whether the qualitative mechanisms predicted by the theory persist beyond the synthetic and MLP settings. At iteration $t$, we monitor the squared norm of the gradient,

$$\|g_t\|_2^2, \tag{162}$$

and report its averaged trajectory across independent runs.

**Noise injection and compression.** For the DCSGD experiments, we use the same noisy compressed-gradient protocol as in the MLP setting, with Gaussian noise level $\sigma = 0.1$ and learning rate $\eta = 0.1$. At each iteration $t$, we inject additive Gaussian gradient noise with affine variance,

$$Z_t \sim \mathcal{N}(0, \sigma^2 \|g_t\|_2^2 I_d), \tag{163}$$

and then apply unbiased random sparsification at compression levels $\omega \in \{4, 8, 16\}$, where larger $\omega$ corresponds to more aggressive sparsification. We compare DCSGD without any scheduler/normalization against DCSGD with the adaptive normalization suggested by Thm. 4.2 through Eq. 15. We also report a baseline that applies plain Normalized SGD under the same noise and compression.

**DCSGD.** Without any scheduler or normalization, DCSGD becomes unstable on both ResNet-18 and ViT, and the divergence worsens as the compression level $\omega$ increases. In contrast, using the adaptive normalization prescribed by Eq. 15 stabilizes training and yields convergence for all tested values of $\omega$. The plain Normalized SGD baseline is also stabilized by normalization, but exhibits a less stable profile in these experiments.

**DSignSGD.** For the DSignSGD experiments, we use learning rate $\eta = 0.001$ and inject Student's t gradient noise with $\nu = 1$ and scale parameters $\sigma \in \{0.25, 0.5, 1, 2, 8, 16\}$. With a constant stepsize, DSignSGD remains stable but does not converge to zero. With the diminishing schedule prescribed by Thm. 4.3, namely $\eta_t = 1/\sqrt{t+1}$, DSignSGD converges across the tested noise scales. These results are consistent with the prediction that DSignSGD inherits an implicit normalization from the sign map, whereas DCSGD requires an explicit noise- and compression-dependent normalization to remain stable.

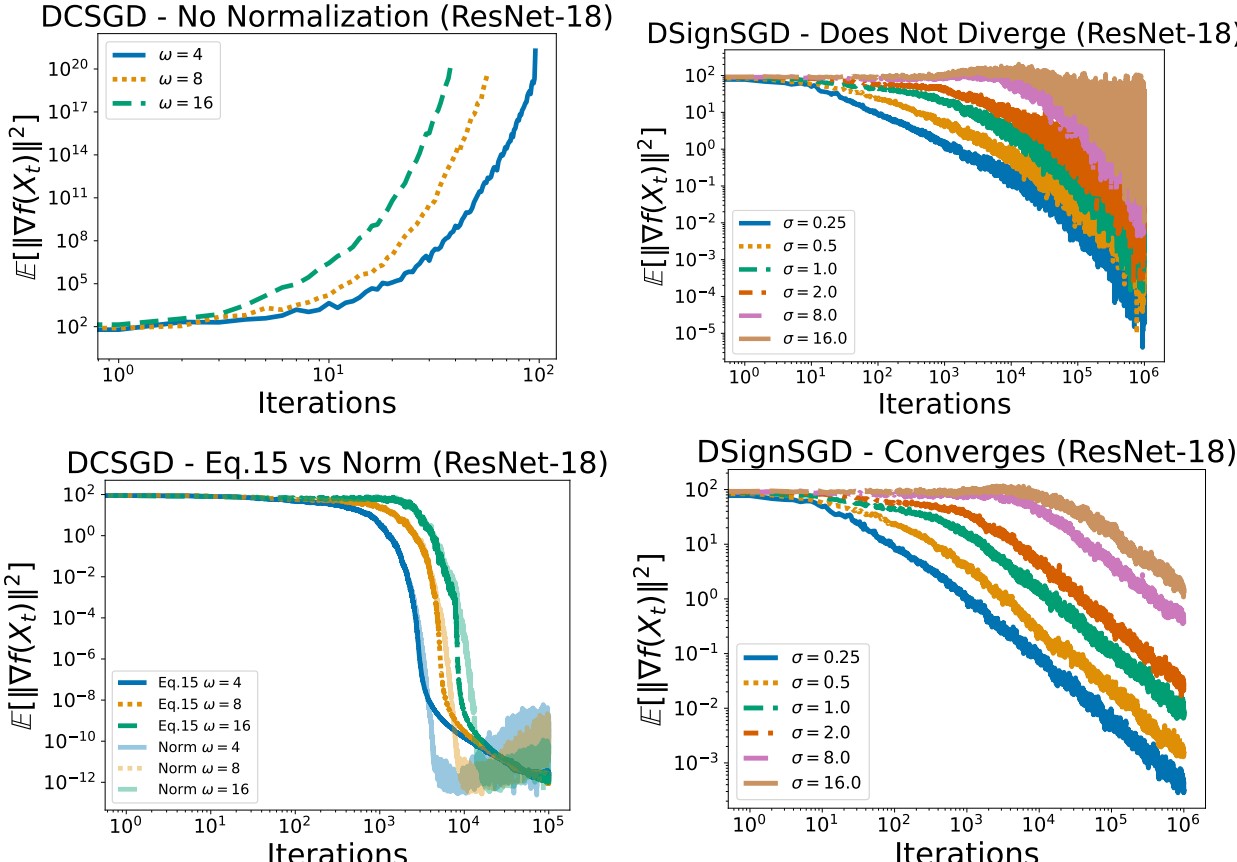

*Figure 3.* Sanity checks for our stability prescriptions on noisy, compressed ResNet-18 training on CIFAR-10 with $N = 8$ clients. For DCSGD with unbiased sparsification, we inject additive Gaussian gradient noise with affine variance $Z_t \sim \mathcal{N}(0, \sigma^2 \|g_t\|_2^2 I)$ and then apply random sparsification at compression levels $\omega$. Without any scheduler or normalization, DCSGD becomes unstable and the divergence worsens as $\omega$ increases (Top-Left). Using the adaptive normalization suggested by Thm. 4.2 through Eq. 15 stabilizes training and yields convergence for all tested $\omega$ (Bottom-Left). We also report a baseline that applies plain Normalized SGD under the same noise and compression, which exhibits a less stable profile. For DSignSGD under heavy-tailed noise, we inject Student's t gradient noise with $\nu = 1$ and different scale values $\sigma$. With a constant stepsize, DSignSGD remains stable but does not converge to zero (Top-Right), whereas the diminishing schedule prescribed by Thm. 4.3, here $\eta_t = 1/\sqrt{t+1}$, yields convergence across noise scales (Bottom-Right).

## E.5. Constructive Form of the Normalization Condition

The sufficient conditions for convergence of DCSGD (see Eq. 15) indicate that the learning rate schedule $\eta\eta_t$ should scale inversely with $\mathbb{E}\|\nabla f(X_t)\|$. While this may appear abstract, it admits a natural and practical implementation in the distributed setting.

**Client-side estimation.** At iteration $t$, each client $i$ already computes a stochastic gradient $\nabla f_{i,\gamma_i}(X_t)$ on a local mini-batch $\gamma_i$. We define the local norm estimate as

$$\hat{g}_i^t = \|\nabla f_{i,\gamma_i}(X_t)\|. \tag{164}$$

This requires no additional computation beyond what is standard for mini-batch gradient methods.

**Server-side aggregation.** The server maintains an estimate of the global gradient norm by averaging the client-side estimates as

$$\hat{G}_t = \frac{1}{N} \sum_{i=1}^{N} \hat{g}_i^t, \tag{165}$$

which provides a consistent approximation of $\mathbb{E}\|\nabla f(X_t)\|$.

This construction is intended for the standard server-aggregated distributed setting studied in this paper, where a central server already averages the client updates. It is not meant as a fully decentralized protocol: in peer-to-peer or gossip-style topologies, estimating the same global norm would require additional communication or a different decentralized estimator.

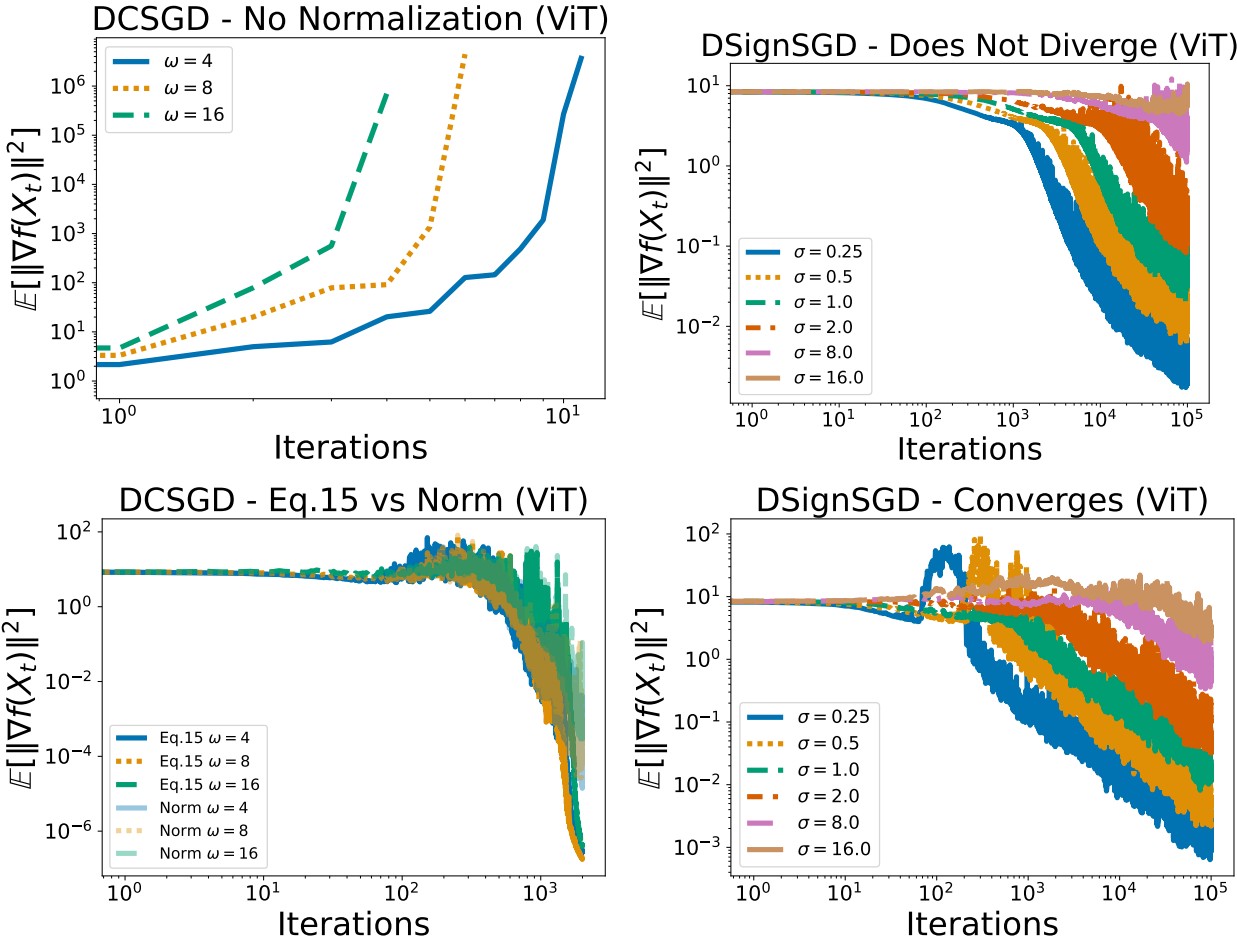

*Figure 4.* Sanity checks for our stability prescriptions on noisy, compressed ViT training on CIFAR-10 with $N = 8$ clients. For DCSGD with unbiased sparsification, we inject additive Gaussian gradient noise with affine variance $Z_t \sim \mathcal{N}(0, \sigma^2 \|g_t\|_2^2 I)$ and then apply random sparsification at compression levels $\omega$. Without any scheduler or normalization, DCSGD becomes unstable and the divergence worsens as $\omega$ increases (Top-Left). Using the adaptive normalization suggested by Thm. 4.2 through Eq. 15 stabilizes training and yields convergence for all tested $\omega$ (Bottom-Left). We also report a baseline that applies plain Normalized SGD under the same noise and compression, which exhibits a less stable profile. For DSignSGD under heavy-tailed noise, we inject Student's t gradient noise with $\nu = 1$ and different scale values $\sigma$. With a constant stepsize, DSignSGD remains stable but does not converge to zero (Top-Right), whereas the diminishing schedule prescribed by Thm. 4.3, here $\eta_t = 1/\sqrt{t+1}$, yields convergence across noise scales (Bottom-Right).

**Normalized learning rate.** A learning rate of the form

$$\eta \eta_t \sim \frac{\eta_0}{1 + \hat{G}_t} \tag{166}$$

satisfies the normalization condition in our bounds up to stochastic error. This adjustment can be implemented with negligible communication overhead, requiring each client to transmit only a single scalar per iteration.

