# OpenReview forum: "On the Interaction of Batch Noise, Adaptivity, and Compression, under $(L_0,L_1)$-Smoothness: An SDE Approach"
_ICML.cc/2026/Conference — ICML 2026 regular_

### Official Review · Reviewer_GvAq · 2026-03-02

**Soundness:** 3
**Presentation:** 3
**Significance:** 3
**Originality:** 3
**Overall Recommendation:** 4
**Confidence:** 2

**Summary:**

The paper developed a unified framework based on corrected SDE surrogates that more accurately captures the discrete time behavior, to analyze distributed compressed SGD and distributed sign SGD. The authors studies the interaction of stochasticity, compression and adaptive/normalized updates within the framework under the $(L_0, L_1)$-smoothness condition.

**Compliance With Llm Reviewing Policy:**

Affirmed.

**Final Justification:**

I would like to keep my positive score.

**Key Questions For Authors:**

See strengths and weaknesses.

**Limitations:**

Yes.

**Strengths And Weaknesses:**

Strengths:

1. Presentation: The paper is well organized. The logic is clearly depicted and easy to understand.
2. Soundness: The proposed correction is not presented as a weak heuristic only, there are sanity checks and justification for why the sign and term is chosen.
3. Significance: The unification is useful, since many of the components coexist in distributed training.
4. Originality: Novel, the correction is based on the standard SDE literature, the combination of consideration in this setting is not considered before.

Weaknesses:
1. Most guarantees are for the SDE surrogates only but not the algorithm we actually run. Though the authors argue that the surrogate has better predicting power, but the link is still a bit implicit.
2. Empirical validations are minimal.

---

> ### Author Rebuttal · Authors · 2026-03-26
>
> Thank you for the positive assessment and for highlighting the organization, soundness, and usefulness of the unified viewpoint. Please find below our replies, which include **new experiments with a ResNet-18 and a ViT trained on CIFAR-10**, which we believe significantly strengthen the experimental validation of our theoretical contribution.
>
> ### 1. **On guarantees being for the SDE models rather than the discrete algorithm**
>
> We respectfully point the Reviewer to the existing ‘**Scope of SDE guarantees and results**’ discussion on Page 4 of our paper, where we cover this aspect and point to the relevant literature.
>
> This paper is indeed a **modeling paper**: the continuous-time processes are not intended to reproduce the iterate-by-iterate trajectory exactly. The formal link is through **weak approximation on finite horizons**, and the main reason we emphasize this modeling viewpoint is precisely that weak approximation alone does not guarantee the kind of **stability fidelity** we care about. In fact, one of the paper’s central observations is that multiple weakly accurate models can behave very differently with respect to stepsize stability.
>
> Our contribution is therefore twofold:
> 1. to identify that classical models can fail on the stability question, and
> 2. to construct a model that is still first-order weakly accurate while also being **faithful to the relevant stability behavior**.
>
>
> ### 2. **On empirical validation being minimal**
>
> We agree that the experiments are intentionally modest in scope. We emphasize that this paper does **not** propose a new optimizer; rather, it studies **existing distributed optimizers** that are standard and well-studied in the literature under $L$ smoothness, while we extend the theoretical analysis to the more general $(L_0,L_1)$-smoothness. Accordingly, the empirical goal is not to demonstrate state-of-the-art downstream performance, but to test the **qualitative mechanisms** predicted by the theory:
> - whether unnormalized DCSGD becomes unstable in the predicted regimes,
> - whether the normalization suggested by the theory stabilizes it,
> - and whether DSignSGD exhibits the expected robustness pattern.
>
> The synthetic examples isolate these mechanisms as cleanly as possible, and the MLP experiment was included as a neural-network sanity check showing the same trend beyond toy scalar dynamics. We appreciate the reviewer’s positive reading of the theoretical contribution, and in the revision, we will make it more explicit that the experiments are intended as **mechanism validation**, consistent with the theoretical nature of the paper, rather than as competitive benchmarking.
>
> To further address the reviewer’s concern, **we have also carried out additional experiments** on **ResNet-18** and **ViT** on **CIFAR-10** in a distributed setting with $N=8$ clients: Results are perfectly in line with our theory and with results already shown in Figure 1. Concretely, for **DCSGD** we inject additive Gaussian gradient noise with affine variance $Z_t\sim\mathcal{N}(0,\sigma^2\|g_t\|_2^2 I)$ and apply unbiased random sparsification at compression levels $\omega\in(4,8,16)$: without scheduler/normalization, DCSGD becomes unstable, and the divergence worsens as $\omega$ increases; with the adaptive normalization suggested by Thm. 4.2 (Eq. 15), training is stabilized and converges for all tested $\omega$, while plain Normalized SGD under the same noise/compression exhibits a visibly less stable profile. For **DSignSGD**, we inject Student’s $t$ gradient noise with $\nu=1$ and scale $\sigma\in(0.25,0.5,1,2,8,16)$: with a constant stepsize, DSignSGD remains stable but does not always converge, whereas the diminishing schedule prescribed by Thm. 4.3 (here $\eta_k=1/\sqrt{k+1}$) yields convergence across noise scales. These new experiments are not intended to change the scope of the paper, but rather to verify that the same qualitative stability patterns predicted by the theory also appear on more realistic architectures. We will include these results and the corresponding implementation details in the revised version. Please find the plots and the respective captions at **[[URL](https://anonymous.4open.science/r/Rebuttal_ICML-C7FF)]**.

---

> > ### Author Rebuttal · Reviewer_GvAq · 2026-04-01
> >
> > I thank the authors for their efforts in addressing my concerns, and I am now inclined to support acceptance of the paper.

---

> > > ### Author Response · Authors · 2026-04-04
> > >
> > > Dear Reviewer GvAq,
> > >
> > > Thank you for your **positive assessment** and for confirming that your concerns have been **"Fully resolved"**. We are very grateful for your encouraging words and for stating that you are _"now inclined to **support acceptance** of the paper"_.
> > >
> > > In line with this and with the **conference guidelines** for reviews marked as **"Fully resolved"**, we would **kindly ask you to consider raising your score accordingly**.
> > >
> > > Thank you again for your constructive engagement, which has contributed to strengthening our paper.
> > >
> > > Best regards,
> > >
> > > The Authors

---

### Official Review · Reviewer_QxeQ · 2026-03-10

**Soundness:** 2
**Presentation:** 1
**Significance:** 2
**Originality:** 3
**Overall Recommendation:** 4
**Confidence:** 4

**Summary:**

This paper analyzes and models the continuous-time dynamics of two distributed optimization algorithms under the recent $(L_0, L_1)$-smoothness assumption. It also considers compressed gradient updates and a more general noise assumption than the standard bounded variance assumption. The authors point out an issue in standard continuous-time models, namely, standard first-order and second-order Stochastic Differential Equations (SDEs) don't capture the discrete-time constraints on the step-size required for stable convergence, often incorrectly predicting unconditional convergence. To resolve this issue, the authors introduce a new "stability-faithful" SDE which correctly reflects the step-size constraints of the discrete counterparts. Using the proposed SDE scheme, convergence results are presented for the continuous versions of two distributed optimization algorithms, viz., Distributed Compressed SGD (DCSGD) and Distributed SignSGD (DSignSGD).

**Compliance With Llm Reviewing Policy:**

Affirmed.

**Final Justification:**

I increased my rating from 3 to 4 after the rebuttal discussion. The authors included some decent discussion about how their proposed SDE model can improve upon existing models, which prompted me to increase my rating. However, I still think this is a borderline paper.

**Key Questions For Authors:**

The proposed SDE also has a $\nabla^2 f (X_t) \nabla f (X_t)$ term like the second-order SDE introduced in the paper. Then why are the authors calling their proposed SDE first-order? Am I missing something here?

**Limitations:**

Limitations have not been discussed. This is a theoretical paper and therefore the authors discuss that there are no direct foreseeable negative societal impacts.

**Strengths And Weaknesses:**

**Strengths:** The authors identify a critical flaw of existing SDE models, namely that they don't reflect the step-size constraints required for stable convergence. The authors propose a new SDE model which doesn't suffer from this issue, and can therefore model the discrete-time algorithm better.

**Weaknesses:**

*Significance:* My biggest concern and complaint is about the value and significance of this work. What is a real tangible use of introducing another SDE model? Will the new introduced SDE be useful for some applications where the existing SDEs (which don't capture the step-size constraint) can't be applied? Or will it improve some applications that the existing SDEs are being currently used for? I couldn't find any meaningful discussion on the (practical) value of the contributions of this paper. My rating is primarily due to my significance concerns.

*Presentation:* There are a few presentation issues:

1. To me the main message of the paper is about a new SDE that captures the step-size constraints of the discrete-time counterpart. To this end, the distributed setting and compression seem like distractions and only complicate the presentation and exposition. I recommend focusing on the main story in the standard centralized setting (without any compression).

2. In Theorem 4.1 and 4.2, what is the meaning of "a random time $\hat{t}$ (...) with distribution $\frac{\eta_t}{\phi_t^{1}}$"? Similar question about the language in Theorem 4.3. This is not standard jargon.

3. The use of $\eta$ and $\eta_k$ is confusing. Also, $\phi_t^2$ can be confused to be the square of $\phi_t$ (maybe use $\phi_t^{(i)}$ for more clarity). Please use better notation.

4. On line 186-187 left column, $L^{1}(\mathbb{R}^d)$ and $L^{2}(\mathbb{R}^d)$ are not introduced.

5. I don't understand what $O_{x_k}(\eta)$ is in eq. (8). Can the authors please derive eq. (8) properly?

---

> ### Author Rebuttal · Authors · 2026-03-26
>
> Thank you for the careful reading and for identifying both the main conceptual contribution and the places where the exposition can be improved.
>
> ### 1. **On the significance of introducing a new SDE model**
> Ours is not just "another SDE": the key point is that **existing SDE models can give qualitatively wrong answers**, as they can predict unconditional convergence in regimes where the actual discrete optimizer is unstable. This matters practically because SDEs are **already used** to guide step-size/batch-size tuning and scaling-law analyses; if the surrogate gets stability wrong, downstream prescriptions can also be wrong.
>
> Our model matters because it restores this missing fidelity:
> - it recovers the correct stability threshold in the quadratic sanity check,
> - it captures the absence of a universal stable constant stepsize in the $(L_0,L_1)$ setting,
> - and it yields concrete qualitative prescriptions for distributed methods, e.g., why DCSGD requires noise/compression-dependent normalization and why DSignSGD behaves differently under heavy-tailed noise.
>
> So the contribution is not "a new SDE for its own sake": it is a **stability-faithful analytical model** for a regime where standard models fail in exactly the aspect we want to study.
>
> ### 2. **On the role of distributed training and compression**
>
> We appreciate this presentation comment. We respectfully prefer to **keep** the distributed/compression setting, because it is not incidental to the paper: it is where the corrected model yields genuinely new stability prescriptions. For example, it explains why DCSGD requires a noise/compression-dependent stepsize normalization, while DSignSGD remains stable under qualitatively different step-size restrictions.
>
> Regarding the organization of the paper, the quadratic/quartic examples are included precisely to isolate the core conceptual mechanism in the cleanest possible way. The distributed/compressed analysis is then the main application layer, where the corrected SDE reveals how stability depends jointly on:
> - $(L_0,L_1)$-geometry,
> - compression strength,
> - dimension,
> - and the noise model.
>
> We agree, however, that the separation between the **core conceptual point** and the **distributed application** can be made clearer. In the revision, we will improve the signposting so that the reader sees more clearly:
> 1. the examples as the diagnostic counterexamples motivating the correction
> 2. the distributed/compressed setting as the main application.
>
> ### 3. **On the “random time” notation in Thms 4.1–4.3**
>
> In line with recent works from the literature [1,2,3] which use a similar formulation and jargon, what we mean is a weighted sampling of a time in $[0,t]$. We will rewrite this explicitly, e.g.
> > let $\hat t$ be a random time in $[0,t]$ with density $s \mapsto \eta_s / \phi_t^{(1)}$.
>
> We will also add one explanatory sentence after the theorem saying that this is simply the standard weighted-average-output approach used in the proof.
>
> [1] Continuous-time models for stochastic optimization algorithms, Orvieto et al. NeurIPS 2019
>
> [2] Unbiased and sign compression in distributed learning: Comparing noise resilience via SDEs, Compagnoni et al. AISTATS 2025
>
> [3] Continuous-Time Analysis of Federated Averaging, Overman et al. Under Review
>
> ### 4. **On notation ($\eta$ vs $\eta_k$, $\phi_t^2$)**
>
> We agree. We will clean up the notation as suggested.
>
> ### 5. **On $L^1(\mathbb{R}^d)$ and $L^2(\mathbb{R}^d)$**
>
> We agree. We will define these spaces explicitly at first use.
>
> ### 6. **On Eq. 8 and the $O(\eta^2,x_k)$ notation.**
>
> We agree that the derivation in the submission was too terse. Here, we spell out Eq. 8 directly from the second-order Taylor expansion of $f$ along the GD step $x_{k+1}=x_k-\eta \nabla f(x_k)$. This gives
>
> \begin{equation}
> \frac{f(x_{k+1})-f(x_k)}{\eta} = -\|\nabla f(x_k)\|_2^2 + \frac{\eta}{2} \nabla f(x_k)^\top \nabla^2 f(x_k) \nabla f(x_k) + O(\eta^2,x_k)
> \end{equation}
>
> where $O(\eta^2,x_k)$ means a term bounded by $C(x_k)\eta^2$ for small enough $\eta$, with $C(x_k)$ independent of $\eta$.
>
> ### 7. **On why the proposed SDE is called “first-order”**
>
> Thank you for raising this; the confusion is understandable and we should clarify it explicitly.
>
> Here, “first-order” refers to the **power of $\eta$ in the weak-approximation error in Def. 3.4** (equivalently, the order of weak approximation in $\eta$), not to the absence of Hessian terms in the drift. A Hessian can appear in a first-order weak approximation. We will add a one-sentence clarification the first time this terminology is used.
>
> ### 8. **On limitations not being discussed**
>
> We apologize that this was easy to miss. The paper does include a limitations discussion in **App. D.4**. There we state, among others, that:
> - we focus on homogeneous clients
> - we do not study error-feedback or general biased compressors
>
> In the revision, we will add a limitations paragraph into the main text so that it is not buried in the appendix.

---

> > ### Author Rebuttal · Reviewer_QxeQ · 2026-04-03
> >
> > Thanks for the detailed response and clarifications.
> >
> > However, I’m still not convinced by the significance and practical utility of the proposed SDE models. In their response, the authors state: “This matters practically because SDEs are already used to guide step-size/batch-size tuning and scaling-law analyses; if the surrogate gets stability wrong, downstream prescriptions can also be wrong.” Can the authors be more precise about how and which (practically-used) learning rate schedules or batch size schemes or scaling law analyses are incorrectly guided by existing SDE models and how their proposal fixes this? I genuinely feel this is an important point to address clearly, given that this is an ML conference.
> >
> > Also, I’m still not a big fan of keeping the distributed setting and compression; after all, these are not first-order aspects people think about while introducing new ideas in optimization (unless such ideas are specifically tailored to these settings, which is not really the case in this paper). The authors say “For example, it explains why DCSGD requires a noise/compression-dependent stepsize normalization, while DSignSGD remains stable under qualitatively different step-size restrictions.” To me this is a nice insight, but it might be primarily due to the inherent difference between the non-adaptive nature of SGD and adaptive(-ish) nature of Sign-SGD, rather than due to the distributed setting or compression, unless I’m mistaken. Please correct me if I’m wrong, but if I’m not wrong, maybe the authors could focus on this difference between non-adaptive and adaptive algorithms? (However, I’m willing to be somewhat lenient regarding this meta-point.)
> >
> > I’ll keep my score.

---

> > > ### Author Response · Authors · 2026-04-04
> > >
> > > ## 1. A broader perspective.
> > >
> > > Our contribution is primarily **methodological**: We introduce **stability-faithful SDEs** especially designed for $(L_0,L_1)$-smoothness, a regime that better models modern DNN landscapes and in which classical SDEs can be qualitatively misleading about **stepsize stability**.
> > >
> > > This is important because SDEs are used in optimization to study convergence bounds, learning-rate schedules, batch-size control, and scaling laws. **Our contribution opens up these same questions in the $(L_0,L_1)$-smooth setting with SDEs that correctly reflect the relevant stability restrictions.**
> > >
> > > **Here are concrete classes of SDE-based analyses** that would change under our corrected models:
> > >
> > > 1. **Batch size schemes.** Zhao et al. (2022) derive a batch size scheme via a classic SDE that imposes no stability constraint: the optimal scheme is derived under the assumption that the dynamics converge for any $\eta/B$. Our new SDE would introduce a constraint into the control problem, ensuring that the schedule does not push $\eta/B(t)$ outside the stable regime.
> > > 2. **Learning rate schedules.** SDEs are often used to derive convergence bounds on suboptimality or gradient norm (Orvieto et al. (2019a), Compagnoni et al. (2024; 2025a;b)). Our SDEs give guarantees that are jointly informative about the convergence rate and the admissible stepsize region. This is practically relevant because, without such restrictions, practitioners relying on the results above would be misinformed about **how to set learning rate schedules**.
> > > 3. **Scaling laws.** Many works use SDEs to derive scaling laws such as the $\eta/ B$ linear scaling rule (Jastrzebski et al., 2018) and the $\eta/\sqrt{B}$ square-root one for Adam (Compagnoni et al., 2025a). Our restrictions provide *validity limits* on those scaling laws, e.g. in terms of $B$.
> > >
> > > Finally, works relying on SDEs have **often focused on insight rather than direct practical results**:
> > > 1. Xie et al. (ICLR 2021) studied expected times for SGD to escape minima, which would lead to different conclusions under $(L_0, L_1)$-smoothness when using our SDEs;
> > > 2. Li et al. (ICML 2017) limited themselves to formalizing SDE derivations for optimization methods;
> > > 3. Cohen et al. (ICLR 2025) proposed a different ODE model for SGD and RMSprop to study the edge of stability;
> > > 4. Simsekli et al. (ICML 2019) showed that SGD's diffusion term must be more complex than the standard one to capture heavy-tailed gradient noise. Similarly, we propose a better drift term to study SGD stability, especially under $(L_0,L_1)$-smoothness.
> > >
> > > **The SDE literature at top ML venues has a strong tradition of insight-driven contributions, and ours fits squarely within it: we show how to adapt SDEs for use in the context of modern neural networks.**
> > >
> > > SDEs have been increasingly adopted, and many applications were not anticipated when they were introduced. **Our conceptual contribution - that different phenomena require different SDEs - establishes a modeling principle that we expect will enable new applications and insights.**
> > >
> > > ## 2. On the role of adaptivity and compression.
> > >
> > > We agree that this insight is rooted in the fact that DCSGD is not adaptive while DSignSGD is, and that this could also have been shown in the single-node setting without compression.
> > >
> > > However, $(L_0,L_1)$-smoothness is increasingly studied in compression and distributed settings, and is receiving even more attention: Appendix A gives a detailed literature review documenting the rapid expansion of this community. Importantly, in modern architectures, $L_0$ is often 0 while $L_1$ is positive ("From Muon to Gluon: Bridging Theory and Practice of LMO-based Optimizers for LLMs", Riabinin et al., 2025). Thus, while much of DL theory and the literature on compression and distributed learning still relies on $L$-smoothness, we provide the first SDE-based angle on this active direction.
> > >
> > > As Table 1 shows, our results are novel and recover known results in simpler cases, thus supporting the validity of our SDE. We respectfully believe that the compression layer strengthens rather than weakens the contribution: it immediately shows that our SDE is useful in cases never analyzed before and can serve a rapidly expanding community.
> > >
> > > Our stability-faithful SDE allows us, for the first time, to *quantitatively characterize* the mechanisms behind this difference. Classical SDE-based results never captured any restriction on the learning rate, and therefore could not reveal *how much* normalization (DC)SGD needs or *why* (D)SignSGD's implicit normalization suffices. Our framework makes this **quantitative**: Eq. 15 shows explicitly how the required normalization strength depends on the compression rate $\omega$, the noise parameters $(\sigma_0^2,\sigma_1^2)$, the $(L_0,L_1)$-geometry, and the dimension $d$.
> > >
> > > We truly appreciate how the Reviewer has contributed to clarifying these aspects of our paper and **would appreciate reconsideration of the score**.

---

### Official Review · Reviewer_VvPr · 2026-03-11

**Soundness:** 3
**Presentation:** 3
**Significance:** 2
**Originality:** 3
**Overall Recommendation:** 3
**Confidence:** 2

**Summary:**

This paper develops a unified SDE (stochastic differential equation) framework to analyze Distributed Compressed SGD (DCSGD) and Distributed SignSGD (DSignSGD) under $(L_0, L_1)$-smoothness. It shows that classical SDE can not accurately capture stability and introduces new first-order SDE models. Based on this model, the paper gives convergence results under noise and compression settings.

**Compliance With Llm Reviewing Policy:**

Affirmed.

**Key Questions For Authors:**

See Weakness

**Limitations:**

The experimental validation is somewhat limited. It would strengthen the paper to include deep learning experiments on benchmark datasets and compare the proposed method with existing distributed optimization methods.

**Strengths And Weaknesses:**

**Strengths**

1. **Clear motivation and illustrative examples.** The paper effectively demonstrates the limitations of classical SDE models using simple quadratic and quartic functions (Section 4.2). These examples make the core problem accessible and convincing.
2. **Unified theoretical framework.** To the best of my knowledge, this is the first SDE-based analysis that simultaneously addresses $(L_0, L_1)$-smoothness, gradient compression, and general noise models. Table 1 clearly shows this contribution compared with prior work.

**Weaknesses**

1. **The ansatz approach is not sufficiently justified** The new SDE model is derived by choosing $\alpha = +1/2$ to match the desired stability behavior (Eq. 11-13). But this "design by target" approach raises my concerns. Could the authors provide deeper justification for why matching the discrete-time generator is more appropriate than matching finite increments (as in classical modified equations)?

2. **Stability conditions depend on unknown, time-varying quantities.** The key condition (Eq. 15) involves $E[\|\nabla f(X_t)\|]$, which is unknown and cannot be computed from a single run. The estimator in Appendix E.4 needs a central node to accumulate the information and needs additional communication instead, which can not be achieved in fully decentralized implementations. This also contradicts the motivation of communication-efficient distributed optimization.

3. **Limited experimental validation.** Experiments are restricted to synthetic functions and simple MLPs. Validation on realistic datasets and architectures would strengthen the paper's practical relevance.

---

> ### Author Rebuttal · Authors · 2026-03-26
>
> Thank you for the thoughtful review and for recognizing the motivation and the novelty of our unified framework.
>
> ### 1. On the ansatz and $\alpha=1/2$
>
> As discussed in Lines 62-69 and 141-151, our intent is **not** to claim that generator matching is universally the right criterion for each question. Our point is that for the specific question studied in this paper, i.e., **stepsize stability**, one needs to match the induced loss drift with the discrete-time generator applied to $f$ rather than use the classic approach as this leads to SDEs which cannot capture the stability thresholds at all.
>
> For GD, the discrete loss expansion has the form
> $$
> \frac{f(x_{k+1})-f(x_k)}{\eta}=-\|\nabla f(x_k)\|_2^2+\frac{\eta}{2}\nabla f(x_k)^\top\nabla^2 f(x_k)\nabla(x_k)+O(\eta^2),
> $$
> while the classical 1st-order model misses the $O(\eta)$ correction and the classical 2nd-order model introduces the same term but with the **wrong sign**. This is exactly what leads classical models to predict benign or even accelerated behavior in regimes where the discrete method is unstable.
>
> Within the ansatz family
> $$
> dX_t=-\nabla f(X_t)dt+\alpha\eta\nabla^2 f(X_t)\nabla f(X_t)dt,
> $$
> matching the induced loss drift with the discrete-time loss generator selects $\alpha = 1/2$ as the unique choice to recover the correct stability threshold. So this is not “design by target” in an arbitrary sense; it is a **property-faithful model selection criterion** tailored to the **stability question**.
>
> ### 2. On the use of $E\|\nabla f(X_t)\|$
>
> As explained in Lines 386 - 392 in our paper, Eq. 15 provides a **sufficient stability region**, analogous to the classical condition $\eta < 2/L$, where $L$ **is not actually computable**. Still, it is **commonly used** in restrictions on the learning rate. Under $(L_0,L_1)$-smoothness, stability can become **geometry dependent**: there may be no single constant stepsize that is stable uniformly over all initializations. In that sense, the appearance of $E\|\nabla f(X_t)\|$ is not just a proof artifact. Similar conditions can be found in
>  - Optimizing $(L_0,L_1)$-Smooth Functions by Gradient Methods, Vankov et al., ICLR 2025;
>  - Why Gradient Clipping Accelerates Training: A Theoretical Justification for Adaptivity, Zhang et al., ICLR 2020;
>  - Methods for Convex $(L_0,L_1)$-Smooth Optimization, Gorbunov et al., ICLR 2025.
>
> Regarding implementation, App. E.4 was mean as a **constructive demonstration** that the condition can be approximated in the standard server-aggregated distributed setting considered in the paper (cf. Eq. 4–5, where global model is updated by averaging over N clients). In that setting, each client computes a batch gradient, and the additional cost is only transmitting **one scalar** (a gradient-norm estimate) per round.
>
> In the revision, we will:
> - clarify that App. E.4 applies to the server-aggregated setting and is not intended as a fully decentralized protocol;
> - add the centralized-topology assumption to the Limitations in App. D.4, noting that fully decentralized implementations are outside our scope.
>
> ### 3. On limited experimental validation
>
> We agree that the experiments are intentionally modest in scope, and we will present them more clearly as **qualitative sanity checks**, as also noted by Reviewer GvAq. We emphasize that this paper does **not** propose a new optimizer; rather, it studies **existing distributed optimizers** that are standard and well-studied in the literature under $L$ smoothness, while we extend the theoretical analysis to the more general $(L_0,L_1)$-smoothness. Accordingly, the empirical goal is not to demonstrate SOTA downstream performance, but to test the **qualitative mechanisms** predicted by the theory:
> - whether unnormalized DCSGD becomes unstable in the predicted regimes,
> - whether the normalization suggested by the theory stabilizes it,
> - whether DSignSGD exhibits the expected robustness.
>
> We appreciate the reviewer’s positive reading of the theoretical contribution, and in the revision we will make it more explicit that the experiments are intended as **mechanism validation**, consistent with the theoretical nature of the paper, rather than as competitive benchmarking.
>
> To address the reviewer’s concern, **we have also carried out additional experiments** on **ResNet-18** and a **ViT** on **CIFAR-10** in a distributed setting with $8$ clients. These new experiments are not intended to change the scope of the paper, but rather to verify that the same qualitative stability patterns predicted by the theory also appear on more realistic architectures. We will include these results and implementation details in the revised version. Please, find the plots and the respective captions at **[[URL](https://anonymous.4open.science/r/Rebuttal_ICML-C7FF)]**, which show patterns similar to those shown in Fig. 1. **For brevity**, we kindly refer the Reviewer to our reply to **Reviewer GvAq** for additional details on the experimental setup and observations.

---

> > ### Author Rebuttal · Reviewer_VvPr · 2026-04-02
> >
> > Thank you for the authors’ response. I will maintain my score.

---

> > > ### Author Response · Authors · 2026-04-04
> > >
> > > Dear Reviewer VvPr,
> > >
> > > Thank you for confirming that your concerns have been **"Fully resolved"**. We are pleased that our responses **have fully addressed your points** regarding:
> > > 1. the justification of the ansatz,
> > > 2. the role of $\mathbb{E}[|\nabla f(X_t)|]$,
> > > 3. and the **additional experiments** on ResNet-18 and ViT.
> > >
> > > Since the three concerns that motivated your original score have now all been **fully resolved**, we would kindly ask you to consider whether an **upward adjustment to your score** may be warranted, in accordance with the **conference guidelines** for reviews marked as **"Fully resolved"**.
> > >
> > > Thank you again for your time and constructive engagement, which has meaningfully improved our paper.
> > >
> > > Best regards,
> > >
> > > The Authors

---

### Official Review · Reviewer_iAxU · 2026-03-12

**Soundness:** 3
**Presentation:** 3
**Significance:** 3
**Originality:** 3
**Overall Recommendation:** 4
**Confidence:** 1

**Summary:**

This paper studies the interaction between **batch noise, gradient compression, and adaptivity** in distributed stochastic optimization under the **\((L_0, L_1)\)-smoothness** assumption. The authors investigate whether commonly used **SDE approximations** faithfully capture the stability behavior of discrete-time optimization algorithms.

The paper argues that classical first- and second-order SDE models may **misrepresent the learning-rate stability conditions** of discrete optimization methods. To address this issue, the authors propose **stability-faithful SDE models** that modify the drift term using curvature-dependent corrections.

Using the proposed framework, the paper analyzes:

- Distributed Compressed SGD (DCSGD) with unbiased compressors and affine variance noise
- Distributed SignSGD (DSignSGD) under heavy-tailed gradient noise

The analysis derives learning-rate stability conditions and convergence properties under \((L_0,L_1)\)-smoothness. The results characterize how **compression level, gradient noise, dimension, and number of clients** influence optimization stability and convergence.

**Compliance With Llm Reviewing Policy:**

Affirmed.

**Final Justification:**

I will keep my score.

**Key Questions For Authors:**

No

**Strengths And Weaknesses:**

### 1. Important theoretical question

The paper addresses an important modeling question: whether continuous-time **SDE approximations faithfully represent discrete optimization dynamics**. Since SDE methods are widely used in optimization theory, understanding their limitations is valuable.

---

### 2. Stability-faithful SDE formulation

The paper proposes a modified SDE model that introduces a curvature-dependent drift correction. This modification helps recover the correct **learning-rate stability thresholds** observed in discrete-time algorithms.

The idea of matching the **discrete-time loss drift expansion** with the SDE drift term is technically interesting.

---

### 3. Unified treatment of multiple system components

The work analyzes several factors simultaneously:

- distributed training
- gradient compression
- affine-variance noise
- heavy-tailed gradient noise
- relaxed smoothness assumptions

Most prior works study these components separately, so this unified perspective is valuable.

---

### 4. Broader noise assumptions

The analysis goes beyond standard bounded-variance noise assumptions and considers:

- affine variance noise
- heavy-tailed gradient noise

This improves the relevance of the theoretical results for practical deep learning scenarios.

---

### 5. Useful qualitative insights

The analysis provides several insights:

- stronger compression reduces stability regions
- higher dimensionality worsens stability
- more clients may stabilize optimization
- gradient normalization is important for DCSGD
- sign-based updates inherently provide normalization

These observations are consistent with empirical practices.

I am **not** familiar this research area, so I can not verify the correctness of the proofs and claims in this paper.

---

> ### Author Rebuttal · Authors · 2026-03-26
>
> Thank you for the positive assessment and for highlighting what you viewed as the main strengths of the paper.
>
> We are glad that the reviewer found the following aspects valuable:
> - the importance of asking whether SDE models faithfully reflect discrete-time optimization dynamics,
> - the stability-faithful drift correction,
> - the unified treatment of distributed training, compression, and nonstandard noise models under $(L_0,L_1)$-smoothness,
> - and the qualitative insights about how compression, dimension, noise, and normalization affect stability.
>
> We appreciate in particular that the reviewer recognized the intended contribution as both **conceptual and technical**.

---

> > ### Author Rebuttal · Reviewer_iAxU · 2026-04-04
> >
> > i will keep my positive score.

---

> > > ### Author Response · Authors · 2026-04-04
> > >
> > > Dear Reviewer iAxU,
> > >
> > > Thank you again for your **positive assessment** and for selecting "**Fully resolved**". We are pleased that our rebuttal confirmed your **positive** view of our work.
> > >
> > > In line with the conference guidelines regarding reviews marked as "**Fully resolved**", we would **kindly ask you to consider updating your score accordingly**.
> > >
> > > Thank you for your time and engagement.
> > >
> > > Best regards,
> > >
> > > The Authors

---

### Decision · Program_Chairs · 2026-04-30

**Decision:**

Accept (regular)

**Comment:**

The paper makes a clear and original theoretical contribution by introducing stability-faithful SDE surrogates for distributed compressed optimization under \((L_0,L_1)\)-smoothness, and the reviewers broadly agree that its core insight is both technically meaningful and timely. In particular, the work is credited for identifying an important limitation of classical first- and second-order SDE models, proposing a principled correction that better captures discrete-time stability behavior, and unifying several factors that are often studied separately, including distributed training, compression, affine-variance and heavy-tailed noise, and normalized or sign-based updates. While the discussion raised concerns about the practical significance of the new SDE models, the scope of the empirical validation, and the extent to which the guarantees apply directly to the surrogate rather than the discrete algorithm, the authors addressed these points constructively in the rebuttal by clarifying the modeling goal of the paper, strengthening the discussion of practical relevance for learning-rate, batch-size, and scaling-law analyses, and adding further experimental evidence on more realistic architectures. Overall, the post-rebuttal record shows that the main concerns were either resolved or substantially mitigated, and the paper now presents a coherent, technically solid, and potentially influential contribution that is worthy of acceptance.